

# Carbon uptake and biogeochemical change in the Southern Ocean, south of Tasmania

Pardo, Paula C.[1], Tilbrook, Bronte[1,2], Langlais, Clothilde.[2], Trull, Tom W.[1,2], Rintoul, Steve R.[1,2]

[1]Antarctic Climate and Ecosystem Cooperative Research Centre, University of Tasmania, Hobart, Australia
[2]Climate Science Centre, CSIRO Oceans and Atmosphere, Hobart, Australia

*Correspondence to*: Paula C. Pardo (paula.condepardo@csiro.au)

**Abstract.** Biogeochemical change in the water masses of the Southern Ocean, south of Tasmania, was assessed for the 16-year period between 1995 and 2011 using data from 4 summer repeats of the WOCE/JGOFS/CLIVAR/GO-SHIP SR03 hydrographic section (at ~140°E). Changes in temperature, salinity, oxygen, and nutrients were used to disentangle the effect
of solubility, biology, circulation and anthropogenic carbon ($C_{ANT}$) uptake on the variability of dissolved inorganic carbon (DIC) for 8 water mass layers defined by neutral surfaces ($\Upsilon^n$). $C_{ANT}$ was estimated using an improved back-calculation method. Warming (~0.0352 ± 0.0170 °C yr$^{-1}$) of Subtropical Central Water (STCW) and Antarctic Surface Water (AASW) layers decreased their gas solubility, and accordingly DIC concentrations increased less rapidly than expected from equilibration with rising atmospheric $CO_2$ (~0.86 ± 0.16 µmol kg$^{-1}$ yr$^{-1}$ versus ~ 1 ± 0.12 µmol kg$^{-1}$ yr$^{-1}$). An increase in
apparent oxygen utilisation (AOU) occurred in these layers due to either remineralization of organic matter or intensification of upwelling. The range of estimates for the increases of $C_{ANT}$ were 0.71 ± 0.08 to 0.93 ± 0.08 µmol kg$^{-1}$ yr$^{-1}$ for STCW and 0.35 ± 0.14 to 0.65 ± 0.21 µmol kg$^{-1}$ yr$^{-1}$ for AASW, with the lower values in each water mass obtained by assigning all the AOU change to remineralization. DIC increases in the Sub-Antarctic Mode Water (SAMW, 1.10 ± 0.14 µmol kg$^{-1}$ yr$^{-1}$) and Antarctic Intermediate Water (AAIW, 0.40 ± 0.15 µmol kg$^{-1}$ yr$^{-1}$) layers were similar to the calculated $C_{ANT}$ trends. For SAMW,
the $C_{ANT}$ increase tracked rising atmospheric $CO_2$. As a consequence of the general DIC increase, decreases in total pH ($pH_T$) and aragonite saturation ($\Omega_{Ar}$) were found in most water masses, with the upper ocean and the SAMW layer presenting the largest trends for $pH_T$ decrease (~ -0.0031 ± 0.0004 yr$^{-1}$). DIC increases in deep and bottom layers (~ 0.24 ± 0.04 µmol kg$^{-1}$ yr$^{-1}$) resulted from the advection of old deep waters to resupply increased upwelling, as corroborated by increasing silicate (~ 0.21 ± 0.07 µmol kg$^{-1}$ yr$^{-1}$), which also reached the upper layers near the Antarctic Divergence (~ 0.36 ± 0.06 µmol kg$^{-1}$ yr$^{-1}$)
and was accompanied by an increase in salinity. The observed changes in DIC over the 16-year span caused a shoaling (~340 m) of the aragonite saturation depth (ASD, $\Omega_{Ar}$ = 1) within Upper Circumpolar Deep Water that followed the upwelling path of this layer. From all our results, we conclude a scenario of increased transport of deep waters into the section and enhanced upwelling at high latitudes for the period between 1995 and 2011, probably linked to a positive trend in the Southern Annular Mode. Although enhanced upwelling lowered the capacity of the AASW layer to uptake atmospheric $CO_2$, it did not limit that
of the newly forming SAMW and AAIW, which exhibited $C_{ANT}$ storage rates (~ 0.41 ± 0.20 mol m$^{-2}$ yr$^{-1}$) twice that of the upper layers.



## 1 Introduction.

The Southern Ocean is a key region in terms of climate change and climate variability, influencing the Meridional Overturning Circulation and therefore modulating the global circulation and oceanic biogeochemical cycles (Sarmiento et al., 1998, 2004;

Orr et al., 2005). Deep waters, formed in the North Atlantic, spread south and enter the Southern Ocean, where they mix with deep layers of the Antarctic Circumpolar Current (ACC) and ultimately upwell close to the Antarctic Shelf. The upwelled waters are eventually transformed into bottom, intermediate and mode waters, which are exported from the Southern Ocean to ventilate the thermocline and bottom layers of the major ocean basins. Some of the Southern Ocean waters subducted into the ocean interior return to the North Atlantic to balance the southward flux of North Atlantic Deep Water (Speer et al., 2000;

Lumpkin and Speer, 2007; Iudicone et al., 2008).

The eastward flow of the ACC is the primary mechanism of water exchange between the major ocean basins. The circumpolar path of the ACC consists of various narrow jets associated with sharp fronts that separate waters with different characteristics (Orsi et al., 1995; Belkin and Gordon, 1996). These jets can reach deep layers and often meander, intensify, merge and split, conditioned by the topography of the ocean floor, the stratification of the ACC, and atmospheric variability (Moore et al.,

1999; Sokolov and Rintoul, 2002, 2009; Peña-Molino et al., 2014). Movements in the jets enhance cross-stream transports and mesoscale activity that can result in local changes of water mass properties that can complicate the computation of long-term changes in water mass properties (Rintoul and Bullister, 1999; Sallée et al., 2008; Peña-Molino et al., 2014).

Water mass formation and ventilation transport heat, salt, and dissolved gases from the atmosphere to the ocean interior and other basins (Sarmiento et al., 2004), with the Southern Ocean contributing ~40% to the anthropogenic $CO_2$ ($C_{ANT}$) inventory

of the ocean (Khatiwala et al., 2009). Circulation and biological processes drive the redistribution of dissolved inorganic carbon (DIC) that ultimately affects the capacity of the waters to uptake more $CO_2$. The uptake of $CO_2$ by the Southern Ocean presents strong spatiotemporal variability (Lenton et al, 2013) and this can lead to conflicting results for observational studies, models and atmospheric inversions depending on the methodology used (Verdy et al., 2007; Lenton et al., 2012; Fay et al., 2014). Quantifying long-term changes in the carbon system is difficult due to the scarcity of data (Lenton et al., 2012; Kouketsu and

Murata, 2014; Fay et al., 2014) and the influence of biological processes. Notably, long term trends in $C_{ANT}$ concentration are difficult to estimate due to its small signal (~3%) with respect to that of DIC in the ocean.

Once $CO_2$ dissolves in the ocean (DIC) it begins the process of ocean acidification, i.e., decreases the pH and the saturation state of calcium carbonate ($CaCO_3$) minerals such as calcite and aragonite (Feely et al., 2004; Bates et al., 2014), with potential to disrupt ecosystems and biological processes (Doney et al., 2009).

Numerous studies have documented warming and freshening of deep and bottom layers of the Southern Ocean in recent decades (see reviews by Jacobs, 2006 and van Wijk and Rintoul, 2014). The abyssal waters of the Australian-Antarctic Basin (A-AB) show the greatest freshening in the last 40 years (van Wijk and Rintoul, 2014). This freshening was accompanied by




a warming of the deep-bottom layers, leading to a contraction in bottom waters by more than half their volume in the basin (Purkey and Johnson, 2012; van Wijk and Rintoul, 2014). Subsurface to intermediate layers have also warmed and freshened south of Australia (Bindoff and Church, 1992; Wong et al., 1999; Aoki et al., 2005). The solubility of gases and its distribution in the ocean depends on the dynamics and properties of water masses, and the thermohaline changes that have occurred south

of Tasmania (Fig. 1) have implications in the carbon system of the region.

A reduction of the carbon sink of the Southern Ocean was observed between the 1980s and the earlies 2000s (Le Quéré et al., 2007; Lovenduski et al., 2008), with more recent studies suggesting a recovery and even intensification of the $CO_2$ uptake by 2011-2012 (Zickfeld et al., 2008; Fay et al., 2014; Landschützer et al., 2015). DeVries et al. (2017), used a global inverse model to postulate that changes in circulation are responsible for most of the variability in the oceanic $CO_2$ uptake, with the

weakening of the upper-ocean circulation being responsible for the increase in oceanic carbon uptake over the past decade. Ocean acidification has been observed in the whole Southern Ocean (Lauvset et al., 2015) and locally in the Atlantic and Pacific sectors (Williams et al, 2015; Hauri, et al., 2015). South of Tasmania, McNeil et al., (2001) reported a $C_{ANT}$ uptake increase between 1968 and 1996 for the region 45-50°S. These authors reported, for the first time, $C_{ANT}$ accumulation in the AABW and highlighted the importance of the formation of bottom and mode waters as a mechanism for transporting $C_{ANT}$ to

the ocean. In terms of ocean acidification, we are not aware of any study about trends in ocean acidification in the water masses of the south of Australia.

Considering the lack of observational estimates for recent biogeochemical changes in the A-AB as well as the large changes in $CO_2$ uptake and storage suggested by recent atmospheric and surface observations in the Southern Ocean (e.g., Fay et al., 2014; Landschützer et al., 2015) there is a need to provide a full ocean depth observational perspective on how the ocean is

changing. The aim of this paper is to provide the first estimates of biogeochemical change in the water masses south of Tasmania, for the period 1995-2011, disentangling the effects that solubility, circulation, biology and $C_{ANT}$ uptake have on the variability of DIC. We use data from four summer repeats of the WOCE/JGOFS/CLIVAR/GO-SHIP hydrographic section SR03 (Fig. 1; Table 1), one of the most revisited in the Southern Ocean sections. Trends in oxygen ($O_2$), nutrients, and the carbon system parameters, i.e., DIC, total alkalinity (TA), anthropogenic carbon ($C_{ANT}$), total pH ($pH_T$) and % aragonite

saturation ($\Omega_{Ar}$) for the period 1995-2011, when both DIC and TA measurements are available. $C_{ANT}$ estimates were obtained with a back-calculation method (Pardo et al., 2014). The changes were evaluated in the different water mass layers of the section defined by neutral surfaces ($\Upsilon^n$, McDougall et al., 1987).

## 2 Hydrography of the region.

The dynamical structure of the region south of Tasmania (Fig. 1) is characterized by a number of fronts that separate the major

water masses of the region (Sokolov and Rintoul 2002, 2007, 2009). At the northern end of the section, the presence of the weak Subtropical Front (STF, Fig. 1) separates warm, salty subtropical waters from cooler and fresher sub-Antarctic waters (Deacon, 1937). The northern end of the section is a complex mixing zone where waters transported down the east coast of



Tasmania in a series of mesoscale eddies from the East Australian Current (EAC, Fig. 1) mix into the Subantarctic Zone, and also meet Zeehan Current (ZC, Fig. 1) waters transported down the west coast of Tasmania (Boland and Church 1981; Baines et al., 1983; Speich et al., 2002; Davis, 2005; Ridgway et al., 2007; Sloyan et al., 2016). The EAC transported south of Tasmania forms a zonal jet towards the southeast Indian Ocean known as the Tasman Outflow that reaches the bottom of the

Tasman slope and that is maintained all year round (Rintoul and Bullister, 1999; Ridgway et al., 2007). The encounter between these currents presents high variability at the northern end of the section (Fig. 1).

Farther south (Fig. 1), the Sub-Antarctic Front (SAF) and the Polar Front (PF) are regions of maximum transport in the ACC (Rintoul and Bullister, 1999; Sokolov and Rintoul 2002). North of the SAF, deep winter convection generates Sub-Antarctic Mode Water (SAMW), a relatively uniform water mass that occupies subsurface layers down to ~600m deep (McCartney,

1977; Rintoul and Bullister, 1999). SAMW constitutes the main part of the upper limb of the MOC, ventilating the thermocline of all the ocean basins (e.g., Speer et al., 2000; Sloyan and Rintoul, 2001). The SAF coincides with the deepening of the salinity minimum at intermediate depths (Whitworth and Nowlin, 1987), which is the signature of the Antarctic Intermediate Water (AAIW). The AAIW underlies the SAMW and also ventilates the global thermocline layers of the ocean. It is mainly formed in the southeast Pacific and it is continuously transformed on its way to the region south of Tasmania (Hanawa and Talley,

2001). South of the SAF, colder and fresher Antarctic Surface Water (AASW) covers the surface ocean. AASW originates from progressive warming of Winter Water (WW, Mosby, 1934), which can be perceived even in summer as a remnant layer of cold water at the base of the AASW (Rintoul et al. 1997).

The Southern ACC Front (SACCF, Fig. 1) is a deep front located south of the PF (Orsi et al., 1995) and can coincide with the southern boundary of the ACC, which is represented by the southern limit of the oxygen minimum (Orsi et al., 1995). The

oxygen minimum is related to the advection of Upper Circumpolar Deep Water (UCDW) that originates in the Indian and Pacific Oceans (Callahan, 1972) from where it spreads south, mixes with deep layers of the ACC and ultimately upwells near the Antarctic continent as part of the lower cell of the MOC. Lower Circumpolar Deep Water (LCDW) is below UCDW and is identified by a salinity maximum and contributes to upwelling around Antarctica. The precursor of the LCDW is North Atlantic Deep Water (NADW), originated in the Labrador and Nordic seas of the North Atlantic polar region (Dickson and

Brown, 1984) that flows southward to enter the ACC as part of the MOC (Callahan, 1972; Orsi et al., 1995; Johnson, 2008).

At the southern end of the section, the Antarctic Slope Front forms the boundary between cold and fresh shelf water and relatively warm and salty waters offshore (Jacobs, 1991). Polynyas along the Adélie and George V Land coast (Fig.1) contribute to the formation of Adélie Land Bottom Water (ALBW), which is a mixture of High Salinity Shelf Water (HSSW) resulting from brine rejection during ice formation and ultra-modified LCDW (Foster and Carmack, 1976; Rintoul, 1998;

Marsland et al., 2004). ALBW constitutes ~ 25% of the total volume of water <0°C in the ocean. The bottom waters near the southern end of the section also contain a component of Ross Sea Bottom Water (RSBW) that originates to the east and is deflected westwards towards the A-AB as it is transported down the continental slope. The RSBW is modified when it arrives at the location of the SR03 section by mixing with deep layers of the ACC and recently formed ALBW (Gordon and Tchernia,





1972; Rintoul, 1998). The ALBW and the modified RSBW together ventilate the abyssal layers of the A-AB before spreading north to ventilate the Indian and Pacific basins (Mantyla and Reid, 1995; Fukamachi et al. 2010).

## 3 Data and Method.

### 3.1 Data.

The SR03 hydrographic section between Tasmania and Antarctica (Fig. 1) was occupied between 1991 and 2011. Measurements of total alkalinity (TA) were only available from the beginning of 1995 and our evaluation of biogeochemical changes is limited to four summer sections occupied for the period 1995-2011 (Table 1). A winter cruise in 1996 was not considered in this study in order to minimise seasonal biases.

Water column salinity, temperature, pressure and dissolved oxygen ($O_2$) were collected from the conductivity-temperature-
depth (CTD) device with an accuracy of $\pm 0.002$ for salinity and temperature, $\pm 0.015$ % of full scale range for pressure and $\pm 1$ % for $O_2$, according to WOCE standards (Joyce, 1994). Samples from the Niskin bottles were analysed for dissolved inorganic carbon (DIC) by coulometry and TA by open cell potentiometric titration (Dickson et al, 2007). Certified reference material provided by A. Dickson, Scripps Institution of Oceanography, were used as reference standards for DIC and TA. The precisions of DIC and TA measurements improved slightly on more recent sections, and for all sections were $\pm 1.2$-1.5 and $\pm 1.5$-1.7 µmol
$kg^{-1}$, respectively, based on analysis of duplicate samples. Samples for dissolved $O_2$ were measured using modified Winkler titrations (Hood et al 2010), with an estimated accuracy and precision of $\pm 1\%$ for the sections. For the 1995-cruise, sensor based $O_2$ was used instead of sampled $O_2$ because of the poor quality of many of the Winkler measurements. The section data are available through the Global Ocean Data Analysis Project (http://cdiac.ornl.gov/oceans/GLODAPv2).

DIC and TA measurements allow estimates of other variables of the dissolved $CO_2$ system. We calculate $pH_T$ and $\Omega_{Ar}$ using
the CO2sys program from Lewis and Wallace (1998) adapted to MATLAB by van Heuven et al. (2011). We use the constants for the carbonic acid from Mehrbach et al (1973) refit by Dickson and Millero (1987), the $CO_2$ solubility equation from Weiss (1974), and dissociation constants for sulphate from Dickson (1990). Aragonite saturation states ($\Omega_{Ar}$) are calculated because it is a less stable form of $CaCO_3$ than calcite and is the predominant biogenic form of $CaCO_3$ precipitated by calcifying organisms.

Data presented here were interpolated to a regular grid, and the water mass layers were defined as layers between neutral surfaces ($\Upsilon^n$) determined using potential temperature and salinity and published literature values (Table 2, section 2). The first 50m of the water column were eliminated in order to reduce the short-time scale variability in surface properties. The use of $\Upsilon^n$ to identify the water mass layers reduces the variability due to isopycnal heave caused for example by eddies and internal waves (McDougall et al., 1987; Bindoff and McDougall, 1994; Jackett and McDougall, 1997). The upper ocean layers south
of the SAF were divided into the AASW layer to the north of the PF, and the $AASW_{upw}$ layer that is composed of surface waters south of the PF (Fig.1). ADLBW and RSBW were included in the AABW layer (Table 2). The ACC fronts along the SR03 section (Table 3) were defined as a function of hydrographic variables following Sokolov and Rintoul (2002).





### 3.2 Estimates of anthropogenic carbon (C$_{ANT}$).

C$_{ANT}$ was estimated using a back-calculation method (Chen and Millero, 1979; Gruber, 1998) with an Optimum Multi-Parameter (OMP) analysis (Tomczak, 1981), described by Pardo et al. (2014). This technique has the advantage of considering water mass mixing and the temporal variability of the air-sea CO$_2$ disequilibrium. The accuracy of the method is ±6 µmol kg$^{-1}$ (Pardo et al., 2014). Back-calculation methods assume the ocean is in steady state for dynamical and biological processes and estimate C$_{ANT}$ (C$_{ANT\_BC}$) as an excess of DIC in the ocean resulting from the increase of atmospheric CO$_2$ due to anthropogenic emissions as:

$$C_{ANT\_BC} = DIC - DIC^{BIO} - DIC^{\pi} \qquad (1)$$

where $DIC^{BIO}$ is the biological contribution to DIC, and $DIC^{\pi}$ is the concentration of DIC in preindustrial times.

The term $DIC^{BIO}$, is calculated (Chen et al., 1982; Ikegami and Kanamori, 1983) by:

$$DIC^{BIO} = \frac{AOU}{R_C} + \frac{1}{2}\left[TA - TA^0 + AOU * \left(\frac{1}{R_N} + \frac{1}{R_P}\right)\right] \qquad (2)$$

where $R_C$, $R_N$ and $R_P$ are the stoichiometric ratios of carbon, nitrate and phosphate respectively, referred to O$_2$ consumption by respiration/remineralization processes that are considered constant (1.45, 9, 125, respectively, Broecker, 1974; Anderson and Sarmiento, 1994; Martiny et al., 2013).

$\frac{AOU}{R_C}$ represents the remineralization of organic matter, with the apparent oxygen utilization (AOU) defined as (AOU = O$_{SAT}$ − O$_2$) the difference between the saturation of oxygen (O$_{SAT}$) at the potential temperature (θ) and salinity of the measured O$_2$. The term $\frac{1}{2}\left[TA - TA^0 + AOU * \left(\frac{1}{R_N} + \frac{1}{R_P}\right)\right]$, represents the dissolution/precipitation of CaCO$_3$, with $TA^0$, the preformed alkalinity, obtained in the water formation sites using regional parameterizations as function of salinity, θ and phosphate (Pardo et al., 2011; Vázquez-Rodríguez et al., 2012; Appendix section A3, Table A3).

The preindustrial term, $DIC^{\pi}$, is the total concentration of carbon dioxide in seawater saturated with respect to the preindustrial pCO$_2$ ($DIC_{SAT}^{\pi}$) and corrected for an air-sea CO$_2$ disequilibrium ($CDIS^{\pi}$) term:

$$DIC^{\pi} = DIC_{SAT}^{\pi} - CDIS^{\pi} \qquad (3)$$

$CDIS^{\pi}$ is time dependent:

$$CDIS^{\pi} = CDIS - \delta CDIS \qquad (4)$$

where $CDIS$ is the current disequilibrium between the ocean and the atmosphere pCO$_2$, and $\delta CDIS$ is the change in the disequilibrium from preindustrial to current times. $CDIS$ was obtained by similar parameterizations to those used for $TA^0$, combined with monthly mean values of atmospheric CO$_2$ values from the NOAA network (Dlugokencky, et al., 2016) (see section A3 of the appendix and Table A3). The $\delta CDIS$ values were obtained from results of the 1/10° resolution carbon model OFAM3-WOMBAT (Appendix section A1).

Interior values of preformed variables ($TA^0$ and $DIC_{SAT}^{\pi}$) and $CDIS^{\pi}$ were obtained using an optimum multiparameter (OMP) analysis to mix end members as described in Appendix sections A2 and A3, using the end members in Table A1. The OMP analysis is based on the assumption that a property measured in a certain point is the result of linear mixing between end





members, known as source water types (SWTs). A system of equations is created for each measurement point and is solved to obtain the fractions of the different SWTs (Appendix section A2). The application of the OMP analysis requires good regional hydrodynamic knowledge as the results are strongly dependant on the definition of the SWTs (Tomczak, 1981). We used 11 SWTs to characterize the biogeochemical properties of the waters in the SR03 section and the SWT properties were assumed

to be constant with time (Appendix Table A1, Fig. A1).

Small negative values of $C_{ANT\_BC}$ can occur due to an overestimation of the $CDIS^{\pi}$ term (Eq. 4), that acts as measure of the age of the water mass and has high values in old deep layers (see Table A1 in the appendix). These negative values of $C_{ANT\_BC}$ were found at some points in deep waters of the section (mainly UCDW and LCDW layers) and were small (between 0 and - 2 µmol kg$^{-1}$, i.e., less than the accuracy of the methodology) and changed to zero for our analysis.

**3.3 Changes in the carbon system.**

Changes in carbon system parameters ($\frac{\partial DIC}{\partial t}$, $\frac{\partial TA}{\partial t}$, $\frac{\partial C_{ANT\_BC}}{\partial t}$, $\frac{\partial pH}{\partial t}$ and $\frac{\partial \Omega_{Ar}}{\partial t}$) were estimated using linear regressions with time for the period 1995-2011 in each one of the water mass layers defined between $\Upsilon^{n}$ (Table 2). The trends were estimated using all the points in each water mass layer and only those linear trends with $p<0.05$ and $r>=0.2$ were considered statistically significant for discussion. We show the value of the root mean square error (RMSE or square root of the variance of the residuals), which

can be interpreted as the standard deviation of the unexplained variance, and has the same units as the response variable.

We also considered changes in the $DIC^{BIO}$ ($\frac{\partial DIC^{BIO}}{\partial t}$) and $DIC^{\pi}$ ($\frac{\partial DIC^{\pi}}{\partial t}$) terms of Eq. (1) (section 3.2). The quantities $\frac{\partial DIC^{BIO}}{\partial t}$ and $\frac{\partial DIC^{\pi}}{\partial t}$ provide information on the processes regulating the changes in $C_{ANT\_BC}$ and DIC and can be used to help interpret the $\frac{\partial C_{ANT\_BC}}{\partial t}$ estimates compared to the total observed changes in DIC, $\frac{\partial DIC}{\partial t}$.

The term $\frac{\partial DIC^{\pi}}{\partial t}$ can be expressed as:

$$\frac{\partial DIC^{\pi}}{\partial t} = \frac{\partial DIC_{SAT}^{\pi}}{\partial t} - \frac{\partial CDIS^{\pi}}{\partial t} \qquad (5)$$

The terms $\frac{\partial CDIS^{\pi}}{\partial t}$ and $\frac{\partial DIC_{SAT}^{\pi}}{\partial t}$ reflect changes in the properties of the water masses over time, primarily temperature and salinity change due to mixing and heating/cooling. $CDIS^{\pi}$ and $DIC_{SAT}^{\pi}$ are defined at the ocean surface (in each of the SWTs: Table A1) and are calculated at each point in the ocean interior using the OMP analysis (Appendix sections A2 and A3). Because a change in temperature and/or salinity in the water is solved by the OMP analysis as a change in the SWTs fractions, this also

produces varying $CDIS^{\pi}$ and $DIC_{SAT}^{\pi}$. No significant trends were obtained for $CDIS^{\pi}$ in any of the layers. $C_{ANT\_BC}$ (Eq. 1) is not affected by the changes in solubility occurring from one voyage to another (thus neither is $\frac{\partial C_{ANT\_BC}}{\partial t}$), since any change in temperature or salinity is cancelled out by the subtraction of $DIC^{\pi}$ with respect to DIC in Eq. (1) (section 3.2) and by $O_2$ respect to $O_{SAT}$ in the $DIC^{BIO}$ term (Eq. 2). However, $\frac{\partial DIC}{\partial t}$ based on the measured DIC in the sections will be affected by





changes in the solubility over time and this difference needs to be accounted for when comparing $\frac{\partial DIC}{\partial t}$ with $\frac{\partial C_{ANT\_BC}}{\partial t}$ to obtain a better approximation for the change in $C_{ANT}$: $\frac{\partial C_{ANT}}{\partial t} = \frac{\partial C_{ANT\_BC}}{\partial t} + \frac{\partial DIC^{\pi}}{\partial t}$.

The term $\frac{\partial DIC^{BIO}}{\partial t}$ can be influenced by changes with time of alkalinity due to changes in the rate of carbonate precipitation/dissolution and of AOU due to changes in the rate of remineralization and in circulation. Numerous studies have

reported a strong influence of biological communities in the seasonal cycle of dissolved $O_2$ in surface waters (Bender et al., 1996; Moore and Abbott, 2000; Sambrotto and Mace, 2000; Trull et al., 2001a). Interannual variability in $O_2$ in upper layers of the Southern Ocean have also been related to changes in the entrainment of deeper waters into the mixed layer due to the mixed layer depth variability (Matear et al., 2000; Verdy et al., 2007; Sabine et al., 2008; Sallée et al., 2012). Nevertheless, long-term trends in $O_2$ due to circulation and remineralization processes have not yet been reported. Thus, the term $\frac{\partial DIC^{BIO}}{\partial t}$

may also contribute to variation in $\frac{\partial C_{ANT\_BC}}{\partial t}$, since part of the changes in AOU with time reflect changes in circulation that we cannot separate from those in remineralization. We consider the best approximation for the change in $C_{ANT}$ as a range depending on the possible effect of biology and circulation processes on $\frac{\partial DIC^{BIO}}{\partial t}$. If the value of $\frac{\partial DIC^{BIO}}{\partial t}$ is due to the variability in the remineralization rates and the change in solubility is considered, the estimate $\frac{\partial C_{ANT\_BC}}{\partial t}$ will be the lower limit of the range, (lower limit of $\frac{\partial C_{ANT}}{\partial t} = \frac{\partial C_{ANT\_BC}}{\partial t} + \frac{\partial DIC^{\pi}}{\partial t}$). For the upper limit of the range, we consider that the value of $\frac{\partial DIC^{BIO}}{\partial t}$ is

due to changes in circulation and the upper limit of the range is obtained by $\frac{\partial C_{ANT}}{\partial t} = \frac{\partial C_{ANT\_BC}}{\partial t} + \frac{\partial DIC^{\pi}}{\partial t} + \frac{\partial DIC^{BIO}}{\partial t}$. We assume that the changes in $DIC^{BIO}$ due to circulation do not affect the amount of DIC in the layer. This assumption is one of the caveats of the methodology, since we cannot know how much of the change in DIC is associated to changes in circulation, i.e., how much of the change in DIC is a change in non-anthropogenic DIC. We will discuss this more in section 4.4.2.

### 4 Results.

The different trends in biogeochemical properties are summarized in Table 4. The he biogeochemical changes between 1995 and 2011 are presented for each of the water mass layers and the effect of changes in solubility, biological processes and circulation in the estimates of $\frac{\partial C_{ANT}}{\partial t}$ and $\frac{\partial DIC}{\partial t}$ are considered along with changes in the aragonite saturation depth and $C_{ANT}$ storage.





### 4.1 Changes in DIC, $C_{ANT}$ and $pH_T$ (1995-2011).

### 4.1.1 Upper ocean layers (STCW, AASW and $AASW_{upw}$).

In the STCW layer, DIC increased between 1995 and 2011 (Fig. 2a, b) at a rate of $0.86 \pm 0.07$ µmol kg$^{-1}$ yr$^{-1}$ ($\frac{\partial DIC}{\partial t}$, Table 4),

leading to a decrease of $pH_T$ of $-0.0027 \pm 0.0001$ yr$^{-1}$ ($\frac{\partial pH_T}{\partial t}$, Table 4, Fig. 2e, f). The trend in $pH_T$ is similar to the one found

by Lauvset et al. (2015) between 1991 and 2011 for the IO-STPS (Indian Ocean subtropical permanently stratified) biome (-

$0.0027 \pm 0.0005$ yr$^{-1}$). We found a decrease of $DIC^{\pi}$ ($\frac{\partial DIC^{\pi}}{\partial t}$ = $-0.34 \pm 0.06$ µmol kg$^{-1}$ yr$^{-1}$, Table 5) in the STCW layer due to a

negative trend of $DIC^{\pi}_{SAT}$ resulting from a decrease in solubility that resulted from a temperature increase (calculated from the

section data) in the STCW layer of $0.0335 \pm 0.0130$ °C yr$^{-1}$ (not shown). The increase in temperature agrees with the warming

trend observed south of Tasmania of 0.2 to 0.3 °C decade$^{-1}$ obtained from satellite data (Armour and Bitz, 2015) and from

combined data and models (0.5 °C / 30 yr$^{-1}$, Aoki et al., 2015). For $\theta$=16 °C and S=35.1 (definition of $SWT_{STW16}$ in the OMP

analysis, Table A1), a change in temperature of 0.03 °C yr$^{-1}$ would lead to a decrease in $DIC^{\pi}_{SAT}$ of -0.27 µmol kg$^{-1}$ yr$^{-1}$, which

is similar to the value obtained for $\frac{\partial DIC^{\pi}}{\partial t}$ in the STCW layer (Table 5). The difference between these trends is related to the

mixing of the different SWTs fractions within the STCW layer established from the OMP analysis (see section 3 and section

A2 of the Appendix). When the solubility change is incorporated into $\frac{\partial C_{ANT\_BC}}{\partial t}$, i.e. $\frac{\partial C_{ANT\_BC}}{\partial t} + \frac{\partial DIC^{\pi}}{\partial t}$ (Table 5, Fig. 2c, d), we

obtain a value of $0.71 \pm 0.08$ µmol kg$^{-1}$ yr$^{-1}$.

There is an increase of $DIC^{BIO}$ in the STCW layer ($\frac{\partial DIC^{BIO}}{\partial t}$, Table 5), that also affects the estimates of $\frac{\partial C_{ANT\_BC}}{\partial t}$. The increase

of $DIC^{BIO}$ is due to an increase in AOU (no changes were found in TA, Eq. (2) in section 3.2) due to a decrease of $O_2$ in the

layer. We cannot separate the effects of circulation and biology on the AOU change and $\frac{\partial C_{ANT}}{\partial t}$ in Table 4 should be considered

a range. If the changes in AOU are only due to the variability in the remineralization rates, the calculated lower limit of $\frac{\partial C_{ANT}}{\partial t}$

is $0.71 \pm 0.08$ µmol kg$^{-1}$ yr$^{-1}$ (Table 4, $\frac{\partial C_{ANT}}{\partial t} = \frac{\partial C_{ANT\_BC}}{\partial t} + \frac{\partial DIC^{\pi}}{\partial t}$ in Table 5). If the changes in AOU are due to changes in

circulation, the upper limit value of $0.93 \pm 0.11$ µmol kg$^{-1}$ yr$^{-1}$ ( $\frac{\partial C_{ANT}}{\partial t} = \frac{\partial C_{ANT\_BC}}{\partial t} + \frac{\partial DIC^{\pi}}{\partial t} + \frac{\partial DIC^{BIO}}{\partial t}$ ) will explain the

increase of DIC in the STCW layer ($\frac{\partial DIC}{\partial t} \approx \frac{\partial C_{ANT}}{\partial t}$, Table 4). The increase of $C_{ANT}$ found in this layer is comparable to the

range of increase (0.8 – 1.3 µmol kg$^{-1}$ yr$^{-1}$) found by Carter et al. (2017) in the Pacific Ocean (P16 WOCE, CLIVAR and

GOSHIP lines) for the two past decades (1990s-2000s and 2000s-2010s).

Changes in the AASW layer are summarised in Table 4. DIC increased at a similar rate of $0.85 \pm 0.14$ µmol kg$^{-1}$ yr$^{-1}$ to the

STCW layer and the trend is similar to the values found by Williams et al. (2015) for the AASW layer in the Pacific sector of

the SO (12-18 µmol kg$^{-1}$ for the period 1992-2011 and 3-5 µmol kg$^{-1}$ for the period 2005-2011). The increase of DIC in the

AASW layer results in a $pH_T$ decrease of $-0.0035 \pm 0.0002$ yr$^{-1}$, close to Williams et al. (2015) estimates for surface waters (~



-0.0023 ± 0.0009 yr$^{-1}$) and Lauvset et al. (2015) estimates of -0.0021 ± 0.0002 yr$^{-1}$ for the Southern Ocean seasonally stratified, SO-SPSS, biome. The AASW layer for our sections warmed at a similar rate (0.0369 ± 0.0109 °C yr$^{-1}$) to the STCW layer, reducing the solubility and influencing $\frac{\partial DIC^{\pi}}{\partial t}$ (Table 5) due to changes in $DIC^{\pi}_{SAT}$. The $DIC^{BIO}$ also increased with time (0.50 ± 0.16 µmol kg$^{-1}$ yr$^{-1}$, Table 5) due to an increase of AOU. Following the same reasoning as for the STCW layer and considering

the trend in $C_{ANT}$ of 0.70 ± 0.06 µmol kg$^{-1}$ yr$^{-1}$ obtained by the back-calculation method ($\frac{\partial C_{ANT\_BC}}{\partial t}$ ; Table 5), the best estimation of $\frac{\partial C_{ANT}}{\partial t}$ in the AASW layer is a range of 0.35 ± 0.14 to 0.85 ± 0.22 µmol kg$^{-1}$ yr$^{-1}$ (Table 4). Our values are within the range of values found by Williams et al. (2015) for the AASW in the Pacific sector between 2005 and 2011 and the upper limit is similar to a $C_{ANT}$ increase of 0.73 - 0.86 µmol kg$^{-1}$ yr$^{-1}$ for waters South of Tasmania for the period 1968-1996 found by McNeil et al. (2001).

DIC in the AASW$_{upw}$ layer increased at a rate of 0.61 ± 0.10 µmol kg$^{-1}$ yr$^{-1}$, and the pH$_T$ decreased -0.0015 ± 0.0004 yr$^{-1}$ (Table 4). We were not able to detect a statistically significant trend in $DIC^{\pi}$ (i.e., solubility) or $C_{ANT}$ from the estimates of the back-calculation method ($\frac{\partial C_{ANT\_BC}}{\partial t}$ Table 5). However, we found an increase of $DIC^{BIO}$ of 0.42 ± 0.28 µmol kg$^{-1}$ yr$^{-1}$ (Table 5) that is due to an increase in AOU. Considering the different drivers of the AOU increase (biology/circulation), the optimal estimation of $\frac{\partial C_{ANT}}{\partial t}$ for this layer is a value between 0 and 0.42 ± 0.28 µmol kg$^{-1}$ yr$^{-1}$ (Table 4).

**4.1.2 Mode waters and intermediate layers (SAMW and AAIW).**

The increase in DIC in the SAMW layer (1.10 ± 0.14 µmol kg$^{-1}$ yr$^{-1}$) for the period 1995-2011 is higher than that of upper ocean layers  and pH$_T$ decreases over the same period at -0.0031 ± 0.0003 yr$^{-1}$ (Table 4). The DIC increase is explained almost entirely by $\frac{\partial C_{ANT\_BC}}{\partial t}$ of 0.92 ± 0.09 µmol kg$^{-1}$ yr$^{-1}$. No significant trend was found in $DIC^{BIO}$ or $DIC^{\pi}$ (i.e, $\frac{\partial C_{ANT}}{\partial t} = \frac{\partial C_{ANT\_BC}}{\partial t}$). In the AAIW layer the DIC trend of 0.40 ± 0.15 µmol kg$^{-1}$ yr$^{-1}$ results in a pH$_T$ decrease of -0.0017 ± 0.0002 yr$^{-1}$ and is also

explained by the increase of $C_{ANT}$ (0.42 ± 0.06 µmol kg$^{-1}$ yr$^{-1}$, $\frac{\partial C_{ANT}}{\partial t} = \frac{\partial C_{ANT\_BC}}{\partial t}$, Tables 4 and 5). As with SAMW, no changes in solubility ($\frac{\partial DIC^{\pi}}{\partial t}$) or biology/circulation processes ($\frac{\partial DIC^{BIO}}{\partial t}$) were detected in the AAIW layer. The values found in the SAMW and AAIW layers are very similar to the mean decadal changes found by Murata et al. (2007) between the 1990s and the 2000s in the subtropical Pacific Ocean (~1 µmol kg$^{-1}$ yr$^{-1}$ for the SAMW layer and 0.4 µmol kg$^{-1}$ yr$^{-1}$ for the AAIW). Waters et al., (2011) used data from the P18 line along ~110°W and estimated an increase in $C_{ANT}$ of 0.89 ± 0.4 µmol kg$^{-1}$ yr$^{-}$

$^1$ for the SAMW and 0.64 ± 0.2 µmol kg$^{-1}$ yr$^{-1}$ in AAIW for the period 1994-2008, which are also comparable to our results.

**4.1.3 Deep-bottom layers (UCDW, LCDW and AABW).**

The UCDW layer shows an increase of DIC of 0.29 ± 0.02 µmol kg$^{-1}$ yr$^{-1}$ between 1995 and 2011 and a change in pH$_T$ of -0.0013± 0.0001 yr$^{-1}$ and are similar to the change in DIC (0.20 ± 0.02 µmol kg$^{-1}$ yr$^{-1}$) and pH$_T$ (-0.0012± 0.0002 yr$^{-1}$) for



LCDW. No statistically significant changes in time were detected of $C_{ANT\_BC}$ or in the $DIC^{BIO}$ and $DIC^{\pi}$ terms for any of these two layers.

The AABW layer also shows an increase of DIC ($0.24 \pm 0.02$ µmol kg$^{-1}$ yr$^{-1}$) during the period 1995-2011 with an associated decrease of pH$_T$ of $-0.0013 \pm 0.0002$ yr$^{-1}$ (Table 4). The increase in $C_{ANT}$ ($\frac{\partial C_{ANT\_BC}}{\partial t}$) of $0.07 \pm 0.01$ µmol kg$^{-1}$ yr$^{-1}$ is low and

this trend indicates an increase in $C_{ANT}$ of ~ 1 µmol kg$^{-1}$, which is less than the accuracy of the back-calculation method ($\pm 6$ µmol kg$^{-1}$).

### 4.2 Changes in the aragonite saturation ($\frac{\partial \Omega_{Ar}}{\partial t}$) and $C_{ANT}$ storage.

There are statistically significant decreases of $\Omega_{Ar}$ in the STCW, AASW and SAMW layers (~ $-0.010 \pm 0.001$ yr$^{-1}$, Table 4) similar to the trends observed at open-ocean time series sites in recent decades (Bates et al., 2014). The decrease of $\Omega_{Ar}$ found

for the AASW layer ($-0.61 \pm 0.19$ % yr$^{-1}$, Table 4) is also similar to the values obtained by Williams et al. (2015) for the Pacific sector of the Southern Ocean ($-0.47 \pm 0.10$ % yr$^{-1}$ for the period 1992 -2011 and $-0.50 \pm 0.20$ % yr$^{-1}$ for the period 2005-2011). Accompanying the decrease of $\Omega_{Ar}$ with time along SR03, is the shoaling of the aragonite saturation depth (ASD, $\Omega_{Ar} = 1$, Fig. 3) at a mean rate of $-13 \pm 3$ m yr$^{-1}$. The shoaling of the ASD is not uniform over the section. North of the PF, the ASD shoals at a rate of $-6 \pm 4$ m yr$^{-1}$ while the rate is 3.5 times greater south of the PF ($-21 \pm 4$ m yr$^{-1}$). North of the PF the shoaling mostly

affects the AAIW layer (Fig. 3a). South of the PF from ~62°S, the movement of the ASD follows the upwelling path of the UCDW layer (Fig. 3) with a shoaling of ~340 m over the 16-year period.

The storage rate of $C_{ANT}$ (Table 6) for the surface and intermediate water mass layers is obtained from $\frac{\partial C_{ANT}}{\partial t}$ (Table 4) with the most storage in SAMW and AAIW due to both their greater thickness and $\frac{\partial C_{ANT}}{\partial t}$ values. The rate of increase of the $C_{ANT}$ storage in the whole longitude band of the SR03 section is $0.30 \pm 0.24$ mol m$^{-2}$ yr$^{-1}$, calculated by computing the mean of the

storage rates of the layers weighted by the mean volume occupied by each of the layers for the period 1995-2011 (Table 6)

### 5 Discussion.

Our results are indicative of a scenario of increased transport of deep waters into the section and enhanced upwelling at high latitudes for the period between 1995 and 2011, probably linked to a positive trend in the Southern Annular Mode (SAM). Several studies have reported a trend in the SAM toward its positive phase from the 1960s until the 2000s (Thompson and

Solomon, 2002; Marshall, 2002, 2003; Lenton and Matear, 2007; Sallée et al., 2008). According to these studies, the positive phase of the SAM is correlated with an intensification and southward movement of the subpolar westerly winds that ultimately lead to the enhancement of northward Ekman transport, meridional overturning and upwelling south of the ACC. Also, surface warming and more intense and frequent pulses in the extension of the EAC at long-time scales have been related to a poleward movement of the westerly winds (Rintoul and Sokolov, 2001; Ridgway, 2007; Hill et al., 2011).



In the northern part of the SR03 section, the area occupied by the STCW has high variability due to the encounter between the EAC and the ZC in the North of the section (Ridgway et al., 2007; Herraiz-Borreguero and Rintoul, 2011; Sloyan et al., 2016). The warming of the STCW layer found in this study ($0.0335 \pm 0.0130$ °C yr$^{-1}$) could be linked to variability in the extension of subtropical waters but it could also be related to atmospheric warming. Aoki et al. (2015) related the 30-year warming found north of the SAF in the South Pacific and Indian oceans to the intensification of the subtropical gyres, which promote the arrival of warmer waters. In the AASW layer that extends approximately between the SAF and the PF we found a similar warming ($0.0369 \pm 0.0109$ °C yr$^{-1}$) to that of the STCW. This could indicate that the increase of temperature found in the upper layers of the section could be most likely due to ocean heat uptake and atmosphere warming.

Due to the surface warming, the increase of DIC found in the STCW layer (Table 4) is lower than expected from the increase in atmospheric $CO_2$ ($\sim 1 \pm 0.12$ μmol kg$^{-1}$ yr$^{-1}$). Nevertheless, at least 83% of the increase of DIC in the STCW layer is explained by the increase in $C_{ANT}$ (Table 4). As for the AASW layer, our results indicate that temperature does affect the estimate of $\frac{\partial DIC}{\partial t}$, but the effect of the increase in $DIC^{BIO}$ (due to an increase in AOU) overweigh that of solubility (Table 5).

The seasonal to interannual variability of the AASW layer is also influenced by the variability of the positions of the SAF and PF (Fig. 1, Table 3), that is highly conditioned by the flow of the ACC over the South-East Indian Ridge (Fig. 1). A close relationship between phytoplankton blooms and regions where the ACC fronts interact with large topographic features has been noticed (Moore et al., 1999; Moore and Abbott, 2000). A variability in the remineralization rates due to phytoplankton blooms variability could explain the changes in $DIC^{BIO}$ observed in the AASW layer. Nevertheless, no changes in nutrients (nitrates or phosphates) are measurable in this layer that could indicate intense biological activity.

Furthermore, the AASW layer is also affected by the upwelling of deep waters south of the PF, and an intensification of the upwelling could increase the content of low-$O_2$ DIC-rich waters in the AASW layer leading to an increase in AOU. The increase in $DIC^{BIO}$ found in the AASW$_{upw}$ layer (Table 4), south of the PF, is similar to the increase obtained for the AASW layer, which indicates the likelihood that the upwelling of deep waters results in the increase in AOU. The increase in $DIC^{BIO}$ in the AASW$_{upw}$ layer coincides with an increase of salinity of $0.0029 \pm 0.0001$ yr$^{-1}$ (not shown), that is consistent with increased transport of saltier waters from the deep ocean to subsurface layers. Besides, we also found an increase in dissolved silicate of $0.36 \pm 0.06$ μmol kg$^{-1}$ yr$^{-1}$ ($\frac{\partial SiO_4}{\partial t}$ Table 4) that could be related to the upwelling enhancement as well (Tréguer, 2014). The influence of the upwelling on the DIC budgets (as non-anthropogenic DIC) is clearer in the AASW$_{upw}$ layer than in the AASW layer. For the AASW layer, the lower limit of $\frac{\partial C_{ANT}}{\partial t}$ (i.e., the change in $DIC^{BIO}$ is assumed to be due to biological processes, Table 4) indicates that at least 41% of the increase of DIC in the layer is explained by the increase of $C_{ANT}$ while the lower limit of $\frac{\partial C_{ANT}}{\partial t}$ is zero for AASW$_{upw}$ (Table 4), meaning that the effect of the upwelling over AASW is lower than over AASW$_{upw}$. Matear and Lenton (2008) using carbon models, concluded that the uptake of CO2 by the waters north of the PF is more influenced by the wind variability than by other processes such as the upwelling. An intensification of the winds (due to a positive phase in SAM) could contribute to the increase in $C_{ANT}$ found in the AASW layer. Considering the upper



limits of $\frac{\partial C_{ANT}}{\partial t}$ in both layers (i.e., the change in $DIC^{BIO}$ is assumed to be due to circulation processes), the increase of $C_{ANT}$ in the AASW$_{upw}$ layer represents no more than 69% of the increase in DIC (upper limit of $\frac{\partial C_{ANT}}{\partial t}$, Table 4) while the upper limit of $\frac{\partial C_{ANT}}{\partial t}$ for the AASW equals the increase of DIC. Thus, AASW$_{upw}$ layer, at least ~30% of the increase in DIC (~ 0.18 μmol kg$^{-1}$ yr$^{-1}$) is still not explained and is most probably related to the upwelling of DIC-rich waters. The increase in non-

anthropogenic DIC could be even higher, since we assume that the change in $DIC^{BIO}$ due to circulation does not affect DIC (see section 3.3).

The variability of the SAMW and AAIW layers south of Tasmania has been related to variability in the northward Ekman transport that drives the northward movement of AASW (Rintoul and England, 2002; Sallée et al., 2006, 2012). A scenario of intensification of the upwelling near the Antarctic Divergence would lead to an increase in the northward Ekman transport,

conditioning the properties of these water mass layers and particularly for SAMW, which is mostly form north of the SAF. There is a significant freshening of the SAMW layer (-0.0026 ± 0.0001 psu yr$^{-1}$, not shown) between 1995 and 2011 that could be related to higher inputs of AASW into the SAMW layer and consistent with the increase in Ekman transport. Besides, an intensification of the winds due to the positive trend of the SAM favours the ventilation and thus the increase in $C_{ANT}$ uptake by both water mass layers (Matear and Lenton, 2008). Our results indicate that the change of DIC in the SAMW and AAIW

layers is driven mostly by the uptake of atmospheric $CO_2$ ($\frac{\partial DIC}{\partial t} \approx \frac{\partial C_{ANT}}{\partial t}$, Table 4). The increase of $C_{ANT}$ in the SAMW layer is higher than that found for the upper ocean layers and closer to the expected from the increase of atmospheric $CO_2$ (~1 μmol kg$^{-1}$ yr$^{-1}$). The smaller increase of $C_{ANT}$ in the AAIW layer compared to the SAMW layer (Table 4) agrees with lower ventilation of the AAIW layer south of Tasmania (see section 2) due to the fact that this layer carries recently ventilated waters mixed with older waters ventilated far out the SR03 section. The lack of measurable long-term changes in $DIC^{BIO}$ and $DIC^{\pi}$ in both

AAIW and SAMW layers indicate that circulation and biological processes do not have a large effect on $\frac{\partial DIC}{\partial t}$.

Deep to bottom layers of the section show significant trends for DIC that are not explained by the increase of $C_{ANT}$. These trends are most likely due to the advection of old and DIC-rich waters. Concretely for deep waters (UCDW and LCDW), the trends could result from an intensification of upwelling at high latitudes being offset by enhanced transport of old and $CO_2$-rich waters to replace the upwelled waters, since the increase of DIC follows the upwelling path of the UCDW and LCDW

layers (Fig. 4). We separated the UCDW layer into two latitudinal sectors: north and south of the SAF (Fig. 4). The increase of DIC in the UCDW layer north of the SAF is 0.44 ± 0.04 μmol kg$^{-1}$ yr$^{-1}$ while south of the SAF is smaller at 0.26 ± 0.04 μmol kg$^{-1}$ yr$^{-1}$ (not shown), consistent with a greater supply of waters from the north at depth. A decrease of $O_2$ in the UCDW to the north of the SAF occurs mostly in the upper to middle parts of the UCDW layer (Fig. 4), and this is not observed south of the SAF. The decrease of $O_2$ is also in agreement with the arrival of waters from Indian-Pacific origin since these waters

provide the characteristic oxygen minimum zone that defines UCDW (Callahan, 1972; Talley 2013). Another feature that agrees with the hypothesis of upwelling intensification is the shoaling of the ASD following the path of upwelling of the UCDW layer. This feature was also described by Bostock et al. (2013) in an oceanic climatology of $\Omega_{Ar}$ and could be due to



the naturally lower buffer capacity of the UCDW layer (low value of TA/DIC ≈ 1.043) with respect to upper layers (TA/DIC ≈ 1.06 in the AASW layer). However, the greatest shoaling of the ASD in the UCDW layers compared to the AAIW layer (Table 6) is consistent with the upwelling of UCDW, as both water masses have similar TA/DIC ratios (TA/DIC ≈ 1.043 for the UCDW layer and TA/DIC ≈ 1.042 for the AAIW layer). Furthermore, the increase of $SiO_4$ found in deep-bottom layers

(Table 4) could also indicate the arrival of old waters to the section that are progressively enriched in $SiO_4$ (e.g., Callahan, 1972).

Statistically significant decreases in $pH_T$ with time were observed in all water mass layers (Table 4), with the greatest change in surface water masses, coinciding with the greatest DIC changes. The decrease of $pH_T$ in the STCW and SAMW layers is related to the increase in the uptake of $C_{ANT,}$ while for the AASW and $AASW_{upw}$ layers the $pH_T$ change appears to be linked

to the upwelling of DIC-rich waters at high latitudes. At deep layers, the tongue of water of $pH_T$ =7.9 off the shelf is reduced in 2011 compared to 1995 (Fig. 2e,f), which is consistent with the advection of DIC-rich waters in the section due to the enhanced upwelling. The different rates of $pH_T$ change in the water masses is in part related to the buffering capacity of the waters. AASW layer has lower temperature than the STCW layer (~2.3 °C for the AASW layer compared to ~11.0 °C for the STCW layer, mean values for the period 1995-2011) and lower buffer capacity than the STCW (TA/DIC ≈ 1.058 for the

AASW layer versus TA/DIC ≈ 1.095 for STCW). For similar increases in DIC (Table 4) the decrease of $pH_T$ in the AASW is expected to be higher than in the STCW layer.

In terms of carbon, previous studies concluded that the intensification of the upwelling (as a consequence of the SAM variability) caused a reduction in the uptake of $CO_2$ by the Southern Ocean between 1980s and 2000s due to the outgassing of $CO_2$ near the Antarctic Divergence (Le Quére et al, 2007; Lovenduski et al., 2008). Landschützer et al. (2015) showed that the

efficiency of the Southern Ocean $CO_2$ sink declined through the 1990's, and the trend reversed from about 2002, although the reversal in the sink efficiency was not zonally uniform. The results from Landschützer et al. (2015) are consistent with a carbon sink influenced by the upwelling of DIC-rich waters at high latitudes and superimposed on this is the near surface response to atmospheric forcing that modifies the sink efficiency and could mask longer term trends in the upwelling of DIC-rich waters at high latitudes. A comparison of our results with those of Landschützer et al. (2015) is problematic as their data is restricted

to surface waters and our analysis is on long-term trends in water mass properties below 50m depth. Both data sets do show continued uptake of $CO_2$ throughout the period of study and indicate the importance of the circulation in influencing the regional carbon sink, which has also been established by recent model results (DeVries et al., 2007). Our results also agree with the conclusions from different model simulations done by Matear and Lenton (2008), who established that intense wind regimes (associated to a positive phase in the SAM) favour the uptake of $C_{ANT}$ and ventilation of the SAMW and AAIW layers.

These authors highlighted the complex response of the uptake of $CO_2$ by the Southern Ocean due to the diverse forcing acting on upper layers, which can be also seen in our results (e.g., differences in the biogeochemical changes in the AASW and $AASW_{upw}$ layers). Matear and Lenton (2008) also noticed the complex relationship between the upwelling and subduction areas of the Southern Ocean, with the same drivers acting in opposite direction for the changes in non-anthropogenic DIC with respect to the changes in $C_{ANT}$ uptake.




Our results rely on a limited number of sections spread every 3-7 years and can only provide a long term (decadal) average view of changes in water masses. More surface observations and repeat deep ocean sections are needed to help resolve interannual changes in the Southern Ocean carbon sink and to determine the main drivers and feedback to the carbon-climate system. The effort to maintain hydrographic sections with $CO_2$ system measurements would also benefit from additional direct
measurements of more variables of the carbon system, e.g. pH, which has not been measured on the SR03 section.

## 6 Sensitivity of the results to underlying assumptions.

This section considers the sensitivity of assumptions used to calculate temporal changes in $C_{ANT}$, including errors associated with the assumption of steady state in the oceanic circulation and remineralization processes, and the sensitivity to stoichiometric ratios for the biological processes.

**6.1 Comparison of $C_{ANT}$ changes using other methods.**

We compared the changes of $C_{ANT}$ obtained in our study with the results from two regression-based methodologies (Table 7); the extended multiple linear regression (eMLR) method (Friis et al., 2005) and the two-regression method (Thacker, 2012). These methods use repeat hydrodynamic repeat sections to quantify the temporal change in $C_{ANT}$.

The eMLR method (Friis et al., 2005) estimates the change in $C_{ANT}$ between two repeats of a hydrodynamic section by
establishing MLRs for each section and relating the observed DIC for each observation to a set of other measured oceanic variables:

$$DIC_{(t)} = a_{0(t)} + a_{1(t)}P_{1(t)} + \cdots + a_{n(t)}P_{n(t)} \qquad (6)$$

where $a_{x(t)}$ are the coefficients of the fit between DIC and the n observed variables ($P_1,\ldots P_n$) chosen for the fit, all measured at the time (t) of the survey.

Taking the difference between DIC at two times, t1 and t2, gives an equation for the change in $C_{ANT}$ over the time period between the two hydrographic surveys ($\Delta C_{ANT}$):

$$\Delta C_{ANT} = a_{0(t2)} - a_{0(t1)} + (a_{1(t2)} - a_{0(t1)})P_{1(t2)} + \cdots + (a_{n(t2)} - a_{0(t1)})P_{n(t2)} \qquad (7)$$

The two-regression method was introduced by Thacker (2012) as an improvement in regression-based methods. The region of study is first divided into sub-regions since the empirical relationships between DIC and other environmental variables vary
spatially (Thacker, 2012). MLRs are investigated between DIC and other measured variables (predictors) using a stepwise technique. The procedure is applied for each sub-region using all data from the repeat surveys within the period to be investigated, resulting in an optimal MLR for each sub-region (similar to Eq. 6). A linear regression with time is established for the residuals (observed DIC - predicted DIC) of the regional fits, which directly gives $\Delta C_{ANT}$ averaged over the space-time in each sub-region. The purpose of the first MLR is to remove the natural variability of DIC, leaving the anthropogenic signal
and noise (random variability) in the residuals and the second MLR is used to separate the anthropogenic signal from the noise.



We applied both methodologies within the different water mass layers separated by $\Upsilon^n$ used as sub-regions. The predictor variables of $\theta$, S, $\sigma_0$, nitrate ($NO_3$), $SiO_4$ and AOU were used for the MLR procedures. The three methodologies estimate similar rates of increase in $C_{ANT}$ for most water mass layers (Table 7). In the STCW layer, the value of $\Delta C_{ANT}$ (eMLR) is higher than our maximum estimate of $\frac{\partial C_{ANT}}{\partial t}$ and the value obtained from the two-regression method. The eMLR method is less

suitable for the upper layers of the ocean subject to high seasonal to interannual variability, such as the STCW layer, resulting in large residuals that bias the regression (Friis et al., 2005). For the AAIW layer, the increase of $C_{ANT}$ estimated by the two-regression method is half the increase that is established by our method and the eMLR method. The lower value of the trend estimated by the two-regression method is due to the fact that the two-regression method uses and stepwise MLR. This means that the two-regression method only considers those predictors that give the best fit while the eMLR method is forced to

consider all the predictors in the fit. This is also the cause of the low RMSE obtained with the two-regression method compared to both our trends and those obtained by the eMLR method. For deep to bottom layers, the two-regression method estimates a small increase of $C_{ANT}$ similar to the one found in this study for the AABW layer (Table 7) that can be considered negligible given the resolution of the back-calculation method ($\pm 6\,\mu mol\,kg^{-1}$). The eMLR method finds increases of $C_{ANT}$ (with relatively high uncertainties) higher than the two-regression method that over the 16-year period also give values of $C_{ANT}$ change lower

than the resolution of our back-calculation method (although close to it for the LCDW layer, $\sim 5\,\mu mol\,kg^{-1}$ for the 16-year period).

## 6.2 Circulation and biological processes at steady state.

The back-calculation method assumes the circulation of the ocean and the biological processes are in steady state. The contribution of non-linear mixing is unknown and some of the changes in DIC in the water mass layers could be erroneously

included in the estimates of $C_{ANT}$ rather than as a non-anthropogenic change in DIC. The non-steady state of the circulation in our analysis is included to some extent through the changes in $CDIS^{\pi}$ (Eqs. (3, 4), which is solved by the OMP analysis, which is subjected to the limitations of quantifying the mixing mostly through thermohaline changes in the water masses (section A.2 of the Appendix).

For biological processes, remineralization rates are usually considered to be in steady state (Sarmiento et al., 1992). Climate

change has been suggested as potentially driving changes in carbon fixation and export that can influence the uptake of $CO_2$ by the oceans (Falkowski et al., 1998). Pahlow and Riebesell, (2000) first suggested that decadal changes in remineralization rates occurred in the deep waters of the Northern Hemisphere, although this is still a matter of debate (e.g., Li and Peng, 2002; Najjar, 2009).

Metzl et al. (1999) and Shadwick et al. (2015) observed that the uptake of $CO_2$ over the sub-Antarctic zone (SAZ, between the

SAF and the STF) in summer is mostly controlled by biological processes. If a change in remineralization rates has occurred, i.e. the changes in $DIC^{BIO}$ are due to biological effects, a change in nutrient concentrations of the water masses would be expected. The detection of long-term trends of nutrients at upper layers of the ocean can be masked by short time scale physical




processes such as changes in the mixed layer depth, mesoscale activity and advection (Sambrotto and Mace, 2000; Rintoul and Trull, 2001; Sallée et al., 2010). We did not find a measurable trend in nutrient concentrations for any of the layers in the period 1995-2011, except for an increase of SiO$_4$ over the Antarctic Divergence (Table 4) that is most likely due to the upwelling of SiO$_4$-rich deep waters. We cannot confidently assign the changes of $DIC^{BIO}$ and its impact on $\frac{\partial C_{ANT}}{\partial t}$ in upper

layers (Table 5) to any particular process and, instead we provide a range of values for $\frac{\partial C_{ANT}}{\partial t}$ (Table 4). For the scenario of intensification of the Antarctic upwelling, the increase in low-O$_2$ and DIC-rich waters would increase the content of DIC in subsurface waters, leading (at least for the AASW and AASW$_{upw}$ layers) to values of $\frac{\partial C_{ANT}}{\partial t}$ closer to the lower limit of the range (i.e., $\frac{\partial DIC}{\partial t} >> \frac{\partial C_{ANT}}{\partial t}$, Table 4).

In deep layers of the section, the increase of DIC is not explained by the long-term change of any of the terms in Eq. (1), which

is other implication of considering the circulation in steady state. The differences found in the increase of DIC in the UCDW layer north and south of the SAF (section 4.3, Fig. 4) add consistency to the idea of the advection of older waters to the section. Considering these differences we can assign a change in DIC of at least ~0.20 ± 0.02 µmol kg$^{-1}$ yr$^{-1}$ (lower rate of increase of DIC found in deep-bottom layers, Table 4) due to the upwelling intensification. Since the AASW and AASW$_{upw}$ layers are the most affected by the upwelling we can correct the values of $\frac{\partial C_{ANT}}{\partial t}$ in these layers for this effect (trends with ** in Table 4).

**6.3 Changes in the rates of export of particulate organic carbon from surface layers.**

We assume that the export of particulate organic carbon (POC) from upper ocean layers (STCW, AASW and AASW$_{upw}$) and remineralization in the water column was constant between 1995 and 2011. The high-latitudes are considered important in terms of POC export, mostly because these areas are dominated by large phytoplankton, in particular diatoms (Buesseler, 1998; Sambrotto and Mace, 2000), and rapid carbon export to deep waters from phytoplankton blooms is possible (DiTullio et al.,

2000; Lourey and Trull, 2001). The POC exported is remineralized to DIC below the mixed layer (Wassman et al., 1990; Asper and Smith, 1999; Trull et al, 2001a; Fripiat et al., 2015).

Our results show that the increase of DIC in mode and intermediate waters is fully explained by the uptake of atmospheric CO$_2$, which could indicate that there was no detectable change in the rate of export of POC over the 1995-2011 period. Estimates of POC export in the SAZ (~3800 m), the SAF (~3100 m) and the PFZ (~1500 m) using moored sediment traps near

the section (Trull et al., 2001b) were 0.5, 0.8 and 1.0 g C m$^{-2}$ yr$^{-1}$, respectively. If all this POC is fully remineralized in the UCDW layer (with a mean thickness of 2000m), we obtain a range of 0.02 – 0.04 µmol kg$^{-1}$ yr$^{-1}$ for the maximum increase of DIC due to the export of POC. This increase is close to the uncertainty of the total DIC increase estimated for the UCDW layer and we conclude that changes in export and associated remineralisation are unlikely to influence the calculated trends in the deep water masses.

The observed increase of SiO$_4$ in deep and bottom layers of the ocean is consistent with the transport of SiO$_4$-rich older waters to the section. For UCDW, the increase of SiO$_4$ north of the SAF is higher than at southern latitudes (0.31 ± 0.08 µmol kg$^{-1}$ yr$^-$



<sup>1</sup> respect to 0.19 ± 0.04 µmol kg$^{-1}$ yr$^{-1}$, not shown). Nelson et al., (1995) observed a lower dissolution rate of diatom dominated SiO$_4$ exported in high-latitudes regions compared to lower latitudes. Nelson et al. (1995) estimated a mean silica production rate of 0.7 – 1.2 mol Si m$^{-2}$ yr$^{-1}$ for regions over diatomaceous sediments and concluded that 15 – 25 % of the silica produced in the upper ocean accumulates in the seabed. Of the silica produced in the mixed layer of the upper ocean layers at least 50%

is believed to dissolve in the upper 100 m of the water column (e.g. Nelson et al., 1991; DeMaster et al., 1992). For deep waters, the production rates of Nelson et al. (1995) could result in a mean increase of SiO$_4$ of 0.08 – 0.14 µmol kg$^{-1}$ yr$^{-1}$ in the water column (~3400 m, mean depth). The maximum value of this increase could explain the trends of SiO$_4$ found for the AABW layer (Table 4), but 32-48 % of the increase of SiO$_4$ in deep bottom layers is not explained by the remineralisation of exported silica and is most likely the result of the advection of older SiO$_4$-rich waters to the section.

**6.4 Stoichiometric ratios for biological processes.**

The back-calculation method and the OMP analysis assume constant stoichiometric ratios for remineralization. The theoretical Redfield ratios (Redfield, 1934; 1958) are usually considered as a mean for the whole ocean, although they can vary from the theoretical value due to changes in phytoplankton species composition, the food-web structure and nutrient availability (Martiny et al., 2013).

We carried out a sensitivity analysis on the Redfield ratios following Álvarez et al., (2014), to obtain values of stoichiometric ratios for the section. A battery of OMP analyses with varying values of $R_N$ between 9 and 10 in increments of 0.2, $R_P$ between 120 and 145 in increments of 5, and $R_{Si}$ between 0 and 8 in increments of 2. For each variation in the stoichiometric ratios, an OMP analysis was made for each section in order to determine best-fit R values and if there were differences in time for the stoichiometric ratios along the sections. The smallest residuals (differences between the nutrients measured and the estimated

by the OMP analysis) were obtained for $R_N$ = 9 and $R_P$ = 125. The residuals of SiO$_4$ did not change significantly for any value of $R_{Si}$ and we consider SiO$_4$ as a conservative variable. These results indicate $\frac{R_P}{R_N}$ = 13.8, in agreement with values obtained for the region ($\frac{R_P}{R_N}$ ∈ [8-15], Lourey and Trull, 2001).

**7 Conclusions.**

The results of our analysis south of Tasmania over the 1995-2011 period support a scenario of intensification of upwelling in

the vicinity of the Antarctic Divergence due to an increase in the westerly winds at high latitudes most probably linked to the variability of the SAM. The intensification of the upwelling favours the advection of older waters to deep-bottom layers of the section where we found net increase in DIC over the 16-year period. The enhanced upwelling causes the eventual entrainment of low-O$_2$ and DIC-rich waters into upper layers, explaining the trends of decreasing O$_2$ and increasing non-anthropogenic DIC found in surface waters close to the Antarctic Divergence. This scenario also implies the intensification of the convergence

north of the SAF, implying a more efficient ventilation of the SAMW and AAIW layers and thus an efficient uptake of



atmospheric $CO_2$ by these layers. The enhanced upwelling lowers the uptake of $C_{ANT}$ in the AASW layer but the effect of ventilation more than compensates that of the upwelling allowing the increase in $C_{ANT}$ in this layer. The atmospheric warming reduces the dissolution of $CO_2$ in upper layers north of the PF, presenting increases in DIC lower than expected from the atmospheric $CO_2$ increase.

**Acknowledgements**

The SR03 section was sampled as part of the World Ocean Circulation Experiment/$CO_2$ Survey (WOCE, http://woceatlas.ucsd.edu), and more recently the Global Ocean Ship-Based Hydrographic Investigation Program (GO-SHIPS, http://www.go-ship.org). Carbon system parameters contribute to the International Ocean Carbon Coordination Project of the United Nations Intergovernmental Oceanographic Commission (IOCCP, http://www.ioccp.org). Support for measurements on
the section were provided to S. R. and B. T. by the Antarctic Climate and Ecosystems Cooperative Research Centre (ACE CRC) and the Australian Climate Change Science Program. Logistic support for the section and ship time on the RSV Aurora Australis was provided by the Australian Antarctic Division. The many scientific staff involved in the hydrographic sections and the officers and crew of the ship were critical to obtaining good quality data. We especially want to thank the work by Kate Berry and Mark Pretty for high quality DIC and TA data, and for the hydrochemistry teams and CTD watches of multiple
cruises and especially Mark Rosenberg and Rebecca Cowley. The first author of this paper is a postdoctoral fellow supported by the ACE-CRC Project R2.1: Carbon Uptake and Chemical Change.

**A Supplementary material**

**A.1 Biogeochemical Model.**

The 1/10° biogeochemical ocean simulation is based on the near-global Ocean Forecasting Australia Model configuration (OFAM3) (Oke et al., 2013) with 51 vertical layers (14 layers between the surface and 100 m depth and partial cells to better represent bottom topography), a resolution of 4.7km at 65°S, 7.8km at 45°S and a constant meridional resolution of 11 km. This configuration represents the frontal structure and filament nature of the ACC and captures much of the mesoscale variability and the development of baroclinic eddies (Langlais et al., 2011, 2015). OFAM3 includes the World Ocean Model
of Biogeochemistry and Trophic dynamics (WOMBAT) (Kidston et al., 2011; Oke et al., 2013) that is based on a nutrient, phytoplankton, zooplankton and detritus model, with the addition of an $O_2$ and $CO_2$ cycle. The ocean model is based on the version 4p1d of the Geophysical fluid Dynamics Laboratory Modular Ocean Model (Griffies, 2009). Horizontal mixing is provided by the biharmonic Smagorinsky viscosity scheme (Griffies and Hallberg 2000), and vertical mixing by the K-profile parameterization (KPP, Large et al. 1994). It has two tracers of DIC, one that sees an (pre-industrial) atmospheric value of 280
ppm (natural carbon tracer), and a second tracer that sees the observed rising atmospheric $pCO_2$ (total carbon tracer). The BGC



parameters used with WOMBAT are based on Oschlies and Schartau (2005), with extra parameters for the carbon cycle (Law et al. 2015).

The 1/10° simulation spans from 1979 to 2014, forced by 3-hourly Japanese 55-year Reanalysis (JRA-55; Kobayashi et al. 2015), using bulk formula (Large and Yeager 2004) for wind stress, turbulent sensible and latent fluxes, and evaporation. As

the model is not coupled to a sea-ice model, the effects of sea-ice on heat and freshwater fluxes are accounted for by the use of the JRA-55 sea ice coverage field to mask the applied atmospheric fields. Below 2000m, a non-adaptive relaxation keeps the deep-ocean close to the observed climatology but allows the climate changes signals to penetrate to the deep ocean.

The model BGC fields are initialised with fields constructed from observations. Specifically, the World Ocean Atlas is used to initialise nutrients (phosphorus) and oxygen (Garcia et al 2006a; Garcia et al. 2006b). The Global Ocean Data Analysis

Project (GLODAP) is used to initialise alkalinity and dissolved carbon (Sabine et al. 2004; Key et al. 2004). Phytoplankton is initialised with SeaWIFS observation (NASA, 2014) and zooplankton is initialised as a fraction of phytoplankton (0.05). As the inclusion of the BGC component is computationally expensive, the BGC fields are only integrated between 1992 and 2014. This duration spans the era of relevant satellite sea-colour observations.

**A.2 OMP analysis.**

The OMP method (Thompson and Edwards, 1981; Tomczak, 1981; Mackas et al., 1987; Tomczak and Large, 1989) considers the water samples as nodes of a grid in which the different properties (e.g., S, θ, $O_2$) are measured. The OMP assumes that the value of each property is the result of the linear mixing of the water masses characterizing the region of study, which are called end members or source water types (SWT) and whose characteristics are known. Thus, the value of each node can be expressed as a linear combination of the SWTs:

$$P_i = \sum_j^{Nj} \left( SWT_j * P_j + R_i * \Delta O \right)$$

where $P_i$ is the value of the properite i in the node, $P_j$ is the value of the correspondent properties of the j SWT. Since some of the measured variables are non-conservative (e.g., $O_2$), biogeochemical terms have to be included in the mixing equations ($R_i * \Delta O$) that are based on Redfield ratios and considered constant ($R_i$, Broecker, 1974; Anderson and Sarmiento 1994; Martiny et al., 2013). Here we use $R_N$=9 and $R_P$ = 125, referenced to the oxygen consumption ($\sim \Delta O$) which were the optima

after a sensibility analysis (Álvarez et al., 2014). These stoichiometric ratios are in agreement with previous studies (Le Jehan and Treguer, 1983; Verlencar et al., 1990; Lourey and Trull, 2001, see section 4.4.2).

The system of equations (resulting from considering all the properties measured in each sample plus a mass balance equation) is normalized and solved by a least square method with a positive definite constraint to obtain the fractions of each of the SWTs characterizing the water sample and satisfying the mass balance equation.

Before solving the system, each equation is weighted (the mass equation presents the highest weight to ensure its conservation) based on the accuracy of the property and/or the variability in the region of study (Table A1). Weights were also adjusted so



that the ratios between the Standard Deviations of the Residuals and the analytical error (ε, Table A1) were almost the same for all the SWT properties (Table A1).

Here we consider 11 SWTs that characterize the water masses of the SR03 section, i.e., those that best enclose the main features of the T/S diagram and of other properties of the water masses of the section (Fig. A1). The conservative properties (θ and S) were defined based on bibliography available (Table A1):

- In order to take into account the subtropical waters two points were defined as upper limits in the T/S diagram (Fig. A1). These end members correspond to the main properties of both the Zeehan Current (ZC) and the EAC arriving to the north part of the section (Fig. 1). The end member $SWT_{STW15}$ represents the extension of the ZC (~15°C, ~35) in winter to the region south of Tasmania described by Cresswell (2000). The reference for the characterization of the subtropical waters from the EAC, i.e., $SWT_{STW16}$, is the southern component of the Subtropical Lower Water, characterized by Sokolov and Rintoul (2000) as waters in the range 16-22°C and 35.5-35.7. These two end members are considered together for the study as $SWT_{STW} = SWT_{STW16} + SWT_{STW15}$.

- Two end members are used to represent the seasonal warming of the AASW as it extends from the Antarctic shelf to latitudes of the SAF. $SWT_{AASW}$ is the AASW described by Mosby (1934) and defined by Pardo et al. (2012). $SWT_{SASW}$ represents the warmest type of AASW found in summer based on Chaigneau et al. (2004). In our study, the two endmembers are considered together as $SWT_{AAS} = SWT_{AASW} + SWT_{SASW}$, as is the case of subtropical waters.

- HSSW is produced in coastal polynyas, where ice-formation creates salty surface waters (Orsi et al., 2002). This water is also the precursor in the formation of bottom waters. The definition of the end member $SWT_{HSSW}$ was obtained from the study of Lacarra et al. (2011) within the Adélie -George V Land coast (Fig. 1), one of the areas of formation of ALBW.

- $SWT_{SAMW}$ represents the core of the SAMW that is ventilated south of Tasmania (Rintoul and Bullister, 1999). We defined this end member as a point inside the cluster defined by Herraiz-Borreguero and Rintoul (2010) (8.5-9°C, 34.58-34.68).

- The end member characterizing AAIW, $SWT_{AAIW}$, is the variety of AAIW found south of Tasmania and close to the SAF and is defined by T of 4-4.5°C and S of 34.35 after Rintoul and Bullister, (1999).

- $SWT_{NADW}$ reflects the properties of the NADW in South Atlantic, when it arrives to the ACC and is defined by Pardo et al. (2012).

- The end member $SWT_{CDW}$ refers to waters in the deep bottom layers of the ACC that result from the mix with NADW arriving to the ACC, AAIW from above and the upper layers of the AABW, which, in specific locations of the Antarctic continent contributes to the formation of bottom waters. The properties of this end member are taken from Pardo et al. (2012).




- The end member SWT$_{PIDW}$ refers to waters in the deep layers of the ACC (known as Lower Circumpolar Deep water) that are fed by deep waters from the Pacific and Indian Oceans and are characterized by a silicate maximum, high nutrients and low oxygen (e.g., Callahan, 1972; Withworth et al., 1998). The values from the SWT$_{PIDW}$ were obtained from Talley et al. (2011).

- SWT$_{AABW}$ is the end member representing the bottom waters in the southern end of the section, close to the Antarctic shelf that result from a mixing between recently formed ALBW and RSBW. The definition of this end member is based on the observations from Rintoul and Bullister (1999).

The values of the non-conservative variables of the SWTs were initially extrapolated from regression lines with salinity and temperature (Poole and Tomczak, 1999) and then subjected to an iterative process in OMP in order to obtain the types that
best fit the cruise data.

The number of SWTs included in the mixing depends on the number of properties measured at the node. In order to have enough degrees of freedom to solve the system of equations, we use combinations of water masses that we call mixing groups (Table A2), to solve different regions of the section. Each mixing group is connected to the other by one or more SWTs in order to maintain the continuity of the analysis (Fig. A1), and are defined by considering the vertical characteristics and/or
dynamics of the water masses in the region of study.

The robustness of the OMP analysis is tested through a perturbation analysis of uncertainties (Lawson and Hanson, 1974). The properties of both each SWT and each water sample are perturbed in order to check the sensitivity of the model to variations in the SWTs, due to environmental variability, and in the water samples, due to measurement errors (Leffanue and Tomczak, 2004). The uncertainties of the SWTs fractions (mean standard deviation of 100 perturbation runs) are shown together with
the percentage of variability explained by the OMP analysis for each variable (Table A1). The model is reliable since it explains at least 98% of the variability of all the variables implicated (Table A1).

**A.3 Parameterizations of TA$^0$ and CDIS.**

The values of TA$^0$ and CDIS are defined for each SWT (Table A1) using the parameterizations and values from Pardo et al. (2011) and Pardo et al. (2014) (Table A3). Since mode and intermediate waters (SWT$_{SAMW}$ and SWT$_{AAIW}$, Table A3) are
formed and ventilated in areas influenced by subtropical waters and Antarctic waters, we combine the parameterizations for subtropical ([1]) and Antarctic ([2]) regions to obtain better values of both parameters (Table A3). The values of TA$^0$ and CDIS for the SWTS of mode and intermediate waters are estimated as $TA^0 = \frac{2}{3}(TA^0[1]) + \frac{1}{3}(TA^0[2])$ and $CDIS = \frac{2}{3}(CDIS[1]) + \frac{1}{3}(CDIS[2])$ (Table A2). The values obtain for TA$^0$ and CDIS for each SWT are then extended to the water column using the results from the OMP analysis:

$$TA^0 = \sum_{j=1}^{11}\left(SWT_j * TA_j^0\right); \quad CDIS = \sum_{j=1}^{11}\left(SWT_j * CDIS\right)$$





The appropriate combination of the parameterizations of $TA^0$ for the section is obtained by analysing the differences between TA and $TA^0$ at surface layers of the section, once $TA^0$ is determined by OMP analysis in order to avoid negative values. The values obtained for CDIS were very similar for all the SWTs except for $SWT_{HSSW}$, with an estimated CDIS value 4 times bigger than that some of the SWTs. Since the results from Landschutzer et al. (2015) indicate that the disequilibrium values

5 do not change much between AASW and SAMW, we considered the value of CDIS obtained from the monthly mean values of atmospheric $CO_2$ from the NOAA network (1968-2015, Dlugokencky, et al., 2016) for latitudes > 62°S, as the most appropriate for $SWT_{HSSW}$.

For the $SWT_{NADW}$, the value of $TA^0$ are obtained from the GLODAPv2 climatology (http://cdiac.ornl.gov/oceans/GLODAPv2) in the area of the Indian Ocean and that of CDIS from Pardo et al. (2014). The SWTs for the rest of deep and bottom waters

10 are formed by mixing of other water masses and the values of $TA^0$ and CDIS are obtained through an iterative process considering the composition of each SWT (Table A3) obtained from temperature and salinity. In the iterative process a default value is given to $TA^0$ and CDIS of $SWT_{AABW}$ and iterations are run for the different deep-bottom water masses until the differences between the values of $TA^0$ and CDIS of two consecutive iterations are less than 0.005 for $TA^0$ and 0.05 for CDIS.

15 **References.**

Álvarez, M., Brea, S., Mercier, H. and Álvarez-Salgado, X.A.: Mineralization of biogenic materials in the water masses of the South Atlantic Ocean. I: assessment and results of an optimum multiparameter analysis, Progress in Oceanography, 123, 1-23, doi: 10.1016/j.pocean.2013.12.007, 2014.

Anderson, L.A. and Sarmiento, J.L., 1994. Redfield ratios of remineralization determined by nutrient data analysis. Global

20 Biogeochemical Cycles 8 (1), 65–80, doi: 10.1029/93GB03318, 1994.

Aoki, S., Bindoff, N.L. and Church, J.A.: Interdecadal water mass changes in the Southern Ocean between 30°E and 160°E, 32, L07607, doi:10.1029/2004GL022220, 2005.

Aoki, S., et al.: Atlantic–Pacific asymmetry of subsurface temperature change and frontal response of the Antarctic Circumpolar Current for the recent three decades, J Oceanogr., 71, 623–636, doi:10.1007/s10872-015-0284-6, 2015.

25 Armour, K. and Bitz, C.M.: Observed and projected trends in Antarctic sea ice, US Clivar Variations Newsletter, 13, 4, 12-19, 2015.

Asper, V.L. and Smith, W.O.Jr.: Particle fluxes during austral spring and summer in the southern Ross Sea, Antarctica, J. Geophys. Res., 104, C3, 5345-5359, 1999.

Baines, P.G., Edwards, R.J. and Fandry, C.B.: Observations of a new baroclinic current along the western continental slope of

30 Bass Strait, Aust. J. Mar. Freshwater Res., 34, 155–157, 1983.

Bates, N.R., et al.: A time-series view of changing ocean chemistry due to ocean uptake of anthropogenic $CO_2$ and ocean acidification, Oceanography 27, 1, 126–141, http://dx.doi.org/10.5670/oceanog.2014.16, 2014.




Belkin, I.M. and Gordon, A.L.: Southern Ocean fronts from the Greenwich meridian to Tasmania, Journal of Geophysical Research, 101, C2, 3675-3696, 1996.

Bender, M., et al.: Variability in the O2/N 2 ratio of southern hemisphere air, 1991-1994: Implications for the carbon cycle, Global Biogeochemical Cycles, 10, 1, 9-21, 1996.

Bindoff, N.L. and Church, J.A.: Warming of the Water Column in the Southwest Pacific Ocean, Nature, 357, 59-62, 1992.

Bindoff, N.L. and McDougall, T.J.: Diagnosing Climate Change and Ocean Ventilation using Hydrographic Data, Journal of Physical Oceanography, 24, 1137-1152, 1994.

Boland, F.M. and Church, J.A.: The East Australian Current 1978, Deep-Sea Res., 28A, 937–957, doi:10.1016/0198-0149(81)90011-X, 1981.

Böning, C.W., Dispert, A., Visbeck, M., Rintoul, S.R. and Schwarzkopf, F.U.: The response of the Antarctic Circumpolar Current to recent climate change, Nature geoscience, 1, 864-869, doi:10.1038/ngeo362, 2008.

Bostock, H.C., Mikaloff Fletcher, S.E. and Williams, M.J.M.: Estimating carbonate parameters from hydrographic data for the intermediate and deep waters of the Southern Hemisphere oceans, Biogeosciences, 10, 6199–6213, doi:10.5194/bg-10-6199-2013, 2013.

Broecker, W.S.: "NO" a conservative water mass tracer. Earth and Planetary Science Letters 23, 8761–8776, 1974.

Buesseler, K.O.: The decoupling of production and particle export in the surface ocean, Global Biogeochemical Cycles, 12, 2, 297-310, 1998.

Callahan, J.E.: The structure and circulation of Deep Water in the Antarctic, Deep-Sea Research, 19, 563-575, 1972.

Carter, B. R., et al.: Two decades of Pacific anthropogenic carbon storage and ocean acidification along Global Ocean Ship-
based Hydrographic Investigations Program sections P16 and P02, Global Biogeochem. Cycles, 31, doi:10.1002/2016GB005485, 2017.

Chaigneau, A., Morrow, R.A. and Rintoul, S.R.: Seasonal and interannual evolution of the mixed layer in the Antarctic Zone south of Tasmania, Deep-Sea Research I, 51, 2047–2072, doi:10.1016/j.dsr.2004.06.013, 2004.

Chen, C.-T.A. and Millero, F.J.: Gradual increase of oceanic $CO_2$, Nature, 277, 205–206, 1979.

Chen, C.-T.A., Pytkowicz, M.R. and Olson, E.J.: Evaluation of the calcium problem in the South Pacific, Geochemical Journal, 16, 1-10, 1982.

Cresswell, G.: Currents of the continental shelf and upper slope of Tasmania, In Banks, M.R. & Brown, M.j. (Eds): Tasmania and the Southern Ocean, Pap. Proc. R. Soc. Tasm., 133, 3, 21-30, 2000.

Davis, R.: Intermediate-depth circulation of the Indian and South Pacific oceans measured by autonomous floats, J. Phys.
Oceanogr., 35, 683–707, 2005.

Deacon, G.E.R.: The hydrology of the Southern Ocean, Cambridge University Press, 15, 1-124, ill. maps, plates, 1937.

DeMaster, D.J., et al.: The cycling and accumulation of organic matter and biogenic silica in high-latitude environments: The Ross Sea, Oceanography, 5, 146-153, 1992.



DeVries, T., Holzer, M. and Primeau, F.: Recent increase in oceanic carbon uptake driven by weaker upper-ocean overturning, Nature, 542, 215-218, doi:10.1038/nature21068, 2017.

Dickson, R.R. and Brown, J.: The production of North Atlantic Deep Water: sources, rates, and pathways, Journal of Geophysical Research, 99, C6, 12319-12341, doi:10.1029/94JC00530, 1984.

Dickson, A. G.: Thermodynamics of the dissociation of boric acid in synthetic seawater from 273.15 to 318.15 K, Deep-Sea Res. I, 37, 755–766, doi:10.1016/0198-0149(90)90004-F, 1990.

Dickson, A. G. and Millero, F. J.: A comparison of the equilibrium constants for the dissociation of carbonic acid in seawater media, Deep-Sea Res. I, 34, 1733–1743, doi:10.1016/0198-0149(87)90021-5, 1987.

Dickson, A.G., Sabine, C.L. and Christian, J.R.: Guide to Best Practices for Ocean $CO_2$ Measurements, PICES Special
Publication 3, 191 pp, 2007.

DiTullio, G.R., et al.: Rapid and early export of Phaeocystis antarctica blooms in the Ross Sea, Antarctica, Nature, 404, 595-598, 2000.

Doney, S.C., Victoria, J.F., Feely, R.A. and Kleypas, J.A.: Ocean Acidification: The other $CO_2$ problem, Annu. Rev. Mar. Sci., 1, 169–92, doi:10.1146/annurev.marine.010908.163834, 2009.

Dlugokencky, E.J., Lang, P.M., Mund, J.W., Crotwell, A.M., Crotwell, M.J. and Thoning, K.W.: Atmospheric Carbon Dioxide Dry Air Mole Fractions from the NOAA ESRL Carbon Cycle Cooperative Global Air Sampling Network, 1968-2015, Version: 2016-08-30,ftp://aftp.cmdl.noaa.gov/data/trace_gases/co2/flask/surface/, 2016.

Falkowski, P.G., Barber, R.T. and Smetacek, V.: Biogeochemical Controls and Feedbacks on Ocean Primary Production, Science, 281, 5374, 200-206, doi: 10.1126/science.281.5374.200, 1998.

Fay, A. R., McKinley, G.A. and Lovenduski, N.S.: Southern Ocean carbon trends: Sensitivity to methods, Geophys. Res. Lett., 41, 6833–6840, doi:10.1002/2014GL061324, 2014.

Feely, et al.: The impact of anthropogenic $CO_2$ on the $CaCO_3$ system in the oceans, Sience, 305, 362-366, 2004.

Foster, T.D. and Carmack, E.C.: Frontal zone mixing and Antarctic Bottom Water formation in the southern Weddell Sea, Deep Sea Research 23, 301–307, 1976.

Friis, K., Körtzinger, A., J. Pätsch, J., Wallace, D.W.R.: On the temporal increase of anthropogenic CO2 in the subpolar North Atlantic, Deep-Sea Research I, 52, 681–698, doi:10.1016/j.dsr.2004.11.017, 2005.

Fripiat, F., et al.: Significant mixed layer nitrification in a natural iron-fertilized bloom of the Southern Ocean, Global Biogeochem. Cycles, 29, 1929–1943, doi:10.1002/2014GB005051, 2015.

Fukamachi, Y. et al.: Strong export of Antarctic Bottom Water east of the Kerguelen plateau, Nature, 3, 327-331, 2010.

Garcia, H.E., Locarnini, R.A., Boyer, T.P. and Antonov, J.I.: World ocean atlas 2005, volume 3: dissolved oxygen, apparent oxygen utilization, and oxygen saturation, In: Levitus, S. (Ed.), NOAA Atlas NESDIS 63. U.S. Government Printing Office, Washington, D.C., p. 342, 2006a.

Garcia, H.E., Locarnini, R.A., Boyer, T.P. and Antonov, J.I.: World ocean atlas 2005, volume 4: nutrients (phosphate, nitrate, silicate). In: Levitus, S. (Ed.), NOAA Atlas NESDIS 64. U.S. Government Printing Office, Washington, D.C., p. 396, 2006b.



Gordon, A.L. and Tchernia, P.: Waters of the continental margin off Adélie Coast, Antarctica, In: Antarctic Oceanology II: The Australian–New Zealand Sector, D.E. Hayes, ed., Antarctic Research Series 19, American Geophysical Union, Washington, DC, 59–69, 1972.

Griffies, SM., et al.: Coordinated Ocean-Ice Reference Experiments (COREs), Ocean Modell., 26, 1–46, doi:10.1016/j.ocemod.2008.08.007, 2009.

Griffies, S.M. and Hallberg, R.W.: Biharmonic friction with a Smagorinsky viscosity for use in large-scale eddy permitting ocean models, Mon. Weather Rev. 128, 2935–2946, 2000.

Gruber, N.: Anthropogenic $CO_2$ in the Atlantic Ocean, Global Biogeochemical Cycles, 12, 1, 165–191, 1998.

Hanawa, K. and Talley, L.D.: Mode waters, In: Siedler, G., Church, J., Gould, J. (Eds.), Ocean Circulation and Climate, International Geophysics Series, Academic Press, New York, 373–386, 2001.

Hauri, C., et al.: Two decades of inorganic carbon dynamics along the West Antarctic Peninsula, Biogeosciences, 12, 6761–6779, doi:10.5194/bg-12-6761-2015, 2015.

Herraiz-Borreguero, L. and Rintoul, S.R.,: Subantarctic mode water variability influenced by mesoscale eddies south of Tasmania. J. Geophys. Res. 115, C04004, doi:10/1029/2008JC005146, 2010.

Herraiz-Borreguero, L and Rintoul, S.R: Regional circulation and its impact on upper ocean variability south of Tasmania, Deep-Sea Research II, 58, 2071–2081, doi:10.1016/j.dsr2.2011.05.022, 2011.

Hill, K.L., Rintoul, S.R., Ridgway, K.R. and Oke, P.R.: Decadal changes in the South Pacific western boundary current system revealed in observations and ocean state estimates, J. Geophys. Res., 116, C01009, doi:10.1029/2009JC005926, 2001.

Hood, E. M., Sabine, C. L. and Sloyan, B. M.: The GO-SHIP Repeat Hydrography Manual: a Collection of Expert Reports and Guidelines, IOCCP Report Number 14, OCPO Publication Series Number 134, http://www.go-ship.org/HydroMan.html, 2010.

Ikegami, H. and Kanamori, S.: Calcium-Alkalinity-Nitrate Relationship in the North Pacific and the Japan Sea, Journal of the Oceanographical Society of Japan, 39, 9-14, 1983.

Iudicone, D., Speich, S., Madec, G. and Blanke, B.: The Global Conveyor Belt from a Southern Ocean perspective, 38, 1401-1425, doi:10.1175/2007JPO3525.1, 2008.

Jackett, D.R. and McDougall, T.J.: A Neutral Density Variable for the World's Oceans, Journal of Physical Oceanography, 27, 237-263, 1997.

Jacobs, S. S.: On the nature and significance of the Antarctic Slope Front, Mar. Chem., 35, 9 –24, 1991.

Jacobs, S.: Observations of change in the Southern Ocean, Phil. Trans. R. Soc. A, 364, 1657–1681, doi:10.1098/rsta.2006.1794, 2006.

Johnson, G.C.: Quantifying Antarctic Bottom Water and North Atlantic Deep Water volumes, J. Geophys. Res., 113, C05027, doi:10.1029/2007JC004477, 2008.

Joyce, T. and Corry, C., 1994: Requirements for WOCE Hydrographic Programme Data Reporting, WHPO Publication 90-1 Revision 2, WOCE Report 67/91, Woods Hole, Mass., USA, 1994.



Key, R.M., et al.: A global ocean carbon climatology: Results from Global Data Analysis Project (GLODAP), Global Biogeochem. Cycles, 18, GB4031, doi:10.1029/2004GB002247, 2004.

Khatiwala, S., Primeau, F. and Hall, T.: Reconstruction of the history of anthropogenic $CO_2$ concentrations in the ocean, Nature, 462, 346-350, doi:10.1038/nature08526, 2009.

Kidston, M., Matear, R. J. and Baird, M. E.: Parameter optimisation of a marine ecosystem model at two contrasting stations in the Sub-Antarctic Zone, Deep Sea Res., 58, 2301–2315, 2011.

Kobayashi, S., et al.: The JRA-55 Reanalysis: General specifications and basic characteristics. J. Meteor. Soc. Japan, 93, 5-48, doi:10.2151/jmsj.2015-001, 2015.

Kouketsu, S., Murata, A.M.: Detecting decadal scale increases in anthropogenic CO2 in the ocean, Geophys. Res. Lett., 41,

4594–4600, doi:10.1002/2014GL060516, 2014.

Lacarra, M., et al.: Summer hydrography on the shelf off Terre Ade ́lie/George V Land based on the ALBION and CEAMARC observations during the IPY, Polar Science, 5, 2011, 88-103, doi:10.1016/j.polar.2011.04.008, 2011.

Landschützer, P., et al.: The reinvigoration of the SouthernOcean carbon sink, Science, 349, 1221–1224, doi:10.1126/science.aab2620, 2015.

Langlais, C., Rintoul, S. and Schiller, A.: Variability and mesoscale activity of the Southern Ocean fronts: Identification of a circumpolar coordinate system, doi:10.1016/j.ocemod.2011.04.010, Ocean Modelling, 39, 79–96, 2011.

Langlais, C.E., Rintoul, S.R. and Zika, J.D.: Sensitivity of Antarctic Circumpolar Current Transport and Eddy Activity to Wind Patterns in the Southern Ocean, Journal of Physical Oceanography, 45, 1051-1067, doi:10.1175/JPO-D-14-0053.1, 2015.

Large, W. G., McWilliams, J.C. and Doney, S.C.: Oceanic vertical mixing: a review and a model with a nonlocal boundary

layer parameterization. Rev. Geophys., 32, 363-403, 1994.

Large, W.G. and Yeager, S.G.: Diurnal to decadal global forcing for ocean and sea-ice models: the data sets and flux climatologies, TN-460+STR, NCAR, 111 pp, doi: 10.5065/D6KK98Q6, 2004.

Lauvset, S.K., Gruber, N., Landschützer, P., Olsen, A. and Tjiputra, J.: Trends and drivers in global surface ocean pH over the past 3 decades, Biogeosciences, 12, 1285–1298, doi:10.5194/bg-12-1285-2015, 2015.

Lawson, C.L. and Hanson, R.J.: Solving Least Squares Problems, Prentice-Hall, Englewood Cliffs, NJ., 1974.

Law, et al.: The carbon cycle in the Australian Community Climate and Earth System Simulator (ACCESS-ESM1) – Part 1: Model description and pre-industrial simulation, Geosci. Model Dev. Discuss., 8, 8063–8116, doi:10.5194/gmdd-8-8063-2015, 2015.

Leffaune H. and Tomczak M.: Using OMP analysis to observe temporal variability in water mass distribution, J. Mar. Syst,

48, 3–14, 2004.

Le Jehan, S. and Treguer, P.: Uptake and regeneration ASi/AN/AP ratios in the Indian Sector of the Southern Ocean. Polar Biol. 2: 127-136, 1983.

Lenton, A. and Matear, R.J.: Role of the Southern Annular Mode (SAM) in Southern Ocean $CO_2$ uptake, Global Biogeochem. Cycles, 21, GB2016, doi:10.1029/2006GB002714, 2007.




Lenton, A., et al.: Global Biogeochemical Cycles, 26, GB2021, doi:10.1029/2011GB004095, 2012.

Lenton, A., et al.: Sea–air $CO_2$ fluxes in the Southern Ocean for the period 1990–2009, Biogeosciences, 10, 4037–4054, doi:10.5194/bg-10-4037-2013, 2013.

Le Quéré, C., et al.: Saturation of the Southern Ocean $CO_2$ Sink Due to Recent Climate Change, Science 316, 1735-1738, doi:
10.1126/science.1136188, 2007.

Le Quéré, C., et al.: Global carbon budget 2014, Earth Syst. Sci. Data, 7, 47–85, doi:10.5194/essd-7-47-2015, 2015.

Lewis, E. and Wallace, D.W.R.: Program Developed for $CO_2$ System Calculations. ORNL/CDIAC-105. Carbon Dioxide Information Analysis Center, Oak Ridge National Laboratory, U.S. Department of Energy, Oak Ridge, Tennessee, 1998.

Lourey, K.J. and Trull, T.W.: Seasonal nutrient depletion and carbon export in the Subantarctic and Polar Frontal Zones of the
Southern Ocean south of Australia, J. Geophys. Res., 106, C12, 31,463-31,487, 2001.

Lovenduski, N.S., Gruber, N. and Doney, S.C.: Towards a mechanistic understanding of the decadal trends in the Southern Ocean carbon sink, Global Biogeochemical Cycles, 22, GB3016, doi:10.1029/2007GB003139, 2008.

Lumpkin, R. and Speer, K.: Global Ocean Meridional Overturning, Journal of Physical Oceanography, 37, 2550-2562, doi:10.1175/JPO3130.1, 2007.

Mackas, D.L., Denman, K.L. and Bennett, A.F.: Least squares multiple tracer analysis of water mass composition, Journal of Geophysical Research ,92, C3, 2907–2918, doi: 10.1029/JC092iC03p02907, 1987.

Mantyla, A.W. and Reid, J.L.: On the origins of deep and bottom waters of the Indian Ocean, Journal of Geophysical Research, 100, C2, 2417-2439, 1995.

Marshall, G.J.: Analysis of recent circulation and thermal advection change in the northern Antarctic Peninsula, Int. J.
Climatol., 22, 1557–1567, doi:10.1002/joc.814, 2002.

Marshall, G.J.: Trends in the Southern Annular Mode from observations and reanalyses, Journal of Climate, 16, 4134-4143, 2003.

Marsland, S. J., Bindoff, N.L., Williams, G.D. and Budd, W.F.: Modeling water mass formation in the Mertz Glacier Polynya and Adélie Depression, East Antarctica, J. Geophys. Res., 109, C11003, doi:10.1029/2004JC002441, 2004.

Martiny, A.C., et al.: Strong latitudinal patterns in the elemental ratios of marine plankton and organic matter, Nature, 6, 279-283, doi: 10.1038/NGEO1757, 2013.

Matear, R.J., Hirst, A.C. and McNeil, B.I.: Changes in dissolved oxygen in the Southern Ocean with climate change, Geochem. Geophys. Geosyst., 1, 2000GC000086, ISSN: 1525-2027, 2000.

Matear, R. and Lenton, A.: Impact of historical climate change on the Southern Ocean carbon cycle, Journal of Climate, 21,
5820-5834, doi:10.1175/2008JCLI2194.1, 2008.

McCartney, M.S.: The subtropical recirculation of mode waters. J. Mar. Res., 40 (Suppl.), 427-464, 1977.

McDougall, T.J.: Neutral surfaces in the ocean: implications for modelling, Geophysical Research Letters, 14, 8, 97-800, 1987.

McNeil, B.I., Tilbrook, B. and Matear, R.J.: Accumulation and uptake of anthropogenic $CO_2$ in the Southern Ocean, south of Australia between 1968 and 1996, Journal of Geophysical Research, 106, C12, 31431-31445, 2001.



Mehrbach, C., Culberson, C.H., Hawley, J.E. and Pytkowicz, R.M.: Measurement of the apparent dissociation constants of carbonic acid in seawater at atmospheric pressure, Limnol. Oceanogr., 18, 897–907, doi:10.4319/lo.1973.18.6.0897, 1973.

Metzl, N., Tilbrook, B. and Poisson, A.: The annual fCO2 cycle and the air–sea CO2 flux in the sub-Antarctic Ocean, Tellus, 51B, 849–861, 1999.

Moore, J.K. and Abbott, M.R.: Phytoplankton chlorophyll distributions and primary production in the Southern Ocean, J. Geophys. Res., 105, C12, 28709-28722, 2000.

Moore, J.K., Abbott, M.R. and Richman, J.R.: Location and dynamics of the Antarctic Polar Front from satellite sea surface temperature data, J. Geophys. Res., 104, 3059–3073, 1999.

Morrow, R., Donguy, J.-R., Chaigneau, A. and Rintoul, S.R.: Cold-core anomalies at the subantarctic front, south of Tasmania,
Deep-Sea Res. I, 51, 1417-1440, 2004.

Mosby, H.: The waters of the Atlantic Antarctic Ocean, Scientific Results of the Norwegian Antarctic Expeditions 1927-1928, 1, 11, 131 pp, 1934.

Murata, A., Kumamoto, Y., Watanabe, S. and Fukasawa, M.: Decadal increases of anthropogenic CO2 in the South Pacific subtropical ocean along 32_S, J. Geophys. Res., 112, C05033, doi:10.1029/2005JC003405, 2007.

Najjar, R.: The dark side of marine carbon, Nature geoscience, 2, 603-604, doi: 10.1038/NGEO812, 2009.

Nelson, D.M., Ahern, J.A. and Herlihy, L.J.: Cycling of biogenic silica within the upper water column of the Ross Sea, Mar. Chem, 35, 461-476, 1991.

Nelson, D.M., et al.: Production and dissolution of biogenic silica in the ocean: Revised global estimates, comparison with regional data and relationship to biogenic sedimentation, Global Biogeochemical Cycle, 9, 3, 359-372, 1995.

Oke, P.R., et al.: Evaluation of a near-global eddy-resolving ocean model, Geosci. Model Dev., 6, 591–615, doi:10.5194/gmd-6-591-2013, 2013.

Orr, J.C., et al.: Anthropogenic ocean acidification over the twenty-first century and its impact on calcifying organisms, Nature, 437, 681-686, doi:10.1038/nature04095, 2005.

Orsi, A.H., Smethie Jr., W.M. and Bullister, J.L.: On the total input of Antarctic waters to the deep ocean: A preliminary
estimate from chlorofluorocarbon measurements, J. Geophys. Res., 107, C8, 3122, doi: 10.1029/2001JC000976, 2002.

Orsi, A.H., Whitworth III, T. and Nowlin, W.D.: On the meridional extent and fronts of the Antarctic Circumpolar Current. Dep Sea Research I, 42, 5, 641–673, 1995.

Oschlies A. and Schartau, M.: Basin-scale performance of a locally optimized marine ecosystem model, Journal of Marine Research, 63, 2, 335-358, 2005.

Pahlow, M. and Riebesell, U.: Temporal trends in deep ocean Redfield ratios, Science, 287, 831–833, 2000.

Pardo, P.C., Vázquez-Rodríguez, M., Pérez, F.F. and Ríos, A.F.: $CO_2$ air-sea disequilibrium and preformed alkalinity in the Pacific and Indian Oceans calculated from subsurface layer data, Journal of Marine Systems 84, 67–77, doi:10.1016/j.jmarsys.2010.08.006, 2011.



Pardo, P.C., Pérez, F.F., Khatiwala, S. and Ríos, A.F.: Anthropogenic $CO_2$ estimates in the Southern Ocean: Storage partitioning in the different water masses, Progress in Oceanography, 120, 230–242, doi:10.1016/j.pocean.2013.09.005, 2014.

Pardo, P.C., Perez, F.F., Velo, A. and Gilcoto, M.: Water masses distribution in the Southern Ocean: improvement of an extended OMP (eOMP) analysis. Progress in Oceanography 103, 92–105, doi: 10.1016/j.pocean.2012.06.00, 2012.

Peña-Molino, B., Rintoul, S.R. and Mazloff, M.R.: Barotropic and baroclinic contributions to along-stream and across-stream transport in the Antarctic Circumpolar Current, J. Geophys. Res. Oceans, 119, 8011–8028, doi:10.1002/2014JC010020, 2014.

Poole, R. and Tomczak, M.: Optimum multiparameter analysis of the water mass structure in the Atlantic Ocean thermocline, Deep Sea Res., Part I, 46, 1895– 1921, doi:10.1016/S0967-0637(99)00025-4, 1999.

Purkey, S.G. and Johnson, G.C.: Global Contraction of Antarctic Bottom Water between the 1980s and 2000s, Journal of Climate, 25, 5830-5844, doi:10.1175/JCLI-D-11-00612.1, 2012.

Redfield, A.: On the proportions of organic derivatives in sea water and their relation to the composition of plankton, In Daniel, R.J. (ed James Johnstone Memorial Volume). University Press of Liverpool, 177–192, 1934.

Redfield, A.: The biological control of chemical factors in the environment, Am. Sci., 46, 205–221, 1958.

Ridgway, K.R.: Seasonal circulation around Tasmania: An interface between eastern and western boundary dynamics, J. Geophys. Res., 112, C10016, doi:10.1029/2006JC003898, 2007.

Rintoul, S.R.: On the origin and influence of Adelie Land Bottom Water, Ocean, Ice, and Atmosphere: Interactions at the Antarctic continental margin Antarctic Research Series, 75, 151-171, 1998.

Rintoul, S.R. and Bullister, J.L.: A late winter hydrographic section from Tasmania to Antarctica, Deep –Sea Research I, 46, 1417-1454, 1999.

Rintoul, S.R. and England, M.H.: Ekman Transport Dominates Local Air–Sea Fluxes in Driving Variability of Subantarctic Mode Water, Journal of Physical Oceanography, 32, 1308-1321, 2002.

Rintoul, S.R. and Sokolov, S.: Baroclinic transport variability of the Antarctic Circumpolar Current south of Australia (WOCE repeat section SR3), J. Gephys. Res., 106, C2, 2815-2832, 2001.

Rintoul, S.R. and Trull, T.W.: Seasonal evolution of the mixed layer in the Subantarctic Zone south of Australia, Journal of Geophysical Research, 106, C12, 31447-31462, 2001.

Rintoul, S.R., Donguy, J.R. and Roemmich, D.H.: Seasonal evolution of upper ocean thermal structure between Tasmania and Antarctica, Deep-Sea Research I, 44, 1, 1185-1202, 1997.

Sabine, C.L., et al.: The Oceanic Sink for Anthropogenic CO2, Science, 305, 5682, 367–371, 2004.

Sabine, C.L., et al.: Decadal changes in Pacific carbon, J. Geophys. Res., 113, C07021, doi:10.1029/2007JC004577, 2008.

Sallée, J.B., Matear, R.J., Rintoul, S.R. and Lenton, A.: Localized subduction of anthropogenic carbon dioxide in the Southern Hemisphere oceans, Nature, 5, 579-584, doi: 10.1038/NGEO1523, 2012.

Sallée, J.B., Speer, K. and Morrow, R.: Response of the Antarctic Circumpolar Current to Atmospheric Variability, Journal of Climate, 21, 3020-3039, doi:10.1175/2007JCLI1702.1, 2008.



Sallée, L.B., Speer, K.G. and Rintoul, S.R.: Zonally asymmetric response of the Southern Ocean mixed-layer depth to the Southern Annular Mode, 3, 273-279, 2010.

Sallée, J.-B., Wienders, N., Speer, K. and Morow, R.: Formation of subantarctic mode water in the southeastern Indian Ocean, Ocean Dynamics, 56, 525–542, doi:10.1007/s10236-005-0054-x, 2006.

Sambrotto, R.N. and Mace, B.J.: Coupling of biological and physical regimes across the Antarctic Polar Front as reflected by nitrogen production and recycling, Deep-Sea Res. II, 47, 3339-3367, 2000.

Sarmiento, J.L. and Sundquist, E.T.: Revised budget for the oceanic uptake of anthropogenic carbon dioxide, Nature, 356, 589-593, 1992.

Sarmiento, J.L., Gruber, N., Brzezinski, M.A. and Dunne, J.P.: High-latitude controls of thermocline nutrients and low latitude
biological productivity. Nature, 427, 56–60, 2004.

Sarmiento, J.L., Hughes, T.M.C., Stouffer, R.J. and Manabe, S.: Simulated response of the ocean carbon cycle to anthropogenic climate warming, Nature, 393, 245-249, 1998.

Sedwick, P.N., et al.: Limitation of algal growth by iron deficiency in the Australian Subantarctic region, Geophys. Res. Lett., 26, 18, 2865-2868, 1999.

Shadwick, E. H., et al.: Seasonality of biological and physical controls on surface ocean $CO_2$ from hourly observations at the Southern Ocean Time Series site south of Australia, Global Biogeochem. Cycles, 29, doi:10.1002/2014GB004906, 2015.

Sloyan, B.M. and Rintoul, S.R.: The Southern Ocean Limb of the Global Deep Overturning Circulation, Journal of Physical Oceanography, 31, 143-173, 2001.

Sloyan, B.M., Ridgway, K. and Cowley, R.: The East Australian Current and Property Transport at 27°S from 2012 to 2013,
Journal of Physical Oceanography, 46, 993-1008, DOI: 10.1175/JPO-D-15-0052.1, 2016.

Sokolov, S. and Rintoul, S.R.: Circulation and water masses of the southwest Pacific: WOCE Section P11, Papua New Guinea to Tasmania, Journal of Marine Research, 58, 223–268, 2000.

Sokolov, S. and Rintoul, S.R.: Structure of Southern Ocean fronts at 140°E, Journal of Marine Systems, 37, 151-184, 2002.

Sokolov, S. and Rintoul, S.R.: Multiple jets of the Antarctic Circumpolar Current South of Australia, Journal of Phys. Ocean.
37, 1394-1412, doi: 10.1175/JPO3111.1, 2007.

Sokolov, S. and Rintoul, S.R.: Circumpolar structure and distribution of the Antarctic Circumpolar Current fronts: 1.Mean circumpolar paths, Journal of Geophysical Research, 114, C11018, doi:10.1029/2008JC005108, 2009.

Speer, K., Rintoul, S.R. and Sloyan, B.: The Diabatic Deacon Cell, Journal of Physical Oceanography, 30, 3212-3222, 2000.

Speich, S., et al.: Tasman leakage: A new route in the global ocean conveyor belt, Geophysical Research Letters, 29, 10, 1416,
10.1029/2001GL014586, 2002.

Talley, L.D.: Closure of the global overturning circulation through the Indian Pacific, and Southern Oceans: Schematics and transports, Oceanography 26, 1, 80-97, http://dx.doi.org/10.5670/oceanog.2013.07, 2013.

Talley, L.D., Pickard, G.L., Emery, W.J. and Swift, J.H.: Descriptive Physical Oceanography: An Introduction (Sixth Edition), Elsevier, Boston, 560 pp, 2011.





Thacker, W.C.: Regression-based estimates of the rate of accumulation of anthropogenic CO2 in the ocean: A fresh look, Marine Chemistry, 132–133, 44–55, doi:10.1016/j.marchem.2012.02.004, 2012.

Thompson, D.W.J. and Solomon, S.: Interpretation of Recent Southern Hemisphere Climate Change, Science, 296, 895-899, doi:10.1126/science.1069270, 2002.

Thompson, R.O. and Edwards, R.J., 1981. Mixing and water-mass formation in the Australian Subantarctic. Journal of Physical Oceanography 11 (10), 1399–1406, doi: 10.1175/1520-0485(1981) 011<1399:MAWMFi>2.0.CO;2 , 1981.

Tomczak, M.: A multi-parameter extension of temperature/salinity diagram techniques for the analysis of non-isopycnal mixing, Progress in Oceanography, 10, 147–171, 1981.

Tomczak, M. and Large, D.G.B.: Optimum Multiparameter Analysis of Mixing in the Thermocline of the Eastern Indian Ocean

Journal of Geophysical Research, 94, Cll, 16141-16149, 1989.

Tréguer, P.J.: The Southern Ocean silica cycle, C. R. Geoscience, 346, 279–286, 10.1016/j.crte.2014.07.003, 2014.

Treguer, P., et al.: The Silica Balance in the World Ocean: A Reestimate, Science, 268, 5209, 375-379, doi: 10.1126/science.268.5209.375, 1995.

Trull, T., Rintoul , S.R., Hadfield, M. and Abraham, E.R.: Circulation and seasonal evolution of polar waters south of Australia:

Implications for iron fertilization of the Southern Ocean, Deep-Sea Rs. II, 2439-2466, 2001a.

Trull, T.W., et al.: Moored sediment trap measurements of carbon export in the Subantarctic and Polar Frontal Zones of the Southern Ocean, south of Australia, J. Geophys. Res., 106, C12, 489-31,509, 2001b.

van Heuven, S., Pierrot, D., Rae, J.W.B., Lewis, E. and Wallace, D.W.R.: MATLAB Program Developed for $CO_2$ System Calculations, 2011.

van Wijk, E. M. and Rintoul, S.R.: Freshening drives contraction of Antarctic Bottom Water in the Australian Antarctic Basin, Geophys. Res. Lett., 41, 1657–1664, doi:10.1002/2013GL058921, 2014.

Vázquez-Rodríguez, M., Padín, X.A., Pardo, P.C., Ríos, A.F. and Pérez, F.F.: The subsurface layer reference to calculate preformed alkalinity and air-sea $CO_2$ disequilibrium in the Atlantic Ocean. Journal of Marine Systems 94, 52–63, doi:10.1016/j.jmarsys.2011.10.008, 2012.

Verdy, A., Dutkiewicz, S., Follows, M.J., Marshall, J. and Czaja, A.: Carbon dioxide and oxygen fluxes in the Southern Ocean: Mecahnisms of Interannual variability, Global Biogeochemical Cycles, 21, GB2020, doi:10.1029/2006GB002916, 2007.

Verlencar, X.N., Somasunder, K. and Qasim, S.Z.: Regeneration of nutrients and biological productivity in Antarctic waters, Marine Ecology Progress Series, 61, 41-59, 1990.

Wang, X., Matear, R.J. and Trull, T.W.: Nutrient utilization ratios in the Polar Frontal Zone in the Australian sector of the

Southern Ocean: A model, Global Biogeochem. Cycles, 17,1, 1009, doi:10.1029/2002GB001938, 2003.

Wassmann, P., Vernet, M., Mitchell, B.G. and Rey, F.: Mass sedimentation of Phaeocystis pouchetii in the Barents Sea, Marine Ecology Progress Series, 66, 183-195, 1990.

Waters, J.F., Millero, F.J. and Sabine, C.L.: Changes in South Pacific anthropogenic carbon, Global Biogeochem. Cycles, 25, GB4011, doi:10.1029/2010GB003988, 2011.



Weiss, R.: Carbon dioxide in water and seawater: the solubility of a non-ideal gas, Mar. Chem., 2, 203–215, doi::10.1016/0304-4203(74)90015-2, 1974.

Whitworth III, T. and Nowlin Jr., W.D.: Water masses and currents of the Southern Ocean at the Greenwich meridian, Journal of Geophysical Research, 92, C6, 6462-6476, 1987.

Whitworth III, T., Orsi, A.H., Kim, S.-J. and Nowlin, W.D.: Water masses and mixing near the Antarctic Slope Front. Antarctic Research Series, 75, 1–27, 1998.

Williams, N.L., et al.: Quantifying anthropogenic carbon inventory changes in the Pacific sector of the Southern Ocean, Marine Chemistry, 174, 147–160, doi: /10.1016/j.marchem.2015.06.015, 2015

Wong, A.P.S., Bindoff, N.L. and Church, J.A.: Large-scale freshening of intermediate waters in the Pacific and Indian oceans, Nature, 400, 440-443, 1999.

Xue, L., Gao, L., Cai, W.-J., Yu, w. and Wei, M.: Response of sea surface fugacity of $CO_2$ to the SAM shift south of Tasmania: Regional differences, Geophys. Res. Lett., 42, 3973–3979, doi:10.1002/2015GL063926, 2015.

Zickfeld, K., et al.: Response of the global carbon cycle to human-induced changes in Southern Hemisphere winds, Geophysical Research Letters, 34, L12712, doi:10.1029/2006GL028797, 2007.

Zickfeld, K., Fyfe, J.C., Eby, M. and Weaver, A.J.: Comment on "Saturation of the Southern Ocean $CO_2$ Sink Due to Recent Climate Change", Science 319, 5863, 570b, doi: 10.1126/science.1146886, 2008.





**B Figures**

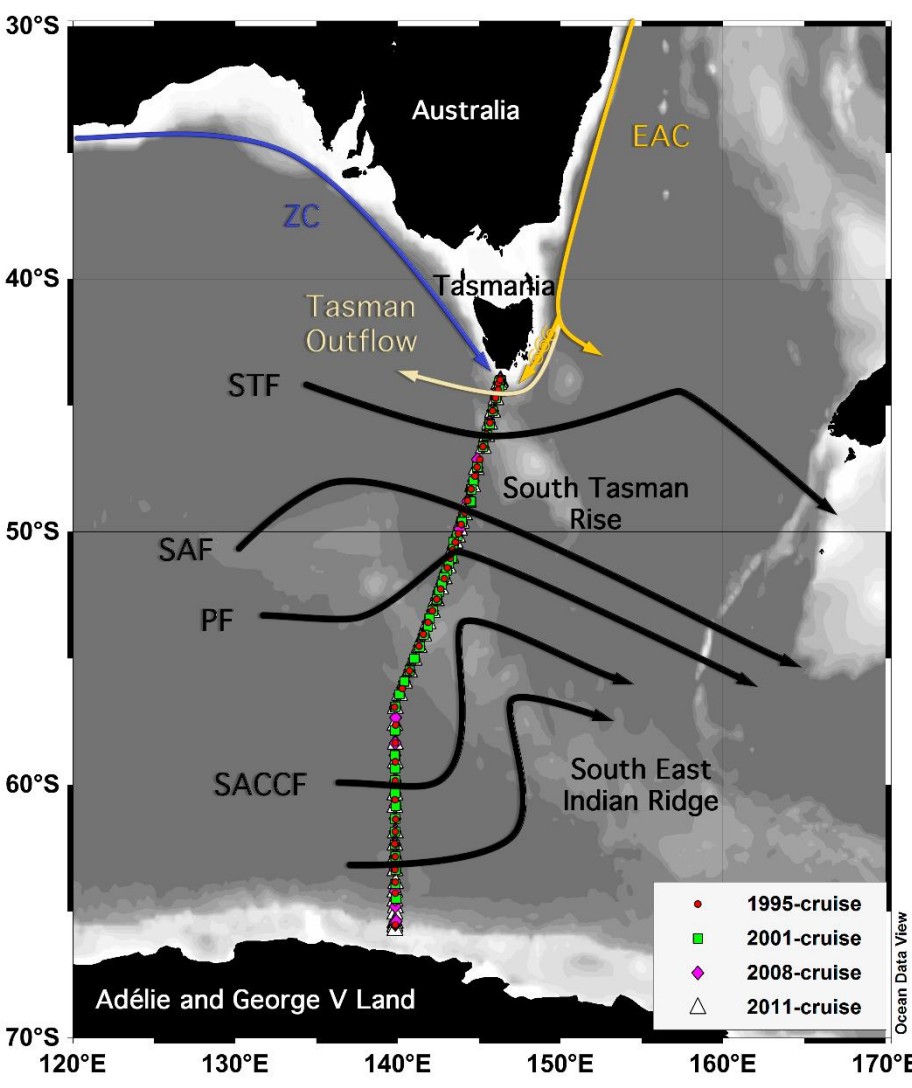

**Fig. 1. Transects of the 4 summer repeats of the G0-SHIP hydrodynamic line SR03 in the Southern Ocean south of Tasmania for the period 1995 – 2011 and main hydrography features of the region. ZC = Zeehan Current. EAC = East Australian Current. Black arrows indicate the flow of the Antarctic Circumpolar Current (ACC), with STF = Subtropical Front; SAF = Sub-Antarctic Front,**
**PF = Polar Front and SACCF = Southern ACC Front.**

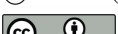



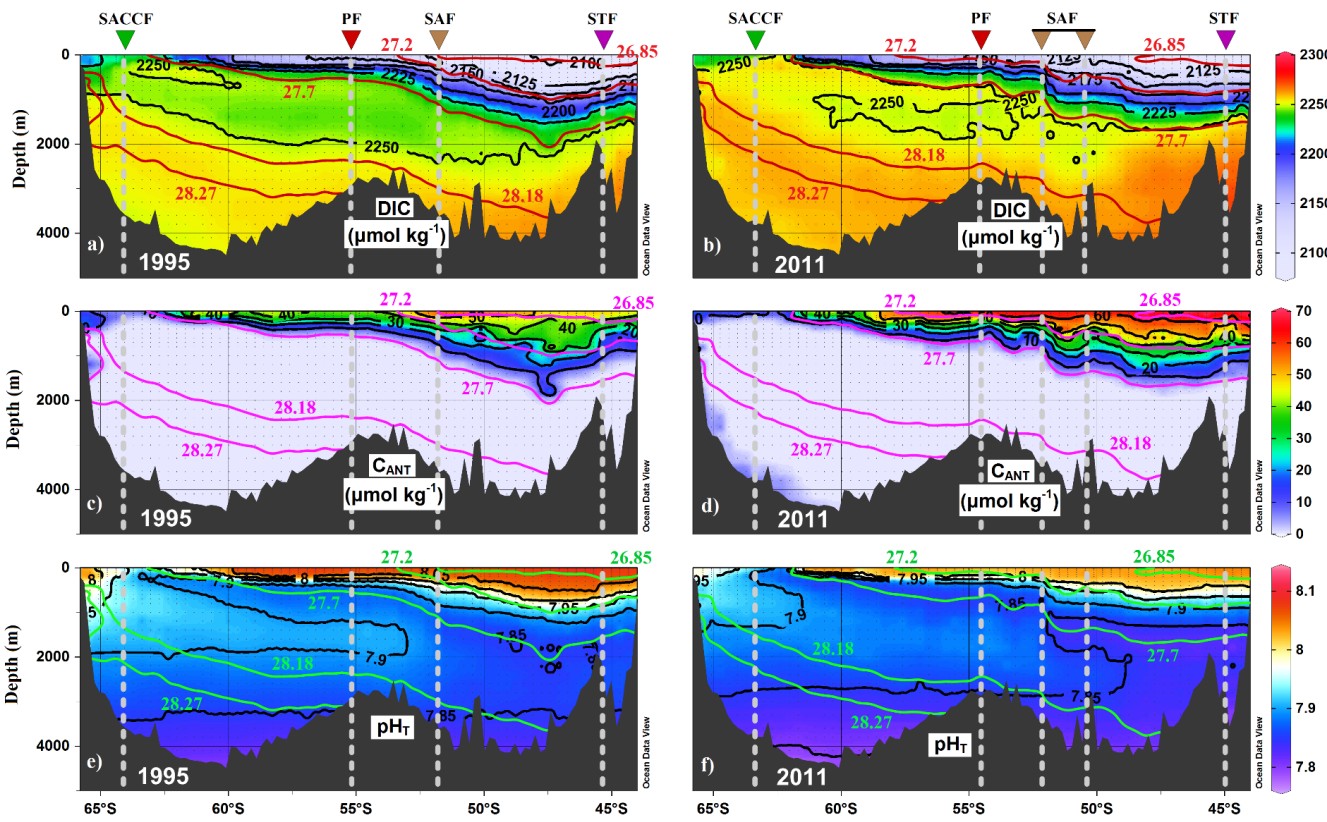

5    **Fig. 2. Distribution of DIC (a and b), C$_{ANT}$ (c and d) and pH$_T$ (e and f) in the SR03 section south of Tasmania for the years 1995 (a,
c, e) and 2011 (b, d, f). Red lines in plots (a) and (b), pink lines in plots (c) and (d) and green lines in plots (e) and (f) indicate the
neutral surfaces that define the different water masses ($\Upsilon^n$). The position of the fronts (coloured triangles and grey dotted lines) in
each of the cruises is also shown.**





**Fig. 3. Aragonite Saturation Depth (ASD, $\Omega_{Ar} = 1$) for the 1995 and 2011 cruises. (a) Distribution of ASD (blue dotted and solid lines) and $\Upsilon^n$ (green and yellow lines) in latitude in the SR03 section. (b) section zoomed for the first 1500 m of the water column. The position of the fronts (triangles) in 1995 (yellow) and 2011 (green) is also shown. The blue palette in the background indicates the distribution of $\Omega_{Ar}$ for the 2011-cruise.**




**Fig. 4.** Distribution of (a and b) dissolved inorganic carbon (DIC) and (c and d) dissolved oxygen ($O_2$) in the 1995 and 2011 cruises for deep-bottom layers of the section. The location of the Sub-Antarctic Front (SAF) is also shown as well as the neutral surfaces ($\Upsilon^n$, white lines) limiting the UCDW, LCDW and AABW layers.



## C Tables

| EXPOCODE | Dates | | | REF. |
|---|---|---|---|---|
| 09AR19941213 | 20/12/1994 | - | 1/02/1995 | 1995 |
| 09AR20011029 | 29/10/2001 | - | 11/12/2001 | 2001 |
| 09AR20080322 | 23/03/2008 | - | 15/04/2008 | 2008 |
| 09AR20110124 | 4/01/2011 | - | 31/01/2011 | 2011 |

**Table 1. Repeats of the GO-SHIP SR03 line on board the Aurora Australis from 1995 to 2011 with the expocode from GLODAPv2 database.**

| Layer | Definition | Complete Name | Reference |
|---|---|---|---|
| STCW | $\Upsilon^n < 26.85$ & $S \geq 34.3$ | Subtropical Central Water | Rintoul (1998) |
| AASW | $\Upsilon^n < 27.7$ & $S < 34.3$ | Antarctic Surface Water | Rintoul (1998); Williams et al. (2015) |
| AASW$_{upw}$ | $\Upsilon^n \geq 27.7$ & $depth \leq 300m$ | Antarctic Surface Water | -------------------------- |
| SAMW | $26.85 \leq \Upsilon^n < 27.2$ & $S \geq 34.3$ | Sub-Antarctic Mode Water | Rintoul (1998); Rintoul and Bullister (1999) |
| AAIW | $27.2 \leq \Upsilon^n < 27.7$ & $S \geq 34.3$ | Antarctic Intermediate Water | Rintoul and Bullister (1999) |
| UCDW | $27.7 \leq \Upsilon^n < 28.18$ & $depth > 300m$ | Upper Circumpolar Deep water | Williams et al. (2015) |
| LCDW | $28.18 \leq \Upsilon^n < 28.25$ | Lower Circumpolar Deep Water | Lacarra et al. (2011); Williams et al. (2015) |
| AABW | $\Upsilon^n \geq 28.25$ | Antarctic Bottom Water | Rintould and Bullister (1999); Williams et al. (2015) |

**Table 2. Definition of the water mass layers between neutral surfaces ($\Upsilon^n$) and referenced used to accordingly decide the limits of the layers. We also consider limits in salinity, depth and position of the ACC fronts (see Table 3) to differentiate better some of the layers. PF = Polar Front.**

| Cruise | STF | SAF | PF | SACCF |
|---|---|---|---|---|
| 1995 | ~45.4°S | ~51.8°S | ~55.2°S | ~63.8°S |
| 2001 | ~46.4°S | ~50.1°S | ~54.5°S | ~63.7°S |
| 2008 | ~46.4°S | ~52.2°S | ~54.6°S | ~63.5°S |
| 2011 | ~45°S | ~52.16-50.5°S * | ~54.5°S | ~63.4°S |

**Table 3. Location of the ACC fronts south of Tasmania for each of the cruises following the definitions of Sokolov and Rintoul (2002) for hydrographic data.**



| Layer | $\partial DIC/\partial t$ (µmol kg$^{-1}$ yr$^{-1}$) 1995-2011 | $\partial C_{AN}/\partial t$ (µmol kg$^{-1}$ yr$^{-1}$) 1995-2011 | $\partial SiO_4/\partial t$ (µmol kg$^{-1}$ yr$^{-1}$) 1995-2011 | $\partial pH_T/\partial t$ (yr$^{-1}$) 1995-2011 | $\partial \Omega_{Ar}/\partial t$ (yr$^{-1}$) 1995-2011 | % $\partial \Omega_{Ar}/\partial t$ (% yr$^{-1}$) 1995-2011 |
|---|---|---|---|---|---|---|
| STCW | 0.86 ± 0.07, RMSE=7 | [0.71 ± 0.08 - 0.93 ± 0.08] | ----- | -0.0027 ± 0.0001, RMSE=0.01 | -0.009 ± 0.001, RMSE=0.15 | -0.42 ± 0.17 |
| AASW | 0.85 ± 0.14, RMSE=21 | [0.35 ± 0.14 - 0.85 ± 0.21] -------------------------- * [0.35 ± 0.14 - 0.65 ± 0.21] | ----- | -0.0035 ± 0.0002, RMSE=0.03 | -0.009 ± 0.001, RMSE=0.14 | -0.61 ± 0.19 |
| AASW$_{upw}$ | 0.61 ± 0.10, RMSE=9 | [0 - 0.41 ± 0.16] -------------------------- * [0 - 0.21 ± 0.16] | 0.36 ± 0.06, RMSE=6 | -0.0015 ± 0.0004 RMSE=0.04 | ----- | ----- |
| SAMW | 1.10 ± 0.14, RMSE=16 | 0.92 ± 0.09, RMSE=10 | ----- | -0.0031 ± 0.0003 RMSE=0.03 | -0.011 ± 0.001, RMSE=0.17 | -0.67 ± 0.20 |
| AAIW | 0.40 ± 0.15, RMSE=24 | 0.42 ± 0.06, RMSE=10 | ----- | -0.0017 ± 0.0002 RMSE=0.03 | ----- | ----- |
| UCDW | 0.29 ± 0.02, RMSE=5 | ----- | 0.22 ± 0.04, RMSE=12 | -0.0012 ± 0.0001 RMSE=0.03 | ----- | ----- |
| LCDW | 0.20 ± 0.02, RMSE=4 | ----- | 0.27 ± 0.02, RMSE=4 | -0.0012 ± 0.0002 RMSE=0.03 | ----- | ----- |
| AABW | 0.24 ± 0.02, RMSE=2 | ** 0.07 ± 0.01, RMSE=2 | 0.15 ± 0.05, RMSE=11 | ----- | ----- | ----- |

**Table 4. Trends in the water mass layers for the period 1995-2011 of dissolved inorganic carbon ($\partial DIC/\partial t$), anthropogenic carbon ($\partial C_{ANT}/\partial t$), silicate ($\partial SiO_4/\partial t$) total pH ($\partial pH_T/\partial t$), aragonite saturation ($\partial \Omega_{Ar}/\partial t$) and % of change in the aragonite saturation (% $\partial \Omega_{Ar}/\partial t$). RMSE = root mean square error. * Trends in C$_{ANT}$ for the AASW and AASW$_{upw}$ layers considering an approximate value**

5 **for the increase in DIC due to the advection of old deep waters to the section (see section 4.4.1 in the text). ** The value of $\frac{\partial C_{ANT}}{\partial t}$ in the AABW layer is considered negligible because it falls below the accuracy of the back-calculation method.**

| Layer | $\partial C_{ANT\_BC}/\partial t$ (µmol kg$^{-1}$ yr$^{-1}$) 1995-2011 | $\partial DIC^{BIO}/\partial t$ (µmol kg$^{-1}$ yr$^{-1}$) 1995-2011 | $\partial DIC^{\pi}/\partial t$ (µmol kg$^{-1}$ yr$^{-1}$) 1995-2011 |
|---|---|---|---|
| STCW | 1.05 ± 0.05, RMSE=5 | 0.22 ± 0.08, RMSE=8 | -0.34 ± 0.06, RMSE=6 |
| AASW | 0.71 ± 0.06, RMSE=9 | 0.50 ± 0.16, RMSE=23 | -0.36 ± 0.13, RMSE=17 |
| AASW$_{upw}$ | ----- | 0.41 ± 0.16, RMSE=28 | ----- |

**Table 5. Trends of anthropogenic carbon from estimates of the back-calculation method and of the terms DIC$^{BIO}$ and DIC$^{\pi}$ (see**

10 **section 3.2 of the text). RMSE = root mean square error.**





| Layers | Thickness (m) | $C_{ANT}$ storage rates (mol m$^{-2}$ yr$^{-1}$) |
|---|---|---|
| STCW | 230 | [0.17 ± 0.06 - 0.22 ± 0.08] |
| AASW | 280 | [0.10 ± 0.04 - 0.24 ± 0.07] |
| AASW$_{upw}$ | 275 | [0 - 0.11 ± 0.04] |
| SAMW | 450 | 0.43 ± 0.14 |
| AAIW | 900 | 0.39 ± 0.14 |
| SR03 section | --- | 0.30 ± 0.24 |

Table 6. Approximated rates of $C_{ANT}$ storage based on the trends from Table 4.

| Layer | $\partial C_{AN}/\partial t$ (µmol kg$^{-1}$ yr$^{-1}$) 1995-2011 | $\Delta C_{ANT}$ (Two-regression) (µmol kg$^{-1}$ yr$^{-1}$) 1995-2011 | $\Delta C_{ANT}$ (eMLR) (µmol kg$^{-1}$ yr$^{-1}$) 1995-2011 |
|---|---|---|---|
| STCW | [0.71 ± 0.08 - 0.93 ± 0.08] | 0.72 ± 0.04,  RMSE=5 | 1.21 ± 0.09,  RMSE=3 |
| AASW | [0.35 ± 0.14 - 0.85 ± 0.21] | 0.61 ± 0.04,  RMSE=6 | 0.67 ± 0.10,  RMSE=10 |
| AASW$_{upw}$ | [0 - 0.41 ± 0.16] | 0.20 ± 0.03,  RMSE=3 | 0.30 ± 0.04,  RMSE=3 |
| SAMW | 0.92 ± 0.09,  RMSE=10 | 0.94 ± 0.05,  RMSE=5 | 0.96 ± 0.09,  RMSE=10 |
| AAIW | 0.42 ± 0.06,  RMSE=10 | 0.28 ± 0.02,  RMSE=4 | 0.43 ± 0.02,  RMSE=7 |
| UCDW | ----- | 0.09 ± 0.01,  RMSE=2 | 0.15 ± 0.09,  RMSE=3 |
| LCDW | ----- | 0.08 ± 0.01,  RMSE=2 | 0.28 ± 0.15,  RMSE=3 |
| AABW | ** 0.07 ± 0.01,  RMSE=2 | 0.09 ± 0.01,  RMSE=1 | 0.18 ± 0.08,  RMSE=2 |

5    Table 7. Comparison between $\frac{\partial C_{ANT}}{\partial t}$ from the present study and the values of $\Delta C_{ANT}$ from the two-regression and eMLR methods. RMSE= root mean square error. ** The value of $\frac{\partial C_{ANT}}{\partial t}$ in the AABW layer is considered negligible because it falls below the accuracy of the back-calculation method.

## D Figures & Tables Appendix

| | θ | S | $SiO_4$ | $NO_3^0$ | $PO_4^0$ | $O_2^{0*}$ | $TA^0$ | $DIC^{\pi}_{SAT}$ | CDIS | $CDIS^{\pi}$ | Fractions uncertainties (%) |
|---|---|---|---|---|---|---|---|---|---|---|---|
| $SWT_{STW16}$ | 16 | 35.1 | 0.9 | 1.2 | 0.04 | 243 | 2290 | 1990 | -19 | 1 | 0.04 |
| $SWT_{STW15}$ | 15 | 35.66 | 0.6 | 0.0 | 0.12 | 247 | 2328 | 2026 | -22 | -2 | 0.04 |
| $SWT_{AASW}$ | -1.85 | 33.8 | 45 | 30.7 | 2.10 | 360 | 2289 | 2137 | -23 | -19 | 0.06 |
| $SWT_{SASW}$ | 5 | 33.8 | 3 | 23.3 | 1.55 | 310 | 2264 | 2064 | -13 | -4 | 0.06 |
| $SWT_{HSSW}$ | -1.91 | 34.71 | 80 | 28.3 | 2.02 | 300 | 2351 | 2188 | -21 | 0 | 0.08 |
| $SWT_{SAMW}$ | 8.8 | 34.63 | 6 | 13.2 | 0.92 | 280 | 2290 | 2053 | -10 | 2 | 0.03 |
| $SWT_{AAIW}$ | 4 | 34.35 | 34 | 29.2 | 1.97 | 220 | 2299 | 2099 | -16 | -6 | 0.04 |
| $SWT_{NADW}$ | 3.28 | 34.91 | 28 | 27.5 | 1.19 | 220 | 2355 | 2152 | -27 | -10 | 0.08 |
| $SWT_{CDW}$ | 0.65 | 34.707 | 115 | 30.8 | 2.12 | 220 | 2351 | 2168 | -23 | -2 | 0.03 |
| $SWT_{PIDW}$ | 1.44 | 34.75 | 125 | 34.2 | 3.40 | 96 | 2360 | 2168 | -24 | -4 | 0.04 |
| $SWT_{AABW}$ | -0.6 | 34.66 | 130 | 30.7 | 2.13 | 259 | 2355 | 2181 | -22 | -2 | 0.05 |
| **Weights** | 20 | 10 | 0.5 | 1 | 1 | 1 | | | | | |
| $r^2$ | 0.99 | 0.99 | 0.98 | 0.99 | 0.99 | 0.99 | | | | | |

**Table A1. Properties of the SWTs characterizing the water masses of the SR03 section. All SWTs are defined with preformed values of the variables (°). *The values of preformed oxygen ($O_2°$) are not in equilibrium for end members representing waters from the Antarctic shelf or old deep waters. The uncertainties in the fractions of the SWTS, the weights given to each variable in the OMP analysis and the square correlation coefficient ($r^2$) between the observed values and the OMP estimates are also listed.**

| Mixing Groups |
|---|
| STW15 + SAMW + SASW + STW16 |
| SASW + SAMW + AAIW + AASW |
| SAMW + AAIW + NADW |
| AAIW + PIDW + NADW |
| AASW + AAIW + CDW |
| CDW + AABW + HSSW + AASW |

**Table A2. Mixing groups used in the OMP analysis. Note that only the subscripts of the names of the SWTs are written, i.e., STW15 instead of $SWT_{STW15}$.**




**Table A3. Parameterizations for estimating $AT^0$ and CDIS in each of the SWTs. PO = $R_P PO_4 + O_2$. For SWT$_{HSSW}$ *, CDIS is obtained from the NOAA database of latitudinal mean values of atmospheric $CO_2$. For SWT$_{NADW}$**, $AT^0$ is obtained from the climatology of GLODAPv2 and CDIS from Pardo et al. (2014).**

| SWTs | Parameterizations | Reference |
|---|---|---|
| SWT$_{STW16}$ | $AT^\circ$ ($\pm6$) = 2288.3 + 62.8*(S - 35) - 0.9*($\theta$ -16) + 0.1*(PO - 300)<br>CDIS ($\pm5$) = -47.9 + 2.31*$\theta$ + 0.16*(PO - 300) | Parameterizations for the South Subtropical region-Pacific Ocean (SSTIP, Pardo et al., 2011) |
| SWT$_{STW15}$ | $AT^\circ$ ($\pm6$) = 2288.3 + 62.8*(S - 35) - 1.6*($\theta$ -16) + 0.1*(PO - 300)<br>CDIS ($\pm5$) = 51.3 + 2.31*$\theta$ + 0.16*(PO - 300) | Parameterizations for the South Subtropical region-Indian Ocean (SSTIP, Pardo et al., 2011) |
| SWT$_{HSSW}$*<br><br>SWT$_{AASW}$<br><br>SWTS$_{SASW}$ | $AT^\circ$ ($\pm4$) = 2296.7 + 94.7*(S - 35) + 0.3*(PO - 300)<br>CDIS ($\pm5$) = $-84.3 - 12.95$*(S - 35) + 5.75*$\theta$ + 0.17*(PO - 300) | Parameterizations for the Antarctic region (AAIP, Pardo et al., 2011)<br>*(Dlugokencky, et al., 2016) |
| SWT$_{SAMW}$<br><br><br>SWT$_{AAIW}$ | $AT^\circ$[1] ($\pm6$) = 2288.3 + 62.8*(S - 35) -(0.9//1.6)*($\theta$ - 16) + 0.1*(PO - 300)<br>$AT^\circ$[2] ($\pm4$) = 2296.7 + 94.7*(S - 35) + 0.3*(PO - 300)<br>CDIS[1] ($\pm5$) = -47.9 + 2.31*$\theta$ + 0.16*(PO - 300)<br>CDIS[2] ($\pm5$) = $-84.3 - 12.95$*(S - 35) + 5.75*$\theta$ + 0.17*(PO - 300) | Parameterizations for the South Subtropical regions and for the Antarctic region (SSTIP, Pardo et al., 2011) |
| SWT$_{NADW}$ | --------------------------------- | GLODAPv2 // Pardo et al. (2014) ** |
| SWT$_{CDW}$ | $AT^\circ$ // CDIS = 0.69*($AT^\circ$ // CDIS)$_{AABW}$ + 0.26*($AT^\circ$ // CDIS)$_{NADW}$ + 0.05*($AT^\circ$ // CDIS)$_{AAIW}$ | --------------------------------- |
| SWT$_{PIDW}$ | $AT^\circ$ // CDIS = 0.71*($AT^\circ$ // CDIS)$_{CDW}$ + 0.25*($AT^\circ$ // CDIS)$_{NADW}$ + 0.04*($AT^\circ$ // CDIS)$_{AAIW}$ | --------------------------------- |
| SWT$_{AABW}$ | $AT^\circ$ // CDIS = 0.51*($AT^\circ$ // CDIS)$_{CDW}$ + 0.44*($AT^\circ$ // CDIS)$_{HSSW}$ + 0.05*($AT^\circ$ // CDIS)$_{AASW}$ | --------------------------------- |





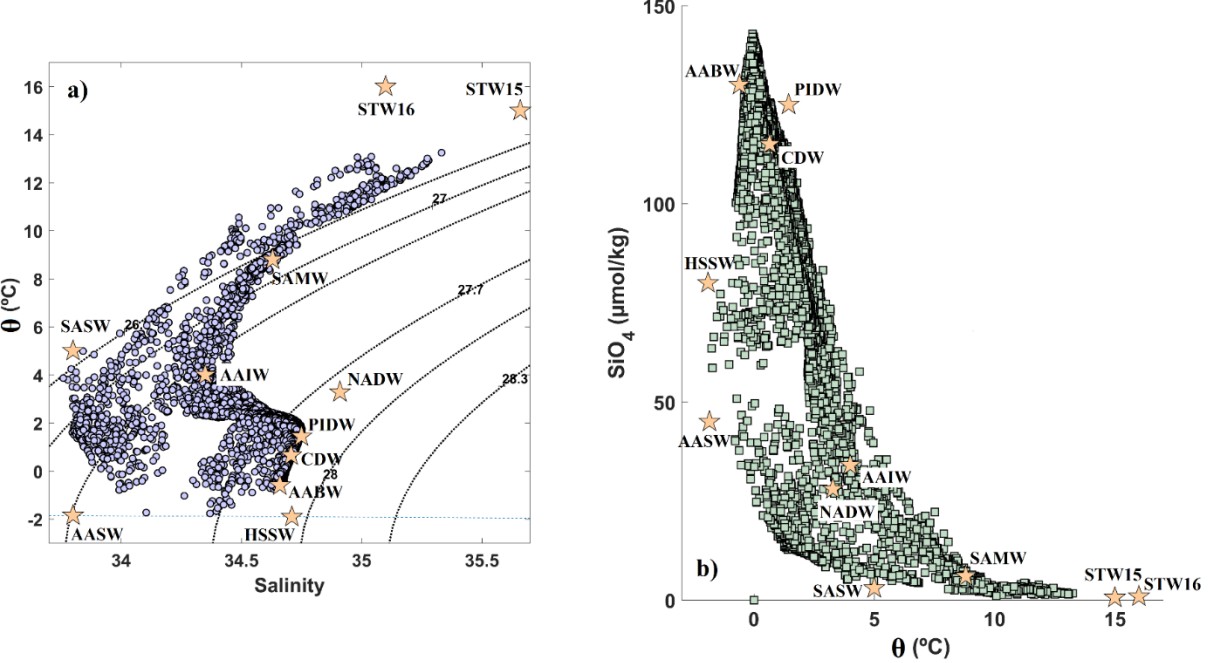

Fig. A1. (a) T/S diagram, with θ = potential temperature. Blue dots represent the data from all cruises for the period 1995-2011. Dotted lines indicate density (σ₀, kg m⁻³). (b) SiO₄/T diagram, with θ = potential temperature. Green squares represent the data from all cruises for the period 1995-2011. Pentagrams represents the SWTs included in the OMP analysis (only subscripts are written, i.e., AASW instead of SWT_AASW).