# Peer review of "Carbon uptake and biogeochemical change in the Southern Ocean, south of Tasmania"

_Biogeosciences, 2017_

## Referee Comment (RC1) · Anonymous Referee #1 · 7 Aug 2017

This is an important manuscript presenting data from repeat sections in the Southern Ocean south of Australia. The data analysis is comprehensive and the results are interesting and seem to be robust; they can be compared with other studies from the Southern Ocean, thus completing the view of this vital ocean region. Actually, I think there are even a few more studies that have recently appeared and addressing similar issues in other areas of the Southern Ocean, which the authors could incorporate in the discussion of the results.

The authors explain their results through the intensification of winds due to changes in the SAM. Their repeats cover the time period 1995 to 2011. However, they also write, almost at the beginning of the Discussion: "Several studies have reported a trend in the SAM toward its positive phase from the 1960s until the 2000s (Thompson and

[Figure]

Solomon, 2002; Marshall, 2002, 2003; Lenton and Matear, 2007; Sallée et al., 2008)."
During at least the second half of the time period, the SAM has not been in its positive
phase anymore, and thus there will probably not be elevated winds anymore, which will
not enhance upwelling. I encourage the authors to identify this and add comments to
the manuscript.

Although all cruises used for the analysis were conducted around summer, they were
not in the same month. Actually, data may be 3-4 months apart. Certainly, in the deeper
water masses, this will not have a big effect on the results. However, for the surface
and sub-surface layers the seasonal changes in biologically-mediated properties are
large and thus this is likely to have an effect on the computed rates. I think this caveat
should be treated in the manuscript. Please comment on this and analyze the possible
and expected effects on the results.

P1, line 11: Is this the correct symbol for neutral density (also at other places in the
manuscript)?

P2, line 6 " . . . and ultimately upwell close to the Antarctic Shelf" I think this only holds
for part of this water, and possibly not even the major part. Please change the wording
to take that into account.

P2, line 11 I suggest a modified sentence: Within the eastward flow of the ACC major
water exchange between the three ocean basins takes place.

P2, line 16 Because of twice the word "that" in this sentence I suggest: . . . water mass
properties and this may complicate . . .

P2, line 19 I think it is fair to cite older work of observationalists here, which actually
laid the basis for this knowledge.

P3, line 12-13 Change to . . . reported an increase in CANT uptake . . .

P3, line 16 delete "of the"

P3, line 23 Change to . . . one of the most revisited sections in the Southern Ocean.

P3, line 23-25 This sentence is an anacoluthon. Please correct.

P3, line 31 This concerns surface waters, I presume. Please add that term.

P4, line 30 Is there a reference for this?

P5, line 13-15 You only give the precision for DIC and TA measurements. Please also supply the accuracy, which is much more important here. It should be less good than the precision.

P6, line 2 combined with and

P7, line 12 . . . defined by their ÏŠn condition (Table 2).

P8, line 8-9 "Nevertheless, long-term trends in O2 due to circulation and remineralization processes have not yet been reported." This is not correct. See:

Matear, R. J., A. C. Hirst, and B. I. McNeil (2000), Changes in dissolved oxygen in the Southern Ocean with climate change, Geochem. Geophys. Geosyst., 1, 1050, doi:10.1029/2000GC000086. (cited in the present manuscript)

van Heuven SMAC, Hoppema M, Jones EM, de Baar HJW, 2014. Rapid invasion of anthropogenic CO2 into the deep circulation of the Weddell Gyre. Phil. Trans. R. Soc. A 372: 20130056. http://dx.doi.org/10.1098/rsta.2013.0056

The last paragraph of section 5 is clearly a conclusion, and should thus be moved to the Conclusions section.

P15, line 13 delete one "repeat"

P16, line 8 uses stepwise MLR (delete "and")

P18, lines 16-17 This sentence is incomplete.

P23, line 8 When using data from GLODAPv2, please cite GLODAPv2 manuscript,

[Figure]

Olsen et al 2016 ESSD.

P24, line 31 delete info near end of line

P25, line 24 Deep-Sea (hyphen)

P27, line 6 Deep-Sea (hyphen)

P27, line 22 add NCAR technical note

P27, line 26 Law et al. as shown here is the Discussions paper. There is also a final paper in Geosci. Model Dev. from 2017.

P32, line 11 Comptes Rendus Geoscience

P32, line 26 Mechanisms (typo)

Table 2 caption: references (typo)

PF is defined in the caption but does not occur in Table 2

---

## Referee Comment (RC2) · M. Álvarez (Referee) · 8 Aug 2017

**REVIEW of "Carbon uptake and biogeochemical change in the Southern Ocean, south of Tasmania" by Paula C. Pardo et al.**

This manuscript (MS) is a very nice piece of work dealing with real biogeochemical data in the Southern Ocean south of Tasmania. As an biogeochemical observationist I do really appreciate high quality sustained ocean observations in harsh regions as the SO. Maintaining the funding for such expensive long term programs is always difficult. State agencies always prefer quick, 3-4 years projects with high impact results for society or policy makers. The slow science done with time series, either fixed or oceanographic lines is precious to detect global change in a comprehensive way (Henson, 2014, dx.doi.org/10.1098/rsta.2013.0334).

After the nice words, **I do conclude that this MS should be accepted but before, it needs some MINOR improvements**, they regard to two main points organization and data & calculations.

**With regard to the organization of the MS:**

- the resolution or uncertainty of the back-calculation method for CANT is given in page 16, line 16. It should be given in section 3.2.

- I would suggest including section 6 as Supplementary material, a slight change in the organization / order of this section, the general title is OK for me, but the sections would be:

  - CANT estimation

    - OMP analysis

    - parameterization for $TA^0$ and CDIS

    - biogeochemical model estimating $\delta$CDIS, please give numbers for this term, as far as I understand is not given in Table A1.

  - DIC changes

    - CANT estimated with other methods

    - circulation and biological processes at steady state: here I have doubts & thoughts, the methodology to calculate DIC changes and attribute them to any oceanographic process (CANT increase, change in circulation, warming, higher upwelling,..) should be clarified in the main MS. On one side, the BC method for CANT assumes steady state in the circulation / biology (= constant stoichiometric ratios), the main reason behind is that biology activity or ocean circulation/mixing is not affected by the CANT increase. On the other side, when talking about DIC is completely untrue that is not affected by changes in circulation (higher/lower upwelling, higher/lower transport of water masses). Any change on circulation would be detected in AOU (as nutrients are always more problematic due to precision and

exactitude issues). But also in transient tracers that you do not show (this is an important caveat of the whole analysis!!). Please clarify in this section which processes affect CANT and which DIC and how your method deals with them.

- changes in the rates of export of POC: POC export is related to primary production in the upper layer, higher POC production implies higher primary production. If this POC is mineralized AOU would increase and DIC increase as well, only if we keep the same circulation. An intensification of circulation for the same POC input would mean a lower AOU as bacteria would have less time to work. I insist that without transient tracers distangling the influence on DIC from circulation and biology is difficult.

- stochiometric ratios

- the title of the MS is "carbon uptake and biogeochemical change", so I understand is not only CANT changes, it is mainly about DIC changes and the processes causing them, of course, one of them the CANT increase. I do follow section 3.2, but I do not follow section 3.3. In fact I think section 3.3 should be introduced in first order compared to section 3.2. And I question in the current section 3.3 the explanations in page 8 to estimate CANT / DIC changes and the reasoning behind. It seems that two types of CANT are in the ocean, the back calculated and another one with something else. IT seems as well that CANT changes could also contain DIC changes related to AOU (circulation/biology) which is clearly separated in section 6. As section 3.3 is difficult to follow the results in Tables 4 & 5 too. I will comment more about this issue in the data & calculation section of this review.

- I do miss some figure with the temporal evolution of DIC & CANT & DIC$^{BIO}$ at the different layers.

- Table 6 should be included in the corresponding section of the Sup information if changes!

**With regard to the data & calculations:**
- Cruise data: please state that those cruises were included in GLODAPv2 and therfore checked, DIC for the 2008 was corrected. Please confirm that there is no transient tracer data (CFCs or/and SF6). It is very very surprising to report oxygen accuracy and precision as 1%, please correct me if I am wrong, so for O2 in deep bottom water, 230 umol/kg.. the

estimation error would be 2.3 umol/kg, it means an error in DICBIO of 1.6 umol/kg. Quite high.

- CO2SYS calculation: please state that you calculate pHT and omega Ar in situ from DIC and TA. Which is the borate constant?.

- Table 3: which is the meaning of "*" for the 2011 cruise in the SAF region?

- CANT estimation: either you include here current section 6.1 or you keep the CANT-OMP method as the only one, and the others in the Supl. material
With regard to the CANT-OMP method, the reference paper is Pardo et al. (2014), I suggest to use the same notation. Your Eq 1 defining CANT-OMP is different from Eq 4 in Pardo2014. I always understood CANT as the difference in preformed conditions from now (or whenever) to preindustrial times. So what is the meaning of $DIC^{\pi}$ in current Eq1, it should be $DIC^{0\,\pi}$.
Current Eq 3 is also different from Eq 3 in Pardo2014. The MS $DIC^{\pi}$ is different from $DIC^{0}{}^{\pi}$??. However current Eq 4 equals Eq4 in Pardo2014.
TAble 1 in Pardo2014 and Table A1 & A3 here should be comparable. Maybe in Table 1 2014 there is a typo $\Delta Cdis^{\pi}$ should just be $\Delta Cdis$.I am very confused about the CANT-OMP calculation for the disequilibrium term and the info given in Table 1 here and that in Pardo2014. Please clarify. And please give in the section the error for CANT-OMP.
The reference for Thacker 2012 is not included in the references.
Silicate is included as a non-conservative variable in the OMP, but no Ratio is given. I think is more coherent to write $\Delta O_2$ in the section A.2.
In TAble A1 the analytical error, $\epsilon$, is not given, page 21 line 6.

- DIC changes: as suggested previously I suggest to start the methods sections with the proposed estimation of DIC changes and driving factors.

$$\Delta DIC = \Delta DIC^{BIO} + \Delta DIC^{CANT} + \Delta DIC_{phys}$$

please see Álvarez et al. (doi:10.1029/2010JC006475, 2011)
$\Delta DIC$ with your real measurements
$\Delta DIC^{BIO}$ will be computed as your Eq2, this term contains changes in mineralization associated or not with changes in circulation / biology BUT correlated with AOU / nutrients for sure. As you lack of transient tracers information you cannot account for changes in the transport, circulation independently.
$\Delta DIC^{CANT}$ with the OMP method (forget about the dis terms )

$\Delta DIC^{phys}$ as the difference between DIC- $DIC^{CANT}$-$DIC^{BIO}$, this term would contain any changes in DIC not associated with AOU, mixing / Ventilation.... not accounted for in the CANT-OMP.

Following this methodology the blurred assumptions in page 8, lines 1 to 22 are avoid. and of course I think it would help to explain the results.

I hope to have been helpful.

---

## Author Response (AR1)

The code **Rx.y** was applied to connect the comments of the reviewers with the changes done in the manuscript. **x** is the number of the reviewer and **y** is the number of the comment of the review **x** associated to a change in the manuscript.

**Author's answers to Anonymous Referee #1**

This is an important manuscript presenting data from repeat sections in the Southern Ocean south of Australia. The data analysis is comprehensive and the results are interesting and seem to be robust; they can be compared with other studies from the Southern Ocean, thus completing the view of this vital ocean region.

Thank you very much for your words.

Actually, I think there are even a few more studies that have recently appeared and addressing similar issues in other areas of the Southern Ocean, which the authors could incorporate in the discussion of the results.

We provide comparisons to many-recent works (12 studies published between 2007 and 2017, 9 of them since 2014). If the referee recommends comparison to specific works beyond these, we are open to consider it.

**1)** The authors explain their results through the intensification of winds due to changes in the SAM. Their repeats cover the time period 1995 to 2011. However, they also write, almost at the beginning of the Discussion: "Several studies have reported a trend in the SAM toward its positive phase from the 1960s until the 2000s (Thompson and Solomon, 2002; Marshall, 2002, 2003; Lenton and Matear, 2007; Sallée et al., 2008)." During at least the second half of the time period, the SAM has not been in its positive phase anymore, and thus there will probably not be elevated winds anymore, which will not enhance upwelling. I encourage the authors to identify this and add comments to the manuscript.

The referee is correct. The SAM does not present a constant positive phase from 2000. However, it does exhibit considerable interannual variability, including, for the years 2008 and 2011 (two last occupations of SR03), relatively high SAM index values comparable to those at the end of the positive trend (https://climatedataguide.ucar.edu/climate-data/marshall-southern-annular-mode-sam-index-station-based), indicating strong winds over the region. This is shown in Figure A2, which will be added in the supplementary material:

[Figure]

*Fig. A2. Seasonal values of the observation-based SAM index. The smooth black curve shows decadal variations. Figure obtained from Marshall, Gareth & National Center for Atmospheric Research Staff (Eds). "The Climate*

*Data Guide: Marshall Southern Annular Mode (SAM) Index (Station-based)." Retrieved from https://climatedataguide.ucar.edu/climate-data/marshall-southern-annular-mode-sam-index-station-based.*

Lines 27-29 of the Abstract will be corrected to:

*"From all our results, we conclude a scenario of increased transport of deep waters into the section and enhanced upwelling at high latitudes for the period between 1995 and 2011 linked to strong westerly winds."*

Line 23 in the Discussion section will be changed to:

*" ... linked to strong westerly winds."*

At the end of the first paragraph of the Discussion section we'll add:

*"From the 2000s on, the SAM index no longer presents a positive trend but, although exhibiting considerable interannual variability (Fig. A2 in the supplementary material), the SAM index remains in its positive phase, favouring strong winds over the region."*

**R1.2)** Although all cruises used for the analysis were conducted around summer, they were not in the same month. Actually, data may be 3-4 months apart. Certainly, in the deeper water masses, this will not have a big effect on the results. However, for the surface and sub-surface layers the seasonal changes in biologically-mediated properties are large and thus this is likely to have an effect on the computed rates. I think this caveat should be treated in the manuscript. Please comment on this and analyze the possible and expected effects on the results.

The effect of the seasonal variability is implicitly included in the errors of the trends and strongly reflected in the RMSE values within each water mass. That is why we only show statistically significant trends.

To clarify further, we'll change lines 14-15 of page 7 (section 3.3) to:

*"We show the value of the root mean square error (RMSE or square root of the variance of the residuals), which can be interpreted in large part as unexplained variance caused by short-time scale processes including the different seasonal timings of the cruises. RMSE has the same units as the response variable."*

**R1.3)** P1, line 11: Is this the correct symbol for neutral density (also at other places in the manuscript)?

No, it's not. Thanks for noticing. The epsilon will be changed to a gamma throughout the reviewed manuscript.

**R1.4)** P2, line 6 " : : : and ultimately upwell close to the Antarctic Shelf" I think this only holds for part of this water, and possibly not even the major part. Please change the wording to take that into account.

The referee is right. We'll modify the text in page 2 lines 5-6 to:
*"...and ultimately upwell between the Southern ACC Front and the Polar Front."*

**R1.5)** P2, line 11 I suggest a modified sentence: Within the eastward flow of the ACC major water exchange between the three ocean basins takes place.

Thanks. We accept this modification.

**R1.6)** P2, line 16 Because of twice the word "that" in this sentence I suggest: : : : water mass properties and this may complicate : : :

Thanks. We'll correct this.

**R1.7)** P2, line 19 I think it is fair to cite older work of observationalists here, which actually laid the basis for this knowledge.

You are right. Two more references will be added:
        " Sabine et al., 2004; Gruber et al., 2009"

**R1.8)** P3, line 12-13 Change to : : : reported an increase in CANT uptake : : :

Thanks. We'll correct this.

**R1.9)** P3, line 16 delete "of the"

Thanks. We'll correct this.

**R1.10)** P3, line 23 Change to : : : one of the most revisited sections in the Southern Ocean.

Thanks. We'll correct this.

**R1.11)** P3, line 23-25 This sentence is an anacoluthon. Please correct.

Sorry for that. The sentence will recover the missing verb:

        "Trends in oxygen ($O_2$), nutrients, and the carbon system parameters, i.e., DIC, total alkalinity (TA), anthropogenic carbon ($C_{ANT}$), total pH ($pH_T$) and % aragonite saturation ($\Omega_{Ar}$) were estimated for the period 1995-2011, when both DIC and TA measurements are available."

**R1.12)** P3, line 31 This concerns surface waters, I presume. Please add that term.

Yes, you are right. We'll change line 31 in page 3 (section 2) to:

        "separates warm, salty subtropical surface waters from cooler and fresher sub-Antarctic surface waters."

**R1.13)** P4, line 30 Is there a reference for this?

Yes, it was included in the previous phrase. The reference (*Rintoul, 1998*) will be added in this phrase as well.

**R1.14)** P5, line 13-15 You only give the precision for DIC and TA measurements. Please also supply the accuracy, which is much more important here. It should be less good than the precision.

Thanks for noticing. We'll change the phrase to:

*"The precisions and accuracy of DIC and TA measurements improved slightly on more recent sections, and for all sections were better than ± 2 µmol kg⁻¹, for both variables, based on analysis of duplicate samples and certified reference material".*

**R1.15)** P6, line 2 combined with and

Thanks. We'll correct this.

**R1.16)** P7, line 12 : : : defined by their ÏŠn condition (Table 2).

Thanks. We'll add this.

**R1.17)** P8, line 8-9 "Nevertheless, long-term trends in O2 due to circulation and remineralization processes have not yet been reported." This is not correct. See:
Matear, R. J., A. C. Hirst, and B. I. McNeil (2000), Changes in dissolved oxygen in the Southern Ocean with climate change, Geochem. Geophys. Geosyst., 1, 1050, doi:10.1029/2000GC000086. (cited in the present manuscript)
van Heuven SMAC, Hoppema M, Jones EM, de Baar HJW, 2014. Rapid invasion of anthropogenic CO2 into the deep circulation of the Weddell Gyre. Phil. Trans. R. Soc. A 372: 20130056. http://dx.doi.org/10.1098/rsta.2013.0056

We agree with the referee that this phrase is not correct and that our summary of previous work on this issue was not complete. The intention was to indicate that statistically significant trends *different from zero* in surface waters had not been reported, rather than that no investigations had occurred. With respect to the two studies mentioned, we accordingly note that neither found statistically significant trends in surface waters (except in the deepest layer of waters in the Weddell Sea in Van Heuven et al., 2014).

In order to clarify this, we'll change the text in 3.3 to (new text is underlined):

*"The term $\frac{\partial DIC^{BIO}}{\partial t}$ can be influenced by changes with time of alkalinity due to changes in the rate of carbonate precipitation/dissolution and of AOU due to changes in the rate of remineralization and in circulation. In the present study only surface waters of the SR03 section present changes $DIC^{BIO}$ between 1995 and 2011. Numerous studies have reported a strong influence of biological communities in the seasonal cycle of dissolved $O_2$ in surface waters (Bender et al., 1996; Moore and Abbott, 2000; Sambrotto and Mace, 2000; Trull et al., 2001a). Interannual variability in $O_2$ in upper layers of the Southern Ocean have also been related to changes in the entrainment of deeper waters into the mixed layer due to the mixed layer depth variability (Matear et al., 2000; Verdy et al., 2007; Sabine et al., 2008; Sallée et al., 2012). Although some studies found long-term decreases in $O_2$ due to circulation in deep waters of the Weddell Sea (van Heuven et al., 2014\*) and for the first 1000 m of the global ocean (Helm et al. 2011\*\*), significant long-term trends in $O_2$ due to circulation and remineralization processes have not yet been reported for surface waters of the Southern Ocean."*

*\*van Heuven, S., Hoppema, M., Jones, E.M., de Baar, H.J.W.: Rapid invasion of anthropogenic $CO_2$ into the deep circulation of theWeddell Gyre. Phil. Trans. R. Soc. A 372: 20130056. http://dx.doi.org/10.1098/rsta.2013.0056, 2014.*
*\*\* Helm, K. P., Bindoff, N.L., and Church, J.A.: Observed decreases in oxygen content of the global ocean, Geophys. Res. Lett., 38, L23602, doi:10.1029/2011GL049513, 2011.*

We'll also add some comments in section 5, (after first paragraph in page 13) in order to acknowledge previous efforts:

*"In terms of the change in oxygen, Helm et al. (2011) found an average decrease in the concentration of $O_2$ between 100 and 1000 m from 1970 to 1992 of ~ -0.23 µmol $l^{-1}$ for the Southern Ocean (27% of the estimated global average change, -0.93 ± 0.23 µmol $l^{-1}$). Considering the volume of the first 1000 m of the water column of the Southern Ocean to be 19400 · $10^{-9}$ l (obtained using ETOPO1 doi:10.7289/V5C8276M) and the volume of the first 1000 m of the SR03 section to be 2700 · $10^{-9}$ l, the decrease of $O_2$ found by Helm et al. (2011), if constant in time, would correspond to a decrease of ~ -1.7 µmol $l^{-1}$ $yr^{-1}$. We only found changes in oxygen within the surface water mass layers (STCW, AASW and $AASW_{upw}$) that approximately fill the first 300 m of the water column of the SR03. Then, the decrease of ~ -1.7 µmol l-1 would correspond to an average change of $O_2$ of ~ -0.32 µmol kg-1 yr-1 for surface waters of the SR03. This means that values of ~ 0.20 µmol kg-1 yr-1 due to circulation processes can be expected in $\frac{\partial DIC^{BIO}}{\partial t}$ for surface waters, which is comparable to the average of our findings (Table 5), 0.32 ± 0.24 µmol kg-1 yr-1 and could indicate that the change in $O_2$ is related to circulation processes."*

**R1.18)** The last paragraph of section 5 is clearly a conclusion, and should thus be moved to the Conclusions section.

We agree. Thanks. We will move this paragraph.

**R1.19)** P15, line 13 delete one "repeat"

Thanks. We'll correct this.

**R1.20)** P16, line 8 uses stepwise MLR (delete "and")

Thanks. We'll correct this.

**R1.21)** P18, lines 16-17 This sentence is incomplete.

We'll complete the sentence:

*"A battery of OMP analyses were done with varying values of $R_N$ between 9 and 10 in increments of 0.2, $R_P$ between 120 and 145 in increments of 5, and $R_{Si}$ between 0 and 8 in increments of 2."*

**R1.22)** P23, line 8 When using data from GLODAPv2, please cite GLODAPv2 manuscript, Olsen et al 2016 ESSD.

The references will be added:
*Key et al., 2015; Olsen et al., 2016* in section 3.1
*Lauvset et al., 2016* in section A.3 page 23

**R1.23)** P24, line 31 delete info near end of line

Thanks. We'll correct this.

**R1.24)** P25, line 24 Deep-Sea (hyphen)

Thanks. We'll correct this.

**R1.25)** P27, line 6 Deep-Sea (hyphen)

Thanks. We'll correct this.

**R1.26)** P27, line 22 add NCAR technical note

Thanks. We'll correct this.

**R1.27)** P27, line 26 Law et al. as shown here is the Discussions paper. There is also a final paper in Geosci. Model Dev. from 2017.

The reference will be changed to Law et al. (2017):

*"Law, R. M. et al.: The carbon cycle in the Australian Community Climate and Earth System Simulator (ACCESS-ESM1) Part 1: Model description and pre-industrial simulation, Geosci. Model Dev., 10, 2567–2590, https://doi.org/10.5194/gmd-10-2567-2017, 2017."*

**R1.28)** P32, line 11 Comptes Rendus Geoscience

Thanks. We'll correct this.

**R1.29)** P32, line 26 Mechanisms (typo)

Thanks. We'll correct this.

**R1.30)** Table 2 caption: references (typo)

Thanks. We'll correct this.

**R1.31)** PF is defined in the caption but does not occur in Table 2

Thanks. We'll correct this.

**Author's answers to Referee #2 (Dr. Marta Álvarez)**

This manuscript (MS) is a very nice piece of work dealing with real biogeochemical data in the Southern Ocean south of Tasmania. As an biogeochemical observationist I do really appreciate high quality sustained ocean observations in harsh regions as the SO. Maintaining the funding for such expensive long term programs is always difficult. State agencies always prefer quick, 3-4 years projects with high impact results for society or policy makers. The slow science done with time series, either fixed or oceanographic lines is precious to detect global change in a comprehensive way (Henson, 2014, dx.doi.org/10.1098/rsta.2013.0334).

Thank you very much for your comment. We really appreciate it and the very thorough review, which has added to the paper.

After the nice words, I do conclude that this MS should be accepted but before, it needs some MINOR improvements, they regard to two main points organization and data & calculations.

With regard to the organization of the MS:
- the resolution or uncertainty of the back-calculation method for CANT is given in page 16, line 16. It should be given in section 3.2.

The uncertainty of $C_{ANT\_BC}$ was already mentioned in section 3.2 (page 6, line 4).

- I would suggest including section 6 as Supplementary material, a slight change in the organization / order of this section, the general title is OK for me, but the sections would be:

Thank you very much for the advice. Nevertheless, we consider that section 6 should stay as it is for two main reasons:
        - When talking about $C_{ANT}$ it is always good to have some reference values from other methodologies.
        - Explaining the sensitivity of the results completes the manuscript and explores the reliability of the results.

Besides, the scheme proposed for the inclusion of section 6 in the supplementary material does not change too much the structure of section 6 as it is right now.

The numerical model configuration, the OMP analysis and the particularities of the AT0 and *CDIS* estimates should be in the supplementary material to make the reading in the main text easier. In fact, we think that it could be better to upload this supplementary material in a separate file.

Before continuing, there are some points that we would like to clarify. There has been a misunderstanding with the methodology, mainly because we misled the sign in equation 3 and also because we need to clarify certain aspects of the methodology. Also, we think that there was a misunderstanding between this work and the previous work by Pardo et al. (2014), probably favoured by the error in the sign of Equation 3. We would like to clarify that we refer to Pardo et al. (2014) because the methodologies are similar. Since a similar methodology was applied before in the Southern Ocean we consider that it is fair to cite this reference. Nevertheless, we do not redirect the reader to that

work because we prefer to explain the methodology again in order to help readers not familiar with back-calculation methods. We will try to clarify this in the following answers.

We answered the comments as they were made, i.e., following the proposed organization. We did not answer those comments that are just a title for the proposed organization scheme.

- CANT estimation

If this comment refers to moving section 3.2 to the supplementary material, we do not agree. We would like to maintain the methodology in the main text in order to help readers not familiarized with back-calculation methods.

- OMP analysis

- parameterization for TA0 and CDIS - biogeochemical model estimating $\delta CDIS$, please give numbers for this term, as far as I understand is not given in Table A1.

$\delta CDIS$ can be easily obtained through Eq. 4 (section 3.2):

$$CDIS^{\pi} = CDIS - \delta CDIS$$

and for this reason we do not list it separately (redundantly) in Table A1.

- DIC changes

- CANT estimated with other methods

**R2.1)** - circulation and biological processes at steady state: here I have doubts & thoughts, the methodology to calculate DIC changes and attribute them to any oceanographic process ($C_{ANT}$ increase, change in circulation, warming, higher upwelling,..) should be clarified in the main MS.

We agree with Dr. Álvarez in that there is a need to clarify more the estimates of the change in DIC explained in section 3.3. We will add the next text after the first paragraph of section 3.3:

*"Respect to the total change of DIC ($\frac{\partial DIC}{\partial t}$), our goal is to disentangle the effects that solubility, circulation, biology and $C_{ANT}$ uptake have on the variability of DIC. The total change of DIC ($\frac{\partial DIC}{\partial t}$) in a water mass is due to changes in the atmosphere-ocean exchange, biological processes and circulation processes. In order to account for the change in DIC due the atmosphere-ocean exchange and biological processes, we compare $\frac{\partial DIC}{\partial t}$ to $\frac{\partial C_{ANT\_BC}}{\partial t}$ and $\frac{\partial DIC^{BIO}}{\partial t}$, Eq. (2). The change in DIC not explained by $\frac{\partial C_{ANT\_BC}}{\partial t}$ or $\frac{\partial DIC^{BIO}}{\partial t}$ will then be due to circulation processes.*

*In order to compare $\frac{\partial DIC}{\partial t}$ to $\frac{\partial C_{ANT\_BC}}{\partial t}$ we need to consider how the change in $\frac{\partial DIC^{BIO}}{\partial t}$ and $DIC^{\pi}$ ($\frac{\partial DIC^{\pi}}{\partial t}$) (terms of Eq (1), section 3.2) affect the changes in $C_{ANT\_BC}$ ."*

We believe that all the following comments also belong to this same point: "- circulation and biological processes at steady state". We will answered them individually following the assigned numbers.

**R2.2):**

1) On one side, the BC method for CANT assumes steady state in the circulation / biology (= constant stoichiometric ratios), the main reason behind is that biology activity or ocean circulation/mixing is not affected by the CANT increase.
2) On the other side, when talking about DIC is completely untrue that is not affected by changes in circulation (higher/lower upwelling, higher/lower transport of water masses).
3) Any change on circulation would be detected in AOU (as nutrients are always more problematic due to precision and exactitude issues).
4) But also in transient tracers that you do not show (this is an important caveat of the whole analysis!!).
5) Please clarify in this section which processes affect CANT and which DIC and how your method deals with them.

    1) We agree and we stated that the $C_{ANT\_BC}$ assumes steady state in the circulation /biology (section 3.2):
   *"Back-calculation methods assume the ocean is in steady state for dynamical and biological processes."*

    2) We agree. We never stated that the DIC is not affected by the changes in circulation. In section 3.3 we say:
   *"We assume that the changes in $DIC^{BIO}$ due to circulation do not affect the amount of DIC in the layer. This assumption is one of the caveats of the methodology, since we cannot know how much of the change in DIC is associated to changes in circulation, i.e., how much of the change in DIC is a change in non-anthropogenic DIC. We will discuss this more in section 4.4.2."*

   This comment made us realize that there is an error in the citing of section 4.4.2, which should be "*section 6.2*". We'll correct this.

   We hope that the new text clarifying which processes drive the changes in DIC (please, see answer to the point: "- circulation and biological processes at steady state") will help with this misunderstanding.

    3) We agree and had already explained this aspect of AOU (section 3.3, page 8 lines 10-11):
   *"...since part of the changes in AOU with time reflect changes in circulation that we cannot separate from those in remineralization."*

   We also agree with the comments on nutrients and, as well as the problematic precision of nutrients, we had already noted that upper ocean processes also impact their utility for trend detection:
   *"The detection of long-term trends of nutrients at upper layers of the ocean can be masked by short time scale physical processes such as changes in the mixed layer depth, mesoscale activity and advection."* (Section 6.2, page 16, line 32 and line 1 of page 17).

    4) Unfortunately, using transient tracers is beyond the scope of this manuscript for several reasons:

   - There are only CFCs available for all the cruises considered. Other tracers have been measured but not in all the cruises. $CFC_S$ alone are not the most reliable tracers when stablishing water mass ages in young surface waters, places of frequently deep convection or in bottom waters that are eventually ventilated, because of their decay in the atmosphere.

- Most importantly, our main goal in the study is to make use of most data from the SR03 and also to consider the input of AABW in terms of $C_{ANT}$.

5) We will add a new text after the first paragraph of section 3.3 for further clarification. See answer to the comment on point: "- circulation and biological processes at steady state".

**R2.3)** - changes in the rates of export of POC: POC export is related to primary production in the upper layer, higher POC production implies higher primary production. If this POC is mineralized AOU would increase and DIC increase as well, only if we keep the same circulation. An intensification of circulation for the same POC input would mean a lower AOU as bacteria would have less time to work. I insist that without transient tracers distangling the influence on DIC from circulation and biology is difficult.

We agree that this disentanglement is challenging, and accordingly in section 6.3 we stated that:
*"Our results show that the increase of DIC in mode and intermediate waters is fully explained by the uptake of atmospheric $CO_2$, which could indicate that there was no detectable change in the rate of export of POC over the 1995-2011 period."*

Moreover, we considered the change in the rate of export of POC as a possible reason for the increase of DIC in deep waters. That is why we considered the results from deep sediment traps:
*"Estimates of POC export in the SAZ (~3800 m), the SAF (~3100 m) and the PFZ (~1500 m) using moored sediment traps.."* to check how important a change in the rate of POC export could be on the DIC change found for deep water masses.

Nevertheless, we agree with the comment and we will clarify this further by changing the text at the end of the second paragraph of section 6.3 into:
*"This increase is close to the uncertainty of the total DIC increase estimated for the UCDW layer, which means that in order to generate an increase in DIC similar to that found in the UCDW layer, the rate of POC export should be ~10 times higher than the observed rates. This change should be certainly noticeable in $\frac{\partial DIC^{BIO}}{\partial t}$ in surface waters but most probably in deep waters as well, which we do not see".*

- stochiometric ratios

- the title of the MS is "carbon uptake and biogeochemical change", so I understand is not only CANT changes, it is mainly about DIC changes and the processes causing them, of course, one of them the CANT increase.

I do follow section 3.2, but I do not follow section 3.3. In fact I think section 3.3 should be introduced in first order compared to section 3.2.

This is part of the misunderstanding we were talking about before. We believe that section 3.2 is necessary to understand section 3.3, and therefore prefer not to reorder the text. We hope that with the explanations given it is clearer now.

**R2.4)** And I question in the current section 3.3 the explanations in page 8 to estimate CANT /DIC changes and the reasoning behind. It seems that two types of CANT are in the ocean,the back calculated and another one with something else. IT seems as well that CANT changes could also contain DIC changes related to AOU (circulation/biology) which is clearly separated in section 6. As section 3.3 is difficult to follow the results in Tables 4 & 5 too. I will comment more about this issue in the data & calculation section of this review.

There are not two types of $C_{ANT}$. $C_{ANT\_BC}$ is directly estimated using the back calculation method and $\frac{\partial C_{ANT}}{\partial t}$ is obtained by correcting $\frac{\partial C_{ANT\_BC}}{\partial t}$ for the effects of solubility and change in $DIC^{BIO}$. $\frac{\partial C_{ANT}}{\partial t}$ is the best approximation for the change in $C_{ANT}$ (as we already stated in section 3.3):

*"We consider the best approximation for the change in $C_{ANT}$ as a range depending on the possible effect of biology and circulation processes on $\frac{\partial DIC^{BIO}}{\partial t}$. If the value of $\frac{\partial DIC^{BIO}}{\partial t}$ is due to the variability in the remineralization rates and the change in solubility is considered, the estimate $\frac{\partial C_{ANT\_BC}}{\partial t}$ will be the lower limit of the range, (lower limit of $\frac{\partial C_{ANT}}{\partial t} = \frac{\partial C_{ANT\_BC}}{\partial t} + \frac{\partial DIC^{\pi}}{\partial t}$). For the upper limit of the range, we consider that the value of $\frac{\partial DIC^{BIO}}{\partial t}$ is due to changes in circulation and the upper limit of the range is obtained by $\frac{\partial C_{ANT}}{\partial t} = \frac{\partial C_{ANT\_BC}}{\partial t} + \frac{\partial DIC^{\pi}}{\partial t} + \frac{\partial DIC^{BIO}}{\partial t}$."*

And we also repeated this during the explanation of the results (section 4.1.1) in order to help the reader.

We will add a new figure to improve the understanding of this differences between $\frac{\partial C_{ANT\_BC}}{\partial t}$ and $\frac{\partial C_{ANT}}{\partial t}$:

[Figure]

Figure: Estimation of the ranges in $\frac{\partial C_{ANT}}{\partial t}$ from $\frac{\partial C_{ANT\_BC}}{\partial t}$, depending on the possible effect of biology and circulation processes on $\frac{\partial DIC^{BIO}}{\partial t}$ (section 3.3).

- I do miss some figure with the temporal evolution of DIC & CANT & DICBIO at the different layers.

The temporal evolution is presented in Table 4 with the errors and RMSE. The trends in the table offer the same useful information that the figures of the temporal evolution.

- Table 6 should be included in the corresponding section of the Sup information if changes!

We've decided not to change the structure of the manuscript.

With regard to the data & calculations:

**R2.5)** - Cruise data: please state that those cruises were included in GLODAPv2 and therefore checked, DIC for the 2008 was corrected. Please confirm that there is no transient tracer data (CFCs or/and SF6).

We stated that the data are included in GLODAPv2 (page 5, lines 17-18) but we did not clarify the QC done to the variables. We'll change the phrase to:

*"The section data are available through the Global Ocean Data Analysis Project (http://cdiac.ornl.gov/oceans/GLODAPv2; Key et al., 2015; Olsen et al., 2016). The original data for the different cruises were corrected following the QC recommendations in GLODAPv2."*

We already answered the comment about tracers.

**R2.6)** It is very very surprising to report oxygen accuracy and precision as 1%, please correct me if I am wrong, so for O2 in deep bottom water, 230 umol/kg.. the estimation error would be 2.3 umol/kg, it means an error in DICBIO of 1.6 umol/kg. Quite high.

We agree. It was an error. The accuracy and precision for $O_2$ are 0.3% .

**R2.7)** - CO2SYS calculation: please state that you calculate pHT and omega Ar in situ from DIC and TA. Which is the borate constant?.

We will make that clearer by changing the previous phrase (line 19 page 5):
*"DIC and TA measurements allow estimates of other variables of the dissolved $CO_2$ system."*
To:

*"Other variables of the dissolved $CO_2$ system were calculated from the DIC and TA measurements."*

And also the second sentence in the paragraph to:

*"We calculate $pH_T$ and $\Omega_{Ar}$ (from measured DIC and TA) using the CO2sys program …"*

The reference for the borate constant is: Uppström. (1974). It will be added in the text.

**R2.8)** - Table 3: which is the meaning of "*" for the 2011 cruise in the SAF region?

The explanation will be added to Table 3 caption:

*"* The range in the location of the SAF for the 2011-cruise could be related to a diversification of the PF or SAF into different jets (Sokolov and Rintoul, 2009) but also to crossing the meander of the front twice."*

- CANT estimation: either you include here current section 6.1 or you keep the CANT-OMP method as the only one, and the others in the Supl. Material

In order to understand the results we need to maintain the two different notations. $\frac{\partial C_{ANT}}{\partial t}$ is the best approximation we can offer to compare to $\frac{\partial DIC}{\partial t}$. We already explained the differences between $\frac{\partial C_{ANT}}{\partial t}$ and $C_{ANT\_BC}$ (or CANT-OMP, as you call it). We hope that adding the new Figure will help to understand this better.

With regard to the CANT-OMP method, the reference paper is Pardo et al. (2014), I suggest to use the same notation.

This is another misunderstanding. We would like to say that we use Pardo et al. (2014) as reference because the methodologies are similar but we never redirect the reader to that work. We already explained that we rewrote the methodology in order to help those readers not familiar with back-calculation methods.

Similarly to what we did before, we are going to answer the following comments using the assigned numbers.

**R2.9):**

1) Your Eq 1 defining CANT-OMP is different from Eq 4 in Pardo2014.
2) I always understood CANT as the difference in preformed conditions from now (or whenever) to preindustrial times. So what is the meaning of DICπ in current Eq1, it should be DIC 0 π.
3) Current Eq 3 is also different from Eq 3 in Pardo2014.
4) The MS DIC π is different from DIC 0 π ??. However current Eq 4 equals Eq4 in Pardo2014.
5) TAble 1 in Pardo2014 and Table A1 & A3 here should be comparable.
6) Maybe in Table 1 2014 there is a typo ΔCdis π should just be ΔCdis. I am very confused about the CANTOMP calculation for the disequilibrium term and the info given in Table 1 here and that in Pardo2014. Please clarify
7) And please give in the section the error for CANT-OMP.

1) Equation 4 in Pardo et al. (2014) is not the definition of $C_{ANT\_BC}$ (or CANT-OMP as you call it). Probably you refer to Eq. 1 in the present manuscript compared to Eq. 1 in Pardo et al. (2014). These equations are not exactly the same. Nevertheless, if you follow both texts a little, it is easy to find the similarity.

*2)* You understood correctly, CANT is the difference from preformed conditions. DICπ is always a preformed variable. The notation in Pardo et al. (2014) relates to that used by Chen (1982)*. The "origin" of CANT for a water mass is: at preindustrial times and in its region of formation, which is the surface ocean. This means that DICπ is always going to be a preformed value whatever the notation chosen. There is a small problem with the notation "0" in DICπ. While Chen (1982) does not consider the disequilibrium term we do, thus the real "preformed" term should be DICπSAT. That is why we decided to avoid the notation "0" this time. Nevertheless, to avoid more confusion, we will add "preformed" in the text when explaining the term DICπ: a) in page 6 line 17: *"...and $DIC^{\pi}$ is the preformed concentration of DIC in preindustrial times."* and b) in page 6 line 28: *"The preformed preindustrial term, $DIC^{\pi}$ ,…".*

*Chen, C.-T. A.: On the distribution of anthropogenic $CO_2$ in the Atlantic and Southern Ocean, Deep-Sea Research, 29, 5, 563-580, 1982.

3) Thank you very much for noticing. We made a mistake with the sign of Eq. 3 and it will be corrected.

$$DIC^{\pi} = DIC_{SAT}^{\pi} + CDIS^{\pi}$$

It was just a typo in the equation, the estimates were done correctly.

4) Yes, those equations are and should be the same. All is okay.

5) The tables are not equal but are comparable.

   In Pardo et al. (2014) Table 1 is shorter than Table A1 (present manuscript) because there was a previous manuscript describing the OMP analysis (Pardo et al., 2012) done for the whole Southern Ocean, which was the base for the study of CANT in Pardo et al. (2014). Nevertheless, we'll add the uncertainties of the parameters of definition of the SWTs to Table A1 in order to make them more comparable

   In the present manuscript, we changed some things in the OMP analysis with respect to Pardo et al., (2012) because this is a local study. In a local study it is really important to consider the local varieties of the water masses in order to maintain the reliability of the OMP. This is why we have to show the whole definition of the SWTs in Table A1.

   With respect to Table A3 (present manuscript) comparison to Table 1 in Pardo et al. (2014), we consider that this is just a matter of how you present the equations. In detail, the way we estimated $AT^0$ and CDIS in the current manuscript is similar but not identical to how it was done in Pardo et al. (2014).

6) There is no typo in Table 1 of Pardo et al. (2014 and $\Delta$Cdis can be estimated with the equation 4 given in Pardo et al. (2014).

7) The error for $C_{ANT\_BC}$ is already given there (page 6, line 4).

The reference for Thacker 2012 is not included in the references.

The reference is already in the previous manuscript.

**R2.10)** Silicate is included as a non-conservative variable in the OMP, but no Ratio is given.

Silicate is included as a conservative variable and no Ratio is needed. We mentioned that in section 6.4: *"The residuals of SiO4 did not change significantly for any value of $R_{Si}$ and we consider SiO4 as a conservative variable."*

But we forgot to mention it in section A.2 where it should be. Thanks for noticing. We would add this information in section A.2.

**R2.11)** I think is more coherent to write $\Delta$O2 in the section A.2.

We agree. We'll change $\Delta$O to $\Delta O_2$.

**R2.12)** In TAble A1 the analytical error, $\epsilon$, is not given, page 21 line 6.

Thanks for noticing. We'll change the table A1 to:

| | $\theta$ (°C) | S | $SiO_4$ (µmol/kg) | $NO_3^0$ (µmol/kg) | $PO_4^0$ (µmol/kg) | $O_2^{0\,*}$ (µmol/kg) | $TA^0$ (µmol/kg) | $DIC^{\pi}_{SAT}$ (µmol/kg) | CDIS (µmol/kg) | $CDIS^{\pi}$ (µmol/kg) | Fractions uncertainties (%) |
|---|---|---|---|---|---|---|---|---|---|---|---|
| $SWT_{STW16}$ | 16 ± 0.06 | 35.1 ± 0.07 | 0.9 ± 0.2 | 1.2 ± 0.2 | 0.04 ± 0.3 | 243 ± 2 | 2290 | 1990 | -19 | 1 | 0.04 |
| $SWT_{STW15}$ | 15 ± 0.06 | 35.66 ± 0.07 | 0.6 ± 0.2 | 0 ± 0.2 | 0.12 ± 0.3 | 247 ± 2 | 2328 | 2026 | -22 | -2 | 0.04 |
| $SWT_{AASW}$ | -1.85 ± 0.006 | 33.8 ± 0.005 | 45 ± 2 | 30.7 ± 0.2 | 2.10 ± 0.3 | 360 ± 4 | 2289 | 2137 | -23 | -19 | 0.06 |
| $SWT_{SASW}$ | 5 ± 0.008 | 33.8 ± 0.03 | 3 ± 0.2 | 23.3 ± 0.3 | 1.55 ± 0.5 | 310 ± 3 | 2264 | 2064 | -13 | -4 | 0.06 |
| $SWT_{HSSW}$ | -1.91 ± 0.08 | 34.71 ± 0.006 | 80 ± 1 | 28.3 ± 0.08 | 2.02 ± 0.03 | 300 ± 3 | 2351 | 2188 | -21 | 0 | 0.08 |
| $SWT_{SAMW}$ | 8.8 ± 0.02 | 34.63 ± 0.03 | 6 ± 0.6 | 13.2 ± 0.6 | 0.92 ± 0.8 | 280 ± 7 | 2290 | 2053 | -10 | 2 | 0.03 |
| $SWT_{AAIW}$ | 4 ± 0.01 | 34.35 ± 0.02 | 34 ± 2 | 29.2 ± 0.4 | 1.97 ± 0.9 | 220 ± 8 | 2299 | 2099 | -16 | -6 | 0.04 |
| $SWT_{NADW}$ | 3.28 ± 0.008 | 34.91 ± 0.003 | 28 ± 1 | 27.5 ± 0.3 | 1.19 ± 0.7 | 220 ± 4 | 2355 | 2152 | -27 | -10 | 0.08 |
| $SWT_{CDW}$ | 0.65 ± 0.006 | 34.707 ± 0.003 | 115 ± 7 | 30.8 ± 0.1 | 2.12 ± 0.1 | 220 ± 3 | 2351 | 2168 | -23 | -2 | 0.03 |
| $SWT_{PIDW}$ | 1.44 ± 0.008 | 34.75 ± 0.005 | 125 ± 3 | 34.2 ± 0.01 | 3.4 ± 0.02 | 96 ± 2 | 2360 | 2168 | -24 | -4 | 0.04 |
| $SWT_{AABW}$ | -0.6 ± 0.006 | 34.66 ± 0.006 | 130 ± 5 | 30.7 ± 0.08 | 2.13 ± 0.03 | 259 ± 3 | 2355 | 2181 | -22 | -2 | 0.05 |
| Weights | 20 | 10 | 0.5 | 1 | 1 | 1 | | | | | |
| SDR | 0.004 | 0.003 | 6.0 | 0.50 | 0.04 | 2.0 | | | | | |
| $r^2$ | 0.99 | 0.99 | 0.98 | 0.99 | 0.99 | 0.99 | | | | | |

Table A1. Properties of the SWTs characterizing the water masses of the SR03 section with the correspondent accuracies (ε). All SWTs are defined with preformed values of the variables (°). *The values of preformed oxygen ($O_2°$) are not in equilibrium for end members representing waters from the Antarctic shelf or old deep waters. The uncertainties in the fractions of the SWTS, the weights given to each variable in the OMP analysis, the Standard Deviation of the Residuals (SDR) and the square correlation coefficient (r2) between the observed values and the OMP estimates are also listed.

- DIC changes: as suggested previously I suggest to start the methods sections with the proposed estimation of DIC changes and driving factors.

$\Delta DIC = \Delta DIC^{BIO} + \Delta DIC^{CANT} + \Delta DIC^{phys}$

please see Álvarez et al. (doi:10.1029/2010JC006475, 2011)

$\Delta DIC$ with your real measurements
$\Delta DIC^{BIO}$ will be computed as your Eq2, this term contains changes in mineralization associated or not with changes in circulation / biology BUT correlated with AOU / nutrients for sure. As you lack of transient tracers information you cannot account for changes in the transport, circulation independently.
$\Delta DIC^{CANT}$ with the OMP method (forget about the dis terms )
$\Delta DIC^{phys}$ as the difference between $DIC - DIC^{CANT} - DIC^{BIO}$, this term would contain any changes in DIC not associated with AOU, mixing / Ventilation…. not accounted for in the CANT-OMP.
Following this methodology the blurred assumptions in page 8, lines 1 to 22 are avoid.

and of course I think it would help to explain the results.

We really thank you for the suggestion, however we consider that the proposed estimation of the DIC changes would not really allow us to avoid the lines in section 3.3, since somewhere we must address the issue of non-steady-state circulation and biological processes, and we believe the current paper structure is the best way to do this. As we mentioned before, we'll add some clarifications in the text at the beginning of the section (see answer to the point : "- circulation and biological processes at steady state").

To respond specifically to your suggestions, the problem with using the proposed estimation of the change of DIC in the present manuscript is that:

- we have a range of values for $\Delta DIC^{CANT}$ (which corresponds to $\frac{\partial C_{ANT}}{\partial t}$ of the present manuscript)

- your recommended $\Delta DIC^{phys}$ is going to have as well a range of values that we will have to explain in the same way that we did with $\frac{\partial C_{ANT\_BC}}{\partial t}$ and $\frac{\partial C_{ANT}}{\partial t}$. This may well create more confusion to the reader. That is why we think that adding the new text to the beginning of section 3.3 will help, since at the end of this additional text we'll mention that:
*"The change in DIC not explained by these two quantities is due to circulation processes."* (see the complete text in the answer to the point : "- circulation and biological processes at steady state").

Besides, in section 5 we discuss about the different drivers (solubility, biology or circulation) that are responsible for the change in DIC not explained by the change in $C_{ANT}$ (see page 12 from line 9 to the end and lines 1-6 in page 13).

The use of the notation proposed in Álvarez et al. (doi:10.1029/2010JC006475, 2011) was good for their study. They use Transient tracer distributions (TTDs) to estimate $C_{ANT}$ (TTDs rely on tracers to stablish the amount of $C_{ANT}$ in the deep ocean) and a numerical model to resolve the circulation effect on the AOU term. Thus, the terms with the different drivers affecting the variability of DIC were fully determined. The study was focussed on SAMW, which allowed for the TTD to give reliable values even using just one tracer (CFC-12), because the residence time of CFC-12 is efficient for waters of the age of SAMW. But this is not our situation, and we have previously explained why transient tracers are beyond the scope of this work.

I hope to have been helpful. Yes, of course. We thank you for your thorough review.

**List of changes done to the manuscript:**

- Correcting typos, text.
- Presentation of the supplementary material in a separate file from the main text.
- Addition of new text as a results of the comments from both reviewers
- Addition of tables and figures and correction of captions as a result of the comments from both reviewers.

**Marked-up manuscript and marked-up supplementary material** (blue highlight referring to reviewer-1 comments and green highlight for the changes referred to reviewer-2, see also the comments)

[revised manuscript text omitted]

Le Jehan, S. and Treguer, P.: Uptake and regeneration ASi/AN/AP ratios in the Indian Sector of the Southern Ocean. Polar Biol. 2: 127-136, 1983.

Lourey, K.J. and Trull, T.W.: Seasonal nutrient depletion and carbon export in the Subantarctic and Polar Frontal Zones of the Southern Ocean south of Australia, J. Geophys. Res., 106, C12, 31,463-31,487, 2001.

Mackas, D.L., Denman, K.L. and Bennett, A.F.: Least squares multiple tracer analysis of water mass composition, Journal of Geophysical Research ,92, C3, 2907‑2918, doi: 10.1029/JC092iC03p02907, 1987.

Martiny, A.C., et al.: Strong latitudinal patterns in the elemental ratios of marine plankton and organic matter, Nature, 6, 279-283, doi: 10.1038/NGEO1757, 2013.

Mosby, H.: The waters of the Atlantic Antarctic Ocean, Scientific Results of the Norwegian Antarctic Expeditions 1927-1928, 1, 11, 131 pp, 1934.

Oke, P.R., et al.: Evaluation of a near-global eddy-resolving ocean model, Geosci. Model Dev., 6, 591–615, doi:10.5194/gmd-6-591-2013, 2013.

Orsi, A.H., Smethie Jr., W.M. and Bullister, J.L.: On the total input of Antarctic waters to the deep ocean: A preliminary estimate from chlorofluorocarbon measurements, J. Geophys. Res., 107, C8, 3122, doi: 10.1029/2001JC000976, 2002.

Oschlies A. and Schartau, M.: Basin-scale performance of a locally optimized marine ecosystem model, Journal of Marine Research, 63, 2, 335-358, 2005.

Pardo, P.C., Vázquez-Rodríguez, M., Pérez, F.F. and Ríos, A.F.: $CO_2$ air-sea disequilibrium and preformed alkalinity in the Pacific and Indian Oceans calculated from subsurface layer data, Journal of Marine Systems 84, 67–77, doi:10.1016/j.jmarsys.2010.08.006, 2011.

Pardo, P.C., Perez, F.F., Velo, A. and Gilcoto, M.: Water masses distribution in the Southern Ocean: improvement of an extended OMP (eOMP) analysis. Progress in Oceanography 103, 92‑105, doi: 10.1016/j.pocean.2012.06.00, 2012.

Pardo, P.C., Pérez, F.F., Khatiwala, S. and Ríos, A.F.: Anthropogenic $CO_2$ estimates in the Southern Ocean: Storage partitioning in the different water masses, Progress in Oceanography, 120, 230–242, doi:10.1016/j.pocean.2013.09.005, 2014.

Poole, R. and Tomczak, M.: Optimum multiparameter analysis of the water mass structure in the Atlantic Ocean thermocline, Deep Sea Res., Part I, 46, 1895– 1921, doi:10.1016/S0967-0637(99)00025-4, 1999.

Rintoul, S.R. and Bullister, J.L.: A late winter hydrographic section from Tasmania to Antarctica, Deep –Sea Research I, 46, 1417-1454, 1999.

Sabine, C.L., et al.: The Oceanic Sink for Anthropogenic $CO_2$, Science, 305, 5682, 367–371, 2004.

Sokolov, S. and Rintoul, S.R.: Circulation and water masses of the southwest Pacific: WOCE Section P11, Papua New Guinea to Tasmania, Journal of Marine Research, 58, 223–268, 2000.

Talley, L.D., Pickard, G.L., Emery, W.J. and Swift, J.H.: Descriptive Physical Oceanography: An Introduction (Sixth Edition), Elsevier, Boston, 560 pp, 2011.

Thompson, R.O. and Edwards, R.J., 1981. Mixing and water-mass formation in the Australian Subantarctic. Journal of Physical Oceanography 11, 1399‑1406, 1981.

Tomczak, M.: A multi-parameter extension of temperature/salinity diagram techniques for the analysis of non-isopycnal mixing, Progress in Oceanography, 10, 147–171, 1981.

Tomczak, M. and Large, D.G.B.: Optimum Multiparameter Analysis of Mixing in the Thermocline of the Eastern Indian Ocean Journal of Geophysical Research, 94, Cll, 16141-16149, 1989.

Verlencar, X.N., Somasunder, K. and Qasim, S.Z.: Regeneration of nutrients and biological productivity in Antarctic waters, Marine Ecology Progress Series, 61, 41-59, 1990.

Whitworth III, T., Orsi, A.H., Kim, S.-J. and Nowlin, W.D.: Water masses and mixing near the Antarctic Slope Front. Antarctic Research Series, 75, 1–27, 1998.

**A Tables Supplementary material**

| | θ (°C) | S | SiO$_4$ (μmol/kg) | NO$_3^0$ (μmol/kg) | PO$_4^0$ (μmol/kg) | O$_2^{0*}$ (μmol/kg) | TA$^0$ (μmol/kg) | DIC$^{\pi}_{SAT}$ (μmol/kg) | CDIS (μmol/kg) | CDIS$^{\pi}$ (μmol/kg) | Fractions uncertainties (%) |
|---|---|---|---|---|---|---|---|---|---|---|---|
| SWT$_{STW16}$ | 16 ± 0.06 | 35.1 ± 0.07 | 0.9 ± 0.2 | 1.2 ± 0.2 | 0.04 ± 0.3 | 243 ± 2 | 2290 | 1990 | -19 | 1 | 0.04 |
| SWT$_{STW15}$ | 15 ± 0.06 | 35.66 ± 0.07 | 0.6 ± 0.2 | 0 ± 0.2 | 0.12 ± 0.3 | 247 ± 2 | 2328 | 2026 | -22 | -2 | 0.04 |
| SWT$_{AASW}$ | -1.85 ± 0.006 | 33.8 ± 0.005 | 45 ± 2 | 30.7 ± 0.2 | 2.10 ± 0.3 | 360 ± 4 | 2289 | 2137 | -23 | -19 | 0.06 |
| SWT$_{SASW}$ | 5 ± 0.008 | 33.8 ± 0.03 | 3 ± 0.2 | 23.3 ± 0.3 | 1.55 ± 0.5 | 310 ± 3 | 2264 | 2064 | -13 | -4 | 0.06 |
| SWT$_{HSSW}$ | -1.91 ± 0.08 | 34.71 ± 0.006 | 80 ± 1 | 28.3 ± 0.08 | 2.02 ± 0.03 | 300 ± 3 | 2351 | 2188 | -21 | 0 | 0.08 |
| SWT$_{SAMW}$ | 8.8 ± 0.02 | 34.63 ± 0.03 | 6 ± 0.6 | 13.2 ± 0.6 | 0.92 ± 0.8 | 280 ± 7 | 2290 | 2053 | -10 | 2 | 0.03 |
| SWT$_{AAIW}$ | 4 ± 0.01 | 34.35 ± 0.02 | 34 ± 2 | 29.2 ± 0.4 | 1.97 ± 0.9 | 220 ± 8 | 2299 | 2099 | -16 | -6 | 0.04 |
| SWT$_{NADW}$ | 3.28 ± 0.008 | 34.91 ± 0.003 | 28 ± 1 | 27.5 ± 0.3 | 1.19 ± 0.7 | 220 ± 4 | 2355 | 2152 | -27 | -10 | 0.08 |
| SWT$_{CDW}$ | 0.65 ± 0.006 | 34.707 ± 0.003 | 115 ± 7 | 30.8 ± 0.1 | 2.12 ± 0.1 | 220 ± 3 | 2351 | 2168 | -23 | -2 | 0.03 |
| SWT$_{PIDW}$ | 1.44 ± 0.008 | 34.75 ± 0.005 | 125 ± 3 | 34.2 ± 0.01 | 3.4 ± 0.02 | 96 ± 2 | 2360 | 2168 | -24 | -4 | 0.04 |
| SWT$_{AABW}$ | -0.6 ± 0.006 | 34.66 ± 0.006 | 130 ± 5 | 30.7 ± 0.08 | 2.13 ± 0.03 | 259 ± 3 | 2355 | 2181 | -22 | -2 | 0.05 |
| Weights | 20 | 10 | 0.5 | 1 | 1 | 1 | | | | | |
| SDR | 0.004 | 0.003 | 6.0 | 0.50 | 0.04 | 2.0 | | | | | |
| r$^2$ | 0.99 | 0.99 | 0.98 | 0.99 | 0.99 | 0.99 | | | | | |

**Table A1.** Properties of the SWTs characterizing the water masses of the SR03 section with the correspondent accuracies (ε). All SWTs are defined with preformed values of the variables ($^0$). The values of preformed oxygen (O$_2$°) are not in equilibrium for end members representing waters from the Antarctic shelf or old deep waters. The uncertainties in the fractions of the SWTS, the weights given to each variable in the OMP analysis, the Standard Deviation of the Residuals (SDR) and the square correlation coefficient (r$^2$) between the observed values and the OMP estimates are also listed.

**Mixing Groups**

STW15 + SAMW + SASW + STW16

SASW + SAMW + AAIW + AASW

SAMW + AAIW + NADW

AAIW + PIDW + NADW

AASW + AAIW + CDW

CDW + AABW + HSSW + AASW

**Table A2. Mixing groups used in the OMP analysis. Note that only the subscripts of the names of the SWTs are written, i.e., STW15 instead of $SWT_{STW15}$.**

| SWTs | Parameterizations | Reference |
|---|---|---|
| $SWT_{STW16}$ | $AT^\circ$ ($\pm6$) = 2288.3 + 62.8*(S - 35) - 0.9*($\theta$ -16) + 0.1*(PO - 300)
$CDIS$ ($\pm5$) = -47.9 + 2.31*$\theta$ + 0.16*(PO - 300) | Parameterizations for the South Subtropical region-Pacific Ocean (SSTIP, Pardo et al., 2011) |
| $SWT_{STW15}$ | $AT^\circ$ ($\pm6$) = 2288.3 + 62.8*(S - 35) - 1.6*($\theta$ -16) + 0.1*(PO - 300)
$CDIS$ ($\pm5$) = 51.3 + 2.31*$\theta$ + 0.16*(PO - 300) | Parameterizations for the South Subtropical region-Indian Ocean (SSTIP, Pardo et al., 2011) |
| $SWT_{HSSW}$*

$SWT_{AASW}$

$SWTS_{SASW}$ | $AT^\circ$ ($\pm4$) = 2296.7 + 94.7*(S - 35) + 0.3*(PO - 300)
$CDIS$ ($\pm5$) = $-84.3 - 12.95$*(S - 35) + 5.75*$\theta$ + 0.17*(PO - 300) | Parameterizations for the Antarctic region (AAIP, Pardo et al., 2011)
*(Dlugokencky, et al., 2016) |
| $SWT_{SAMW}$

$SWT_{AAIW}$ | $AT^\circ$[1] ($\pm6$) = 2288.3 + 62.8*(S - 35) -(0.9//1.6)*($\theta$ - 16) + 0.1*(PO - 300)
$AT^\circ$[2] ($\pm4$) = 2296.7 + 94.7*(S - 35) + 0.3*(PO - 300)
$CDIS$[1] ($\pm5$) = -47.9 + 2.31*$\theta$ + 0.16*(PO - 300)
$CDIS$[2] ($\pm5$) = $-84.3 - 12.95$*(S - 35) + 5.75*$\theta$ + 0.17*(PO - 300) | Parameterizations for the South Subtropical regions and for the Antarctic region (SSTIP, Pardo et al., 2011) |
| $SWT_{NADW}$ | ---------------------------------- | GLODAPv2 // Pardo et al. (2014) ** |
| $SWT_{CDW}$ | $AT^\circ$ // $CDIS$ = 0.69*($AT^\circ$ // $CDIS$)$_{AABW}$ + 0.26*($AT^\circ$ // $CDIS$)$_{NADW}$ + 0.05*($AT^\circ$ // $CDIS$)$_{AAIW}$ | -------------------------------- |
| $SWT_{PIDW}$ | $AT^\circ$ // $CDIS$ = 0.71*($AT^\circ$ // $CDIS$)$_{CDW}$ + 0.25*($AT^\circ$ // $CDIS$)$_{NADW}$ + 0.04*($AT^\circ$ // $CDIS$)$_{AAIW}$ | -------------------------------- |
| $SWT_{AABW}$ | $AT^\circ$ // $CDIS$ = 0.51*($AT^\circ$ // $CDIS$)$_{CDW}$ + 0.44*($AT^\circ$ // $CDIS$)$_{HSSW}$ + 0.05*($AT^\circ$ // $CDIS$)$_{AASW}$ | -------------------------------- |

**Table A3. Parameterizations for estimating $AT^0$ and CDIS in each of the SWTs. PO =$R_P PO_4$ + $O_2$. For $SWT_{HSSW}$ *, CDIS is obtained from the NOAA database of latitudinal mean values of atmospheric $CO_2$. For $SWT_{NADW}$**, $AT^0$ is obtained from the climatology of GLODAPv2 and CDIS from Pardo et al. (2014).**

**B Figures Supplementary Material**

[Figure]

**Fig. A1. (a)** T/S diagram, with θ = potential temperature. Blue dots represent the data from all cruises for the period 1995-2011. Dotted lines indicate density ($\sigma_0$, kg m$^{-3}$). **(b)** SiO$_4$/T diagram, with θ = potential temperature. Green squares represent the data from all cruises for the period 1995-2011. Pentagrams represents the SWTs included in the OMP analysis (only subscripts are written, i.e., AASW instead of SWT$_{AASW}$).

[Figure]

A2. Seasonal values of the observation-based SAM index. The smooth black curve shows decadal variations. Figure obtained from Marshall, Gareth & National Center for Atmospheric Research Staff (Eds). "The Climate Data Guide: Marshall Southern Annular Mode (SAM) Index (Station-based)." Retrieved from https://climatedataguide.ucar.edu/climate-data/marshall-southern-annular-mode-sam-index-station-based.